# Contextual Thompson Sampling
# via Generation of Missing Data

**Kelly W. Zhang**
Department of Mathematics
Imperial College London
kelly.zhang@imperial.ac.uk

**Tiffany (Tianhui) Cai**
Department of Statistics
Columbia University
tiffany.cai@columbia.edu

**Hongseok Namkoong**
Decision, Risk, and Operations
Columbia Business School
namkoong@gsb.columbia.edu

**Daniel Russo**
Decision, Risk, and Operations
Columbia Business School
djr2174@columbia.edu

## Abstract

We introduce a framework for Thompson sampling (TS) contextual bandit algorithms, in which the algorithm's ability to quantify uncertainty and make decisions depends on the quality of a generative model that is learned offline. Instead of viewing uncertainty in the environment as arising from unobservable latent parameters, our algorithm treats uncertainty as stemming from missing, but potentially observable outcomes (including both future and counterfactual outcomes). If these outcomes were all observed, one could simply make decisions using an "oracle" policy fit on the complete dataset. Inspired by this conceptualization, at each decision-time, our algorithm uses a generative model to probabilistically impute missing outcomes, fits a policy using the imputed complete dataset, and uses that policy to select the next action. We formally show that this algorithm is a generative formulation of TS and establish a state-of-the-art regret bound. Notably, our regret bound depends on the generative model only through the quality of its offline prediction loss, and applies to any method of fitting the "oracle" policy.

## 1 Introduction

Recent advances in machine learning have transformed our ability to develop high quality predictive and generative models for complex data. This work introduces a framework for developing decision-making algorithms, specifically for contextual bandit problems, that can take advantage of these machine learning advances. By design, we assume the algorithm developer is able to apply these techniques (e.g., minimize a loss via gradient descent) and employ these methods as subroutines in our decision-making algorithm. Moreover, our theory formally connects the quality of effective (self-)supervised learning via loss minimization to the quality of decision-making.

Classically, Thompson sampling (TS) algorithms form a parametric model of the environment and consider the decision-maker's uncertainty as arising from unknown latent parameters of that model [Thompson, 1933, Russo et al., 2020]. The primitive operations used by TS include i) specifying an informative prior for the latent parameter using domain knowledge, ii) sampling from the posterior distribution of the latent parameter, and iii) updating the posterior distribution as more data is collected. Unfortunately, it is well known that all three of these operations are non-trivial to perform with neural networks [Tran et al., 2020, Goan and Fookes, 2020]. In this work, we view missing, but potentially observable, counterfactual outcomes as the source of the decision-maker's uncertainty. This perspective allows us to replace the primitive operations required in the classical view with new

39th Conference on Neural Information Processing Systems (NeurIPS 2025).

ones that are more compatible with neural networks, namely the ability to i) effectively minimize an offline prediction loss, ii) autoregressively generate from a learned sequence model, and iii) fit a desired policy given access to a complete dataset (outcomes from all actions and decision-times).

In the missing data view of uncertainty, if we had a complete dataset, there is no uncertainty because we could simply use the entire dataset to fit a desired "oracle" policy to use to make optimal decisions for that task. Inspired by this idea, at each decision time our algorithm imputes missing outcomes using a pretrained generative model, fits a desired policy using the imputed complete dataset, and selects the best action according to the fitted policy. *We show that this algorithm is a generative implementation of TS.* We demonstrate empirically how to learn a generative model to impute missing outcomes using standard machine learning tools in *meta-bandit* settings, where the algorithm learns from data from previous tasks to perform well on a new task from the same distribution.

We prove a state-of-the-art regret bound for generative TS with three key properties, which each have significant practical implications. First, the generative model used to impute missing outcomes only affects our bound through the offline prediction loss of the model. This means that our theory is applicable to any imputation model architecture, and that the quality of the generative model can be easily optimized for and evaluated via offline training and validation. Second, our bound is unique in that it applies to any procedure for fitting a desired "oracle" policy. This allows one to easily adapt TS to decision-making problems with constraints, e.g., for fairness or balancing. Finally, our proof approach makes important improvements to previous information theoretic analyses, which may be broadly applicable: i) we accommodate infinite policy classes directly without discretization, and ii) our bound quantifies the benefit of prior task information, such as side information on the actions. Our results hold quite generally and do not require restrictions on generative model or policy class. We demonstrate a practical implementation of our framework in Sections 4 and 6.

## 2 Problem formulation

**Meta-contextual bandit problem.** Let bandit tasks $\tau$ be sampled from an unknown distribution $p^*$:

$$\tau \sim p^*, \quad \text{where} \quad \tau = \{Z_\tau, X_{1:T}, \{Y_1^{(a)}, \ldots, Y_T^{(a)}\}_{a \in \mathcal{A}_\tau}\}, \tag{1}$$

where each bandit task $\tau$ consists of prior task information $Z_\tau$, action space $\mathcal{A}_\tau$, context vectors $X_{1:T} = \{X_1, \ldots, X_T\}$, and potential outcomes $\{Y_{1:T}^{(a)}\}_{a \in \mathcal{A}_\tau} = \{Y_1^{(a)}, \ldots, Y_T^{(a)}\}_{a \in \mathcal{A}_\tau}$ [Rubin, 2005]; see Figure 1 for a depiction. We omit subscripting $X_t$ and $Y_t^{(a)}$ with $\tau$ to reduce clutter. Note, in contrast to the design-based inference literature [Neyman, 1992], which conditions on the potential outcomes and treats them as non-random, we assume the potential outcomes $\tau$ are drawn from a task distribution $p^*$. Informally, the agent's objective is to se-

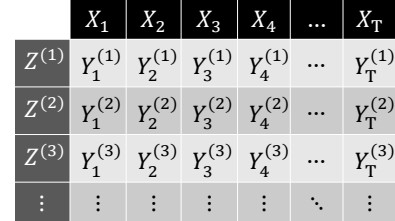

| | $X_1$ | $X_2$ | $X_3$ | $X_4$ | ... | $X_T$ |
|---|---|---|---|---|---|---|
| $Z^{(1)}$ | $Y_1^{(1)}$ | $Y_2^{(1)}$ | $Y_3^{(1)}$ | $Y_4^{(1)}$ | ... | $Y_T^{(1)}$ |
| $Z^{(2)}$ | $Y_1^{(2)}$ | $Y_2^{(2)}$ | $Y_3^{(2)}$ | $Y_4^{(2)}$ | ... | $Y_T^{(2)}$ |
| $Z^{(3)}$ | $Y_1^{(3)}$ | $Y_2^{(3)}$ | $Y_3^{(3)}$ | $Y_4^{(3)}$ | ... | $Y_T^{(3)}$ |
| $\vdots$ | $\vdots$ | $\vdots$ | $\vdots$ | $\vdots$ | $\ddots$ | $\vdots$ |

Figure 1: Potential outcomes table for a task $\tau$.

lect actions to maximize the total expected reward for each encountered task. At the start of a task, the agent observes prior task information $Z_\tau$. For each timestep $t \in [1:T]$, the agent observes context $X_t$, selects action $A_t \in \mathcal{A}_\tau$, observes outcome $Y_t = Y_t^{(A_t)}$, and computes reward $R(Y_t)$, for a fixed, known function $R$ in $[0, 1]$. The history, $\mathcal{H}_t = \{Z_\tau, (X_1, A_1, Y_1), \ldots, (X_{t-1}, A_{t-1}, Y_{t-1}), X_t\}$, includes the current context $X_t$. In contrast to much of the Bayesian contextual bandit literature [Lattimore and Szepesvári, 2019, Russo et al., 2020], **we do not make parametric assumptions** about the distribution of outcomes $Y$ conditional on contexts $X$ and prior task information $Z$.

The agent is able to learn both online *within a single task* (i.e., over the $T$ total decision times), as well as meta-learn *across different tasks* (e.g., learning how task prior information $Z_\tau$ may inform the distribution of $\{Y_{1:T}^{(a)}\}_{a \in \mathcal{A}_\tau}$). The algorithm has access to training data collected from previous tasks, sampled from (1). These previous bandit tasks can be used by the algorithm to meta-learn across tasks, i.e., learn about the distribution $p^*$ itself to improve decision-making quality. Our algorithm's decision-making quality depends on how accurately the agent is able to model the task distribution, as well as the policy fitting procedure the algorithm designer chooses. Rather than relying on strong assumptions on the environment structure, we put the onus on the algorithm designer to i) learn a generative model that accurately captures the environment structure of the meta-bandit task at hand, and ii) choose a meaningful method for fitting a desired "oracle" policy, assuming access

to a complete dataset. Since generative models learned offline routinely perform much better than expected according to existing theory, our theory focuses on formal reductions of decision-making quality to offline learning.

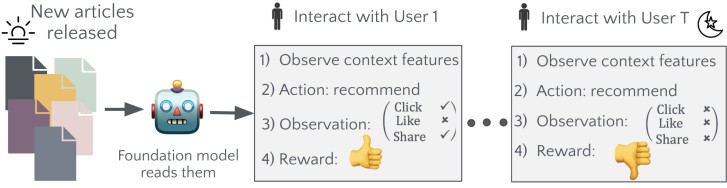

Figure 2: News recommendation meta contextual bandit problem.

**Motivating example: News recommendation.** As depicted in Figure 2, a motivating meta-contextual bandit problem is cold-start news recommendations. Each day, a new set of articles $\mathcal{A}_\tau$ is released, which the agent recommends to users who arrive throughout the day. In contrast to Li et al. [2010], our algorithm meta-learns across news recommendation tasks and uses the article text to improve cold-start decisions. We use $Z_\tau = (Z_\tau^{(a)})_{a \in \mathcal{A}_\tau}$ to denote the *task-specific* prior information; for example, for article $a \in \mathcal{A}$, $Z_\tau^{(a)}$ could be the news article text or other article meta-data (category, style, etc.). The context variables $X_t$ consist of user-specific features, and $Y_t$ are recommendation outcomes observed following the $t^{\text{th}}$ decision. The modern challenge in this setting is that incorporating news article text $Z_\tau$ can greatly improve the recommendation system's decisions, but a foundation model is needed to process this high dimensional text and inform decision-making. This motivates us to i) make very minimal structural assumptions on the relationship between prior information $Z_\tau$, context features $X_t$, and outcomes $Y_t$, and ii) develop an algorithm that can leverage foundation models.

**Policy fitting.** The algorithm designer specifies a procedure for fitting a desired "oracle" policy given access to a complete bandit task dataset $\tau$. This fitting procedure outputs policies in a function class $\Pi$ where each $\pi \in \Pi$ defines a mapping from contexts $X_t$ to an action $a \in \mathcal{A}_\tau$ that does not vary over time. For notational simplicity, the policies in $\Pi$ are assumed to be non-stochastic. Note that we *do not* require that this policy class is "correct". For a particular task $\tau$, we use $\pi^*(\,\cdot\,; \tau)$ to denote a "best-fitting" policy $\pi^* \in \Pi$, where the fitting criterion is defined by the algorithm designer. For example, consider a simple least squares criterion: $\operatorname{argmin}_{\pi \in \Pi} \frac{1}{T} \sum_{t=1}^{T} \left\{ R(Y_t^{(\pi(X_t))}) - \max_{a \in \mathcal{A}_\tau} R(Y_t^{(a)}) \right\}^2$.

One should think of $\pi^*(\,\cdot\,; \tau)$ as the policy one *would* implement if abundant task data, $\tau$, were available. This could involve fitting a model, adding prompt tokens to condition a language model, or maximizing hindsight performance. This policy fitting can also incorporate constraints on the policy, e.g., to ensure fairness. We aim to match this policy's performance via efficient interactive learning.

**Regret.** We consider a best-in-class style regret objective, which is common in the contextual bandit literature [Foster et al., 2020, 2019, Langford and Zhang, 2007, Agarwal et al., 2017]. The objective of the agent $\mathbb{A}$ is to make decisions to minimize the per-period regret against the best-in-hindsight policy $\pi^*(\cdot; \tau)$:

$$\Delta(\mathbb{A}) = \mathbb{E}\left[ \frac{1}{T} \sum_{t=1}^{T} \left\{ R(Y_t^{(\pi^*(X_t; \tau))}) - R(Y_t^{(A_t)}) \right\} \right]. \tag{2}$$

The "best-fitting" or "best-in-hindsight" policy $\pi^*(\cdot; \tau)$ is well-defined and is a well-established concept in the bandit literature, representing a generalization of the optimal policy. Refer, for example, to Section 2 of Beygelzimer et al. [2011] and Chapter 4 of Bubeck et al. [2012] for very analogous objectives. We emphasize that our algorithm does not have access to the best-fitting policy, which would trivialize the problem. They have access to a means to compute the best-fitting policy if they had $\tau$, i.e., observed rewards of every arm in every context.

The expectation in (2) averages over tasks $\tau \sim p^*$ and any randomness in how the algorithm selects actions. $\Delta(\mathbb{A})$ is the long-run per-period regret if the algorithm was deployed across many tasks. Note, increasing the complexity of the policy class $\Pi$ increases the average reward under the best-fitting policy, $\mathbb{E}\left[ \frac{1}{T} \sum_{t=1}^{T} R(Y_t^{(\pi^*(X_t; \tau))}) \right]$. However, this increased complexity also means that large sample sizes are required to learn $\pi^*(\,\cdot\,; \tau)$ accurately and will worsen our regret bound (Section 3.2).

# 3 Generative Thompson Sampling: General algorithm and regret bounds

**Posterior sampling via imputing missing data.** In this work, we view missing data as the source of the decision-maker's uncertainty. This contrasts the classical approach of considering unknown model parameters as the source of uncertainty. As we will explore in the following sections, the missing data viewpoint is very amenable to modern deep learning methods, which can be used to train models that are able to impute missing data probabilistically in a calibrated fashion. First, consider an idealized setting in which we have the true meta task distribution $p^*$. Using $p^*$ we can form exact posteriors sample for task outcomes $\tau = \{Z_\tau, X_{1:T}, \{Y_{1:T}^{(a)}\}\}$ given the history $\mathcal{H}_t$:

$$\hat{\tau}_t \sim p^*\left(\tau \in \cdot \mid \mathcal{H}_t\right). \tag{3}$$

Above, we probabilistically generate values in $\tau$ that have not yet been observed in the history $\mathcal{H}_t$; This consists of future contexts, future outcomes, and outcomes from previous timesteps for actions that were not selected. We discuss how to practically implement such sampling in Section 4. Note, even when $p^*$ is known, $\hat{\tau}_t$ is simply a calibrated posterior sample and is not equivalent to the true $\tau$.

With this exact posterior sample, $\hat{\tau}_t$, we can form posterior samples of any statistic computed using $\hat{\tau}_t$. In particular, we are interested in sampling from the posterior distribution of the fitted policy $\pi^*(\,\cdot\,; \tau)$, which can be computed by finding the fitted policy for the sampled task dataset $\hat{\tau}_t$, i.e., $\pi^*(\,\cdot\,; \hat{\tau}_t)$. Posterior sampling of a best-fitting policy is a common subroutine used in Bayesian decision-making algorithms [Kaufmann et al., 2012, Russo and Van Roy, 2018, Ryzhov et al., 2012]. Thus, our posterior sampling approach can easily integrate with these existing Bayesian algorithms.

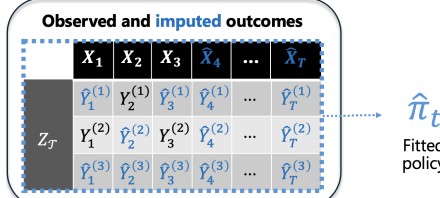

Figure 3: The agent imputes missing outcomes and uses the imputed dataset to fit a policy.

In this work, we focus on Thompson sampling [Russo and Van Roy, 2016, Thompson, 1933], i.e., probability matching, which selects actions according to the posterior probability that they are optimal. Thompson Sampling (TS) can be implemented with a single posterior sample per decision time. In our generative implementation of TS (Algorithm 1) at decision time $t$, after sampling $\hat{\tau}_t$ as in (3), TS fits the policy $\pi^*(\,\cdot\,; \hat{\tau}_t)$, and selects the action $A_t \leftarrow \pi^*(X_t; \hat{\tau}_t)$. See Figure 3 for a depiction. Our algorithm generalizes TS by replacing the true reward-maximizing policy with a best-fitting policy under a given policy class $\Pi$. A more "standard" TS algorithm is recovered when the best-fitting policy is correctly specified. See the discussion below display (2) for more on the best-fitting policy.

---

**Algorithm 1** Generative Thompson Sampling

**Require:** Imputation model $p$, actions $\mathcal{A}_\tau$, task input $Z_\tau$.
1: **for** $t \in \{1, \dots, T\}$ **do**
2:     Observe context $X_t$ and append it to $\mathcal{H}_t$
3:     Generate / sample $\hat{\tau}_t \sim p(\tau \in \cdot \mid \mathcal{H}_t)$
4:     Fit the policy $\pi^*(\,\cdot\,; \hat{\tau}_t)$
5:     Select the action $A_t \leftarrow \pi^*(X_t; \hat{\tau}_t)$
6:     Observe outcome $Y_t \leftarrow Y_t^{(A_t)}$
7:     Update history $\mathcal{H}_{t+1} \leftarrow \mathcal{H}_t \cup \{(X_t, A_t, Y_t)\}$
8: **end for**

---

Under our generative TS Algorithm 1, the polices in $\Pi$ that are best-in-class optimal under some likely generation of $\hat{\tau}_t$ have a chance of being selected. Once no plausible sample of missing outcome $\hat{\tau}_t$ could result in an action being optimal, it is essentially written off. We formalize that our generative algorithm aligns with the abstract definition of Thompson Sampling (probability matching) in Proposition 1 below when using the correct model $p^*$. See Chapter 36.5 of Lattimore and Szepesvári [2020] for further discussion of the probability matching definition of Thompson Sampling.

**Proposition 1** (Algorithm 1 Implements Thompson Sampling)**.** *Algorithm 1 with imputation model $p^*$ implements Thompson Sampling (probability matching), i.e., the following holds almost surely:*

$$\mathbb{P}(A_t = a \mid \mathcal{H}_t) = \mathbb{P}(\pi^*(X_t; \tau) = a \mid \mathcal{H}_t).$$

A key to proving Proposition 1 is showing that $\mathbb{P}(\pi^*(X_t; \tau) = a \mid \mathcal{H}_t) = \mathbb{P}(\pi^*(X_t; \hat{\tau}_t) = a \mid \mathcal{H}_t)$, which holds when $\hat{\tau}_t$ is sampled from the true meta task distribution $p^*$ as in (3). See Appendix A.2 for our proof.

## 3.1 Regret when using a perfectly calibrated imputation model $p^*$.

We develop a novel analysis of contextual TS, which is applicable to infinite policy classes $\Pi$ with finite VC dimension. Our VC dimension bound resembles those from adversarial bandits, but for the first time, we show we can derive this using an information theoretic analysis. We first present a regret bound for Algorithm 1 with a perfectly calibrated imputation model, $p^*$ from (1), and extend to approximate imputation models in Section 3.2. Note that assuming $p^*$ is known is akin to assuming the prior and likelihood of a Bayesian model are known, which is standard in Bayesian regret analyses.

**Notation.** Let $\boldsymbol{\pi}^*(X_{1:T}) := \{\pi^*(X_t; \tau)\}_{t=1}^T$ be the best fitting policy evaluated at contexts $X_{1:T}$. Let $H(Y \mid X)$ denote the conditional entropy of $Y$ (discrete) given $X$; note $H(Y \mid X) = -\mathbb{E}[\sum_y \mathbb{P}(Y = y \mid X) \log \mathbb{P}(Y = y \mid X) dy]$ is a constant. Let $I(Y; X \mid Z)$ be the mutual information between $Y$ and $X$ conditional on $Z$; note $I(Y; X \mid Z)$ marginalizes $Z$ and is also a constant.

**Theorem 1** (Regret bound for Generative TS with a perfectly calibrated imputation model $p^*$). *For Algorithm 1 with imputation model $p^*$, $\mathbb{A}_{\text{TS-Gen}}(p^*)$,*

$$\Delta(\mathbb{A}_{\text{TS-Gen}}(p^*)) \leq \sqrt{\frac{|\mathcal{A}_\tau|}{2T} \cdot H(\boldsymbol{\pi}^*(X_{1:T}) \mid Z_\tau)}.$$

*Moreover, $\Delta(\mathbb{A}_{\text{TS-Gen}}(p^*)) \leq \sqrt{\frac{\bar{\Gamma}}{T} \cdot H(\boldsymbol{\pi}^*(X_{1:T}) \mid Z_\tau)}$, where $\bar{\Gamma}$ bounds the information ratio [Russo and Van Roy, 2016], i.e., $\bar{\Gamma} \geq \max_t \Gamma_t$ a.s. for $\Gamma_t := \frac{\mathbb{E}[R(Y_t^{(\pi^*(X_t;\tau))}) - R(Y_t^{(A_t)}) \mid \mathcal{H}_t]^2}{I(\pi^*(X_t;\tau); Y_t^{(A_t)}, A_t \mid \mathcal{H}_t)}$.*

Note $\bar{\Gamma}$ can be smaller than $|\mathcal{A}_\tau|/2$ when feedback from one action informs learning about other actions (Appendix A.6). The entropy, $H(\boldsymbol{\pi}^*(X_{1:T}) \mid Z_\tau)$, quantifies the benefit of using prior information $Z$. Our bound automatically applies to infinite policy classes since it only depends on the entropy of the optimal policy evaluated at a finite number of contexts, $\boldsymbol{\pi}^*(X_{1:T})$.

**Upper bounding the condition entropy using VC dimension.** We can construct a coarse upper bound for the entropy $H(\boldsymbol{\pi}^*(X_{1:T}) \mid Z_\tau)$ using the VC dimension of the policy class $\Pi$. The VC dimension is a worst-case quantity that has to with the total number of possible assignments of actions given contexts. In contrast, entropy reflects uncertainty based on the task distribution (learned from past tasks) and the information $Z$ (e.g., article texts), as many assignments may be extremely unlikely to be optimal. Since VC dimension is only defined for binary functions, we use the multiclass generalization Nataranjan dimension [Natarajan, 1989] when $|\mathcal{A}_\tau| > 2$.

**Proposition 2** (Complexity bound on entropy). *For policy class $\Pi$ over action space $\mathcal{A}_\tau$ with Nataranjan dimension $d$ (equivalent to VC dimension when $|\mathcal{A}_\tau| = 2$),*

$$H(\boldsymbol{\pi}^*(X_{1:T}) \mid Z_\tau) \leq H(\boldsymbol{\pi}^*(X_{1:T})) = O(d \cdot \log(T \cdot |\mathcal{A}_\tau|)).$$

Note, our bound above depends on the Natarajan dimension of the policy class $\Pi$, **not** the Natarajan dimension of the generative sequence model $p^*$. Furthermore, the Natarajan dimension of $\Pi$ does not change with $T$ for stationary policies. A feature of our result is that our bound, when combined with Theorem 1, can be used to derive regret bounds for a wide range of policy classes $\Pi$.

Using Proposition 2, our regret bound (Theorem 1) resembles adversarial regret bounds that depend on VC dimension [Beygelzimer et al., 2011], showing for the first time how such a result can be established through information theoretic arguments.

**Benefits of our approach and relationship to related work.** Regret bounds for contextual TS bandits with infinite policy classes have been of great interest in the literature. The predominant approach to generalizing information-theoretic analyses for TS beyond multi-armed bandits requires discretizing a latent parameter space [Dong and Van Roy, 2018, Gouverneur et al., 2024, Neu et al., 2022, Min and Russo, 2023] and uses cover-number arguments; our proof approach notably does not require any discretization. Furthermore, our bound can be applied broadly, while existing approaches like Neu et al. [2022], Min and Russo [2023] depends on the entropy of a latent environment parameter, which is only applicable to parametric bandits. By Proposition 2, *our result can directly be applied to infinite policy classes by leveraging existing VC dimension bounds*, e.g., for decision

trees [Asian et al., 2009]. In parametric, stationary bandit settings, our result approximately matches (up to log factors) existing Bayesian regret bounds for linear logistic bandits [Neu et al., 2022] and matches up to a factor of $\sqrt{d}$ and log factors bounds for linear non-contextual bandits [Russo and Van Roy, 2018, Dong and Van Roy, 2018] (Appendix A.6). Finally, though we do not explore it much in this work, since we make minimal assumptions on $p^*$, Theorem 1 applies to nonstationary bandit environments. While the oracle policy $\pi^*$ cannot be time-varying, $\pi^*$ can *effectively* vary over time by including the timestep $t$ as a context feature in $X_t$.

## 3.2 Regret when using an approximate imputation model $p_\theta$.

We now present a regret bound for generative TS with an approximate generative model $p_\theta$. The result is notable because $p_\theta$ only affects the regret bound through its offline prediction loss, which means the result can be applied to *any* model class. Specifically, our regret bound will depend on the following population-level loss (the expectation below averages over the task distribution $p^*$):

$$\ell(p_\theta) = -\mathbb{E}\big[\log p_\theta\big(X_{1:T}, \{Y_{1:t-1}^{(a)}\}_{a \in \mathcal{A}_\tau} \mid Z_\tau\big)\big]. \tag{4}$$

In Section 4, we discuss training and sampling from learned generative imputation models in practice.

**Theorem 2** (Regret bound for Generative TS with an approximate imputation model). *For Algorithm 1 with imputation model $p_\theta$, $\mathbb{A}_{\mathrm{TS-Gen}}(p_\theta)$,*

$$\Delta\big(\mathbb{A}_{\mathrm{TS\text{-}Gen}}(p_\theta)\big) \leq \underbrace{\sqrt{\frac{|\mathcal{A}_\tau|}{2T} \cdot H(\boldsymbol{\pi}^*(X_{1:T}) \mid Z_\tau)}}_{\text{Regret bound for Thompson sampling}} + \underbrace{\sqrt{2\{\ell(p_\theta) - \ell(p^*)\}}}_{\text{Penalty for sub-optimal prediction}} . \tag{5}$$

What is particularly novel about Theorem 2 is that the analysis holds even when the imputation model $p_\theta$ is misspecified and does not correspond to proper Bayesian inference in any way. Comparing Theorem 2 to Theorem 1 from earlier, we can interpret the "cost" of using an approximate model $p_\theta$ as $\sqrt{2\{\ell(p_\theta) - \ell(p^*)\}}$; This penalty depends on how well $p_\theta$ approximates $p^*$.

**Scaling of loss penalty.** While tight theoretical bounds for the penalty term $\ell(p_\theta) - \ell(p^*)$ currently do not exist for complex models like neural networks, we can draw intuition from simpler settings. Consider a stationary, stochastic, Bayesian bandit problem. In this setting, for parametric Bayesian models, where $p_\theta$ and $p^*$ are exchangeable, posterior predictive distributions [Fortini and Petrone, 2023], classic results by Clarke and Barron [1990] show that the gap $\ell(p_\theta) - \ell(p^*)$ scales like $\log T$, under mild regularity conditions. This sublinear growth occurs because in this stationary setting, the Bayesian model $p_\theta$ is better able to approximate the next outcome as it observes more data (a phenomenon closely related to Bayesian consistency [Kleijn and Van der Vaart, 2012] and how the effect of the prior eventually washes out). The difference $\ell(p_\theta) - \ell(p^*)$ also scales with the amount of data used to learn $p_\theta$; This is closely linked to empirical Bayes methods, i.e., approaches to meta-learn a prior distribution from data. When $p_\theta$ and $p^*$ correspond to posterior predictive distributions of Bayesian models with correctly specified likelihoods, $p_\theta$ and $p^*$ differ only in their initial prior distributions. Existing works bounding the regret of TS with misspecified priors are not directly comparable, as Simchowitz et al. [2021] analyzes a modified version of TS that requires multiple posterior samples per decision time, and Liu and Li [2016] bounds the frequentist regret.

**Related work on generative TS algorithms.** Wen et al. [2021] consider a non-contextual, multi-armed TS algorithm that incorporates a generative outcome model. However, they require modeling latent environment parameters, and their bound requires a history-dependent KL divergence term to be small, which differs from our prediction loss penalty. Cai et al. [2024] proves a regret bound with a similar prediction loss penalty generative TS algorithm with misspecified models for a much simpler multi-armed, non-contextual setting. They do not introduce the concept of a general "oracle" policy fitting procedure, and their result does not apply to infinite policy classes. Moreover, we were not able to directly build on their proof approach because they critically rely on the fact that under $p^*$, unobserved outcomes $Y$ are exchangeable given the history. In contrast, our result does not require exchangeability at all and technically applies even if $p^*$ is not exchangeable (e.g., nonstationary).

**Flexibility and advantages of Generative TS.** Generative TS requires the algorithm designer to choose an imputation model $p_\theta$ and a policy class $\Pi$. The modularity of these two components allows one to easily extend TS to more complex, less standard decision-making problems, e.g., (i) **Nonstationarity** can be accommodated with a $p_\theta$ that models trends over time (see discussion before Section 3.2); (ii) **Correlated outcomes** can be modeled using a $p_\theta$ that captures dependencies between

outcomes across actions or over time; (iii) **Constrained decision-making** can be done by choosing a policy class $\Pi$ satisfying such constraints, e.g., to ensure fairness one can use standard constrained optimization approaches to learning decision rules [Corbett-Davies et al., 2017] (Appendix B.9).

## 4 Practically implementing generative Thompson Sampling

We now introduce an example of how to learn $p_\theta$ and implement generative TS. Our overall framework is depicted in Figure 4: In step 1, we use offline data from previous tasks to learn a $p_\theta$ model; Then in step 2, we use the learned $p_\theta$ model to implement generative TS. Here, $p_\theta$ is a sequence model that we meta-learn by pretraining on historical data from previous tasks. As our theory accommodates any $p_\theta$ architecture, our approach can take advantage of recent advances in generative sequence models [Vaswani et al., 2017].

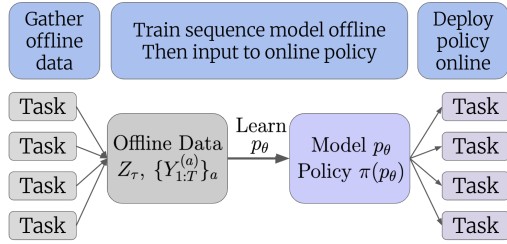

Figure 4: Offline meta-learning and online decision-making across multiple tasks.

### 4.1 Step 1: Offline learning for generative model $p_\theta$.

We now describe learning a generative, sequence model $p_\theta$ from historical data. Our goal is to minimize the loss $\ell(p_\theta)$ from (4). First note that by rules of conditional probabilities, $\ell(p_\theta) =$

$$-\mathbb{E}\left[\sum_{t=1}^{T}\left\{\log p_\theta(X_t|Z_\tau, X_{1:t-1}, \{Y_{1:t-1}^{(a)}\}_{a\in\mathcal{A}_\tau}) + \log p_\theta\big(\{Y_t^{(a)}\}_{a\in\mathcal{A}_\tau}|Z_\tau, X_{1:t}, \{Y_{1:t-1}^{(a)}\}_{a\in\mathcal{A}_\tau}\big)\right\}\right].$$

To make learning $p_\theta$ more practical, the model can make a variety of simplifying approximations. For example, $p_\theta$ could model contexts as evolving independently of past outcomes, i.e., $p_\theta(X_t \mid Z_\tau, X_{1:t-1}, \{Y_{1:t-1}^{(a)}\}_{a\in\mathcal{A}_\tau}) = p_\theta(X_t \mid Z_\tau, X_{1:t-1})$, or model contexts as i.i.d. over time, i.e., $p_\theta(X_t \mid Z_\tau, X_{1:t-1}, \{Y_{1:t-1}^{(a)}\}_{a\in\mathcal{A}_\tau}) = p_\theta(X_t)$. Additionally, $p_\theta$ could model outcomes independently across actions, i.e., $p_\theta(\{Y_t^{(a)}\}_{a\in\mathcal{A}_\tau} \mid Z_\tau, X_{1:t}, \{Y_{1:t-1}^{(a)}\}_{a\in\mathcal{A}_\tau}) = \prod_{a\in\mathcal{A}_\tau} p_\theta(Y_t^{(a)} \mid Z_\tau^{(a)}, X_{1:t}, Y_{1:t-1}^{(a)})$, where $Z_\tau = (Z_\tau^{(a)})_{a\in\mathcal{A}_\tau}$ for action-specific task features $Z^{(a)}$.

Under the chosen simplifying modeling approximations, one can use gradient descent to optimize $p_\theta$ to minimize an empirical loss. For example, in our experiments, our $p_\theta$ makes several simplifying assumptions, and we minimize the following empirical loss to approximately minimize $\ell(p_\theta)$:

$$-\frac{1}{|\mathcal{D}^{\text{offline}}|}\sum_{\tau\in\mathcal{D}^{\text{offline}}}\sum_{t=1}^{T}\left\{\log p_\theta(X_t) + \sum_{a\in\mathcal{A}_\tau}\log p_\theta\big(\{Y_t^{(a)}\}_{a\in\mathcal{A}_\tau} \mid Z_\tau^{(a)}, X_{1:t-1}, Y_{1:t-1}^{(a)}\big)\right\}. \quad (6)$$

Above, $\mathcal{D}^{\text{offline}}$ ideally consists of bandit tasks $\tau \sim p^*$ as described in (1). In practice, one may not have "complete" task datasets $\tau = \{Z_\tau, X_{1:T}, \{Y_1^{(a)}, \dots, Y_T^{(a)}\}_{a\in\mathcal{A}_\tau}\}$, but instead have some partial datasets, e.g., $\{Z_\tau, (X_1, A_1, Y_1), \dots, (X_T, A_T, Y_T)\}$, collected by a behavior policy. In our experiments, we use several heuristics to construct approximate complete tasks $\tilde{\tau}$ from the partial datasets. We use these approximate task datasets to form $\mathcal{D}^{\text{offline}} = \{\tilde{\tau}_1, \tilde{\tau}_2, \tilde{\tau}_3, \dots, \}$. To form $\tilde{\tau}$, we make a simplifying modeling assumption that the tuples $(X_1, Y_1^{(a)}), \dots, (X_T, Y_T^{(a)})$ are exchangeable over time. We then use bootstrap sampling to form approximate complete task datasets $\tilde{\tau}$; see Appendix B.2.2. In this appendix, we also formalize all the simplifying modeling assumptions we make and show how they match standard stochastic contextual bandits with independent actions.

---

**Algorithm 2** Offline training of a sequence model

**Require:** Training data $\mathcal{D}^{\text{offline}}$, model class $\{p_\theta\}_{\theta\in\Theta}$
1: **while** not converged **do**
2:     Sample a mini-batch of tasks $\mathcal{D}^{\text{mini-batch}} \subset \mathcal{D}^{\text{offline}}$
3:     Compute loss in (6) using tasks $\tau \in \mathcal{D}^{\text{mini-batch}}$
4:     Backpropagate and take a gradient step to update $p_\theta$
5: **end while**

---

**Algorithm 3** Posterior sampling via autoregressive generation

**Require:** Sequence model $p_\theta$, actions $\mathcal{A}_\tau$, current timestep $t$, current task history $\mathcal{H}_t$

1: For each $a \in \mathcal{A}_\tau$, define $\mathcal{M}^{(a)}$ as the set of times $i \in [1:T]$ where $Y_i^{(a)}$ was not observed in $\mathcal{H}_t$
2: For each $a \in \mathcal{A}_\tau$, define the ordering $\prec_a$ so that all observed outcomes precede unobserved ones
3: Set $\hat{X}_{1:t} \leftarrow X_{1:t}$ and sample $\hat{X}_{t+1}, \ldots, \hat{X}_T$ from $p_\theta$
4: **for** $a \in \mathcal{A}_\tau$ **do**
5:     **for** $i \in \{1, \ldots, T\}$ in order of $\prec_a$ **do**
6:         **if** $i \notin \mathcal{M}^{(a)}$ **then**
7:             $\hat{Y}_i^{(a)} \leftarrow Y_i^{(a)}$
8:         **else**
9:             Sample $\hat{Y}_i^{(a)} \sim p_\theta(\,\cdot\, \mid Z, \{\hat{X}_j, \hat{Y}_j^{(a)}\}_{j \prec_a i}, \hat{X}_i)$
10:         **end if**
11:     **end for**
12: **end for**
13: **Return:** $\hat{\tau}_t \leftarrow \big\{ Z_\tau, \hat{X}_{1:T}, \{\hat{Y}_{1:T}^{(a)}\}_{a \in \mathcal{A}_\tau} \big\}$

## 4.2 Step 2: Online decision-making using the learned generative model $p_\theta$.

After the sequence model $p_\theta$ is trained offline, it is deployed and used for online decision-making. No additional training of $p_\theta$ is needed. Instead, the sequence model learns from recent online observations "in-context" by conditioning [Brown et al., 2020]. Specifically, to implement the generative step of Generative TS (line 3 of Algorithm 1), we use $p_\theta$ to sample future contexts $X$ and missing outcomes $Y$ to form $\hat{\tau}_t$. We refer to this procedure as *posterior sampling via autoregressive generation*; this is depicted in Figure 5 and formalized in Algorithm 3 below.

In Algorithm 3, we use $\mathcal{M}_a \subset \{1, \ldots, T\}$ to denote the timesteps $t$ for which $Y_t^{(a)}$ has not been observed. When generating outcomes in $\hat{\tau}_t$ for arm $a$, we permute pairs of contexts and outcomes $(X, Y)$ so that observed outcomes always precede missing ones; this way, we always condition on all observed outcomes (and corresponding contexts), matching Figure 5. We use $\prec_a$ to denote this ordering for an action $a \in \mathcal{A}_\tau$; we use $i \prec_a j$ whenever either (a) $i < j$ or (b) $i \notin \mathcal{M}_a$, but $j \in \mathcal{M}_a$.

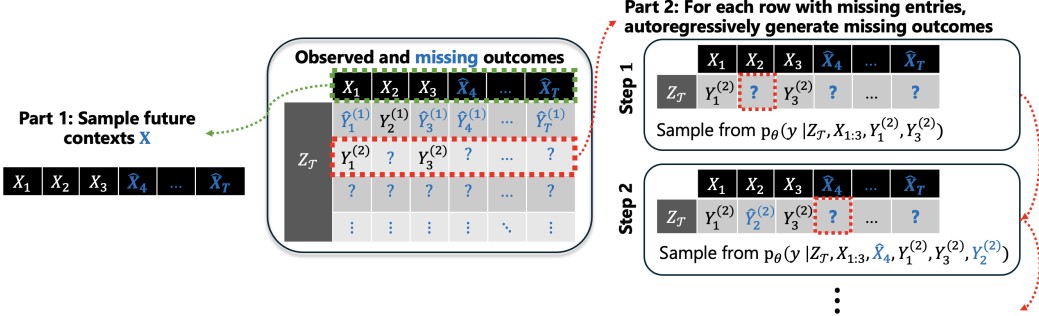

Figure 5: Posterior sampling via autoregressive generation (Algorithm 3).

# 5 Related work

**Decision-making with generative models.** Many recent methods use generative models in decision-making that involve imitation learning, i.e., from demonstrations learn to mimic an expert's actions [Chen et al., 2021, Janner et al., 2021, Hussein et al., 2017]. Lee et al. [2023] discuss how these approaches can be used even without access to expert demonstrations, as long as one is able to fit an approximate "oracle" policy from offline bandit environments. Our work differs significantly from Lee et al. [2023] and other imitation learning based works because our sequence models are used to sample future *outcomes*, instead of predicting optimal actions. Several recent works also use generative models to model future rewards [Mukherjee et al., 2024, Nguyen and Grover, 2022, Müller et al., 2022x, Garnelo et al., 2018, Liu and Li, 2016]. Most previous work on decision-making with sequence models that predict future rewards does not use autoregressive generation

to quantify uncertainty [Mukherjee et al., 2024, Nguyen and Grover, 2022, Müller et al., 2022x, Garnelo et al., 2018]; Instead, their algorithms only consider uncertainty in the single next timestep's reward under each action, e.g., using softmax sampling [Mukherjee et al., 2024]. We find empirically that alternative (non-autoregressive) ways of sampling from the sequence model can lead to inferior decision-making performance (Figure 6).

**(Approximate) TS with neural networks (NN).** Implementing TS with NN has been a longstanding challenge. Riquelme et al. [2018] investigated TS with a variety of Bayesian uncertainty quantification techniques for NN; they found that linear TS with the last layer of a NN as context outperformed many more complex methods. While some TS algorithms directly model uncertainty in NN weights [Zhang et al., 2020, Wang and Zhou, 2020], the foremost approach in the literature implement TS with deep ensembles [Qin et al., 2022, Lu and Van Roy, 2017, Dwaracherla et al., 2020, Osband et al., 2023, Osband and Van Roy, 2015, Osband et al., 2023, Li et al., 2024]. Our generative TS algorithm is critically different from ensembling because a) through offline meta-training we are able to learn informed priors from complex task-specific information $Z$ (like text) with benefits that are explicitly reflected in our bound, and b) our approach allows the generative model to learn *in-context* avoiding retraining online using gradient updates on sub-sampled data, which is sensitive to learning rates.

**Meta-bandits.** In the bandit literature, many algorithms have been proposed for meta-learning settings. Many prior works focus on a different setup, where bandit tasks are encountered sequentially and leveraged for learning across tasks [Lazaric et al., 2013, Basu et al., 2021, Kveton et al., 2021, Wan et al., 2021, Moradipari et al., 2022]. In contrast, our approach uses in-context learning, where a single algorithm adapts to a variety of new task it could, it encounters (see Figure 4). Also, unlike much of the meta-bandit theory literature—which focuses on simple models, e.g., linear [Cella et al., 2020, Cella and Pontil, 2021, Moradipari et al., 2022] or TS with parametric Bayesian priors, including mixture models [Wan et al., 2021, Kveton et al., 2021, Hong et al., 2022]—our method accommodates complex sequence models $p_\theta$ with low loss and any policy class with finite VC dimension. A notable exception is Boutilier et al. [2020], which directly optimizes a non-contextual bandit policy from historical data via gradient descent, but their approach only works for learning differentiable, soft-max based soft-max based algorithms.

# 6 Experiments

**Problem setting.** Throughout, $T = 500$, $|\mathcal{A}| = 10$,[1] outcomes $Y$ are binary, $R(y) = y$, and $Z$ has separate components $Z^{(a)} \in \mathbb{R}^2$ for each action. Our SYNTHETIC setting uses a Bayesian logistic regression data-generating process with contexts $X \in \mathbb{R}^5$. Our SEMI-SYNTHETIC setting mimics a cold-start, news recommendation setting using the MIcrosoft News Dataset [Wu et al., 2020]; $Z^{(a)}$ consists of article headline text, contexts $X \in \mathbb{R}^5$ are user features, and $Y \in \{0, 1\}$ represents whether user click on a recommendation. See Appendix B.1 for details.

**Bandit algorithms.** We use Generative TS (TS-Gen) as described in Section 4. For $p_\theta$, we use a simple recurrent neural network which takes in prior information $Z$, history $\mathcal{H}_{t-1}$, and current context $X$, and outputs a distribution over $Y$. In the SEMI-SYNTHETIC setting, $p_\theta$ embeds the article headline $Z$ using DistilBERT [Sanh et al., 2019]. We use a logistic regression-based policy class $\Pi$ for the SYNTHETIC setting and a multi-layer perceptron (MLP) policy class for the SEMI-SYNTHETIC setting. For baselines, three algorithms use the same $p_\theta$ model as TS-Gen, but select actions differently: 1) GREEDY deterministically selects the action predicted by $p_\theta$ to have the greatest next reward. 2) EPSILON-GREEDY employs GREEDY with probability 0.9 and otherwise selects an action uniformly at random. 3) TS-NEURAL-LINEAR, which uses the output of the last layer of the $p_\theta$ model as the context for a linear TS algorithm with a multivariate Gaussian prior; we consider variants with an uninformative prior and a prior fit using historical data. We also compare to a standard linear TS [Agrawal and Goyal, 2013], where $X_t$ is used as the context, as well as LinUCB [Li et al., 2010].

**Results.** As seen in Figure 6, TS-Gen outperforms other algorithms in both the SYNTHETIC and SEMI-SYNTHETIC settings. TS-Gen's superior performance compared to other algorithms that use the same $p_\theta$ model (GREEDY, EPSILON-GREEDY, TS-NEURAL-LINEAR) validates the benefit of our generative approach to uncertainty quantification and decision-making. We conjecture TS-Gen's advantage compared to LinUCB and TS-Linear is attributable to our pretraining procedure and the

---

[1]Recommendation options are often from a pre-filtered set [Davidson et al., 2010, Covington et al., 2016].

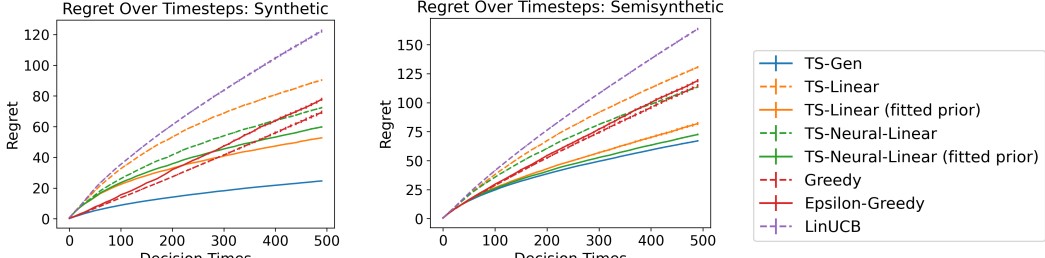

Figure 6: Cumulative regret averaged over 500 bandit tasks. Regret is against the best fitting policy in $\Pi$ (logistic for synthetic and MLP-based for semisynthetic). TS-Gen outperforms methods that use the same $p_\theta$ model (Greedy, $\epsilon$-Greedy, TS-Neural-Linear). Error bars (barely visible) denote $\pm 1$ s.e.

better use of prior information $Z$. We also found, as suggested by Theorem 2, the lower the offline prediction loss of $p_\theta$, the lower the regret of TS-Gen; see Appendix B.4.1.

**Computational costs.** For our semi-synthetic experiments, the generation and policy fitting times per decision were $4.2$ and $2.2$ seconds, respectively, on CPU (Appendix B.8). Various approaches could be investigated to speed up the algorithm. **Distillation:** Policy distillation, transferring knowledge from one policy to another, is commonly used to speed up computation. These approaches could distill TS-Gen into a policy that maps the current context $X_t$ and recent task history $\mathcal{H}_t$ to a distribution over actions [Czarnecki et al., 2019]. **Generation:** Generation could be sped up by truncating or reducing the number of outcomes generated per timestep. For sequence models more broadly, there is great interest in speeding up inference time through architecture changes [Tay et al., 2022] and optimizing around hardware constraints [Aminabadi et al., 2022, Dao et al., 2022]. **Policy fitting:** Policy fitting could be done incrementally instead of being refitted from scratch at each decision time.

## 7 Discussion

We introduce a generative TS algorithm for contextual bandits that is compatible with any generative model with low offline prediction loss and a policy fitting procedure with low VC dimension. We prove a regret bound for our algorithm that allows for misspecification of the generative model, and provides insights into information theoretic analyses for contextual bandits that may be of independent interest. Open directions include i) developing methods to guide how to choose an appropriate policy class $\Pi$ [Foster et al., 2020], ii) quantifying how much offline data is needed to train a high quality generative model (including settings where offline data is collected by a behavior policy), iii) exploring if the generative approach to modeling uncertainty can be extended to more difficult decision-making settings, like Markov decision processes, and iv) investigating methods to reduce computational cost.

**Limitations.** We evaluate our generative TS algorithm in only two experimental settings. As a result, our experiments are primarily a proof-of-concept for the viability of the generative TS approach. Additionally, in practice, our approach requires training a generative model $p_\theta$ to approximate complete task datasets, but in practice one may not have access to complete task datasets. We describe heuristic approaches we use to approximate complete task datasets from partial task datasets in Section 4.1. Further work is needed to assess practical feasibility in more complex settings and to formalize how well our heuristic approaches perform theoretically. Finally, our generative TS algorithm may also be computationally costly, especially when implemented with complex generative models. We discuss potential approaches to improve computation cost at the end of Section 6.

## 8 Acknowledgments

This work was partially supported by the AI Agents Initiative at the Columbia Business School. Tiffany Cai was partially supported by the National Science Foundation Graduate Research Fellowship under Grant No. DGE-2036197.

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

# A Theory

## A.1 Notation

- Throughout, we use $\mathbb{E}_t$ to denote expectations conditional on $\mathcal{H}_t$, i.e., we use

$$\mathbb{E}_t\left[\,\cdot\,\right] = \mathbb{E}\left[\,\cdot\,\mid \mathcal{H}_t\right]. \tag{7}$$

- We use $H(Y)$ to denote the entropy of a discrete random variable $Y$, i.e., $H(Y) = \sum_y \mathbb{P}(Y = y)\log\mathbb{P}(Y = y)dy$. We also use $H_t(Y) = H(Y \mid \mathcal{H}_t)$ to denote the entropy of $Y$ conditional on $\mathcal{H}_t$; Note that is standard in information theory, $H_t(Y)$ *is not* a random variable, rather, it marginalizes over $\mathcal{H}_t$:

$$H_t(Y) := H(Y \mid \mathcal{H}_t) = \mathbb{E}\left[\sum_y \mathbb{P}(Y = y \mid \mathcal{H}_t)\log\mathbb{P}(Y = y \mid \mathcal{H}_t)dy\right];$$

Above, the outer expectation marginalizes over the history $\mathcal{H}_t$.

- We also use $I(Z;Y)$ to denote the mutual information between some random variables $Z$ and $Y$, i.e., $I(Z;Y) = \int_z \int_y \mathbb{P}(Z = z, Y = y)\log\frac{\mathbb{P}(Z=z,Y=y)}{\mathbb{P}(Z=z)\mathbb{P}(Y=y)}dzdy$. We further use $I_t(Z;Y)$ to denote the mutual information between $Z$ and $Y$ conditional on $\mathcal{H}_t$ (which we then marginalize over $\mathcal{H}_t$), i.e.,

$$I_t(Z;Y) := I(Z;Y \mid \mathcal{H}_t)$$
$$= \mathbb{E}\left[\int_z \int_y \mathbb{P}(Z = z, Y = y \mid \mathcal{H}_t)\log\frac{\mathbb{P}(Z = z, Y = y \mid \mathcal{H}_t)}{\mathbb{P}(Z = z \mid \mathcal{H}_t)\mathbb{P}(Y = y \mid \mathcal{H}_t)}dxdy\right]; \tag{8}$$

Above, the outer expectation marginalizes over the history $\mathcal{H}_t$.

- Finally, we use $\mathrm{D_{KL}}\left(p(Z \mid X) \,\|\, p'(Z \mid X)\right)$ to denote the KL divergence, i.e.,

$$\mathrm{D_{KL}}\left(p(Z \mid X) \,\|\, p'(Z \mid X)\right) = \mathbb{E}\left[-\int p(Z \mid X)\log\frac{p(Z \mid X)}{p'(Z \mid X)}\right].$$

Above, the outer expectation marginalizes over $X$.

## A.2 Showing Algorithm 1 implements Thompson Sampling (Probability Matching)

**Proposition 1** (Algorithm 1 Implements Thompson Sampling). *Algorithm 1 with imputation model $p^*$ implements Thompson Sampling (probability matching), i.e., the following holds almost surely:*

$$\mathbb{P}(A_t = a \mid \mathcal{H}_t) = \mathbb{P}(\pi^*(X_t; \tau) = a \mid \mathcal{H}_t).$$

*Proof.* Recall that Algorithm 1 selects actions as follows:

$$\mathbb{P}(A_t = a \mid \mathcal{H}_t) = \mathbb{P}(\pi^*(X_t; \hat{\tau}) = a \mid \mathcal{H}_t).$$

Since $\hat{\tau} \sim p^*(\tau \in \cdot \mid \mathcal{H}_t)$ and from Eq (1) $\tau \sim p^*$, the distributions of $\tau$ and $\hat{\tau}$ are equal given $\mathcal{H}_t$. Hence, with probability 1 for any $j$:

$$\mathbb{P}(\hat{\tau} = j \mid \mathcal{H}_t) = \mathbb{P}(\tau = j \mid \mathcal{H}_t).$$

The above implies that

$$\mathbb{P}(\pi^*(X_t; \hat{\tau}) = a \mid \mathcal{H}_t) = \mathbb{P}(\pi^*(X_t; \tau) = a \mid \mathcal{H}_t).$$

Combining the above statements gives the result. $\qquad\square$

## A.3 Bounding the conditional entropy by VC dimension

**Proposition 2** (Complexity bound on entropy). *For policy class $\Pi$ over action space $\mathcal{A}_\tau$ with Nataranjan dimension $d$ (equivalent to VC dimension when $|\mathcal{A}_\tau| = 2$),*

$$H(\boldsymbol{\pi}^*(X_{1:T}) \mid Z_\tau) \leq H(\boldsymbol{\pi}^*(X_{1:T})) = O(d \cdot \log(T \cdot |\mathcal{A}_\tau|)).$$

*Proof.* The first inequality $H\big(\boldsymbol{\pi}^*(X_{1:T}) \mid Z_\tau\big) \leq H\big(\boldsymbol{\pi}^*(X_{1:T})\big)$ holds by the chain rule for entropy.

Note that $\boldsymbol{\pi}^*(X_{1:T})$ is a random vector of dimension $T$ where each dimension can take $|\mathcal{A}_\tau|$ different values. By a generalization of the Sauer-Shelah lemma [Sauer, 1972, Shelah, 1972], specifically Theorem 2 and Corollary 3 in [Haussler and Long, 1995], if a multi-class function that can take on $|\mathcal{A}_\tau|$ different values has Natarajan dimension $d$, then that function class can produce at most $\sum_{i=0}^{d} \binom{T}{i}(|\mathcal{A}_\tau| - 1)^i = O(T^d |\mathcal{A}_\tau|^d)$ different labelings of any $T$ points. Thus, since a coarse upper bound on the entropy of a random variable is the log of the number of unique values that variable can take, we get that $H\big(\boldsymbol{\pi}^*(X_{1:T})\big) \leq \log \sum_{i=0}^{d} \binom{T}{i}(|\mathcal{A}_\tau| - 1)^i = O\big(d \cdot \log(T \cdot |\mathcal{A}_\tau|)\big).$ $\qquad\square$

### A.4 Regret bound for Generative TS with an approximate imputation model

#### A.4.1 Lemma 1: To minimize loss $p_\theta$ needs to approximate $p^*$.

The next lemma is a standard result connecting the excess expected loss of a sequence model $p_\theta$ to its KL divergence from the true sequence model $p^*$. The expected loss of a sequence model $p_\theta$ is denoted $\ell(p_\theta)$; See (4). To minimize loss, $p_\theta$, the learner needs to closely approximate the true sequence model $p^*$.

**Lemma 1** (Decomposing loss under $p_\theta$). *For the loss $\ell$ as defined in* (4),

$$\ell(p_\theta) = \ell(p^*) + \mathrm{D}_{\mathrm{KL}}\left(p^*\big(X_{1:T}, \{Y_{1:T}^{(a)}\}_{a\in\mathcal{A}_\tau} \mid Z_\tau\big) \,\|\, p_\theta\big(X_{1:T}, \{Y_{1:T}^{(a)}\}_{a\in\mathcal{A}_\tau} \mid Z_\tau\big)\right).$$

*Proof.* By the definition of the expected loss in (4),

$$\ell(p_\theta) - \ell(p^*) = -\mathbb{E}\left[\log p_\theta\big(X_{1:T}, \{Y_{1:t-1}^{(a)}\}_{a\in\mathcal{A}_\tau} \mid Z_\tau\big)\right] + \mathbb{E}\left[\log p^*\big(X_{1:T}, \{Y_{1:t-1}^{(a)}\}_{a\in\mathcal{A}_\tau} \mid Z_\tau\big)\right]$$

$$= \mathrm{D}_{\mathrm{KL}}\left(p^*\big(Y_{1:T}^{(a)}, X_{1:T} \mid Z_\tau\big) \,\|\, p_\theta\big(Y_{1:T}^{(a)}, X_{1:T} \mid Z_\tau\big)\right)$$

Above, the final equality holds by the definition of the KL divergence. $\qquad\square$

#### A.4.2 Lemma 2: Action selection under perfect vs. imperfect imputation models.

**Lemma 2** (KL Divergence in next action distribution). *For any $t$,*

$$\mathrm{D}_{\mathrm{KL}}\left(\mathbb{P}_t\left(\pi^*(X_t; \tau) = \cdot\right) \,\|\, \mathbb{P}_t\left(A_t = \cdot\right)\right) \leq \ell(p_\theta) - \ell(p^*).$$

*Proof.* Note the following:

$$\mathrm{D}_{\mathrm{KL}}\left(\mathbb{P}_t\left(\pi^*(X_t; \tau) = \cdot\right) \,\|\, \mathbb{P}_t\left(A_t = \cdot\right)\right)$$

$$\underbrace{\leq}_{(a)} \mathrm{D}_{\mathrm{KL}}\left(\mathbb{P}_{p^*}\big(X_{1:T}, \{Y_{1:T}^{(a)}\}_{a\in\mathcal{A}_\tau} \mid \mathcal{H}_t\big) \,\|\, \mathbb{P}_{p_\theta}\big(X_{1:T}, \{Y_{1:T}^{(a)}\}_{a\in\mathcal{A}_\tau} \mid \mathcal{H}_t\big)\right)$$

$$\underbrace{\leq}_{(b)} \mathrm{D}_{\mathrm{KL}}\left(\mathbb{P}_{p^*}\big(X_{1:T}, \{Y_{1:T}^{(a)}\}_{a\in\mathcal{A}_\tau} \mid Z_\tau\big) \,\|\, \mathbb{P}_{p_\theta}\big(X_{1:T}, \{Y_{1:T}^{(a)}\}_{a\in\mathcal{A}_\tau} \mid Z_\tau\big)\right) \underbrace{\leq}_{(c)} \ell(p_\theta) - \ell(p^*).$$

- Inequality (a) holds because $\pi^*(X_t; \tau)$ and $A_t$ are both are derived by applying the same function to the contexts $X_{1:T}$ and outcomes $\{Y_{1:T}^{(a)}\}_{a\in\mathcal{A}_\tau}$.

- Inequality (b) holds because by the chain rule for KL divergence,

$$\mathrm{D}_{\mathrm{KL}}\left(\mathbb{P}_{p^*}\big(X_{1:T}, \{Y_{1:T}^{(a)}\}_{a\in\mathcal{A}_\tau} \mid \mathcal{H}_t\big) \,\|\, \mathbb{P}_{p_\theta}\big(X_{1:T}, \{Y_{1:T}^{(a)}\}_{a\in\mathcal{A}_\tau} \mid \mathcal{H}_t\big)\right)$$

$$= \mathrm{D}_{\mathrm{KL}}\left(\mathbb{P}_{p^*}\big(X_{1:T}, \{Y_{1:T}^{(a)}\}_{a\in\mathcal{A}_\tau} \mid Z_\tau\big) \,\|\, \mathbb{P}_{p_\theta}\big(X_{1:T}, \{Y_{1:T}^{(a)}\}_{a\in\mathcal{A}_\tau} \mid Z_\tau\big)\right)$$

$$+ \mathrm{D}_{\mathrm{KL}}\left(\mathbb{P}_{p^*}\big(\mathcal{H}_t, X_{1:T} \mid Z_\tau\big) \,\|\, \mathbb{P}_{p_\theta}\big(\mathcal{H}_t, X_{1:T} \mid Z_\tau\big)\right),$$

and the KL divergence is non-negative.

- Inequality (c) holds by Lemma 1 (Decomposing loss under $p_\theta$). $\qquad\square$

### A.4.3 Lemma 3: Mutual information equivalency.

**Lemma 3** (Mutual information equivalency)**.**

$$I_t\big(\pi^*(X_t;\tau);(Y_t^{(A_t)},A_t)\big)$$
$$= \mathbb{E}\bigg[\sum_{a,\tilde{a}\in\mathcal{A}_\tau}\mathbb{P}_t\big(A_t=a\big)\mathbb{P}_t\big(\pi^*(X_t;\tau)=\tilde{a}\big)\cdot \mathrm{D}_{\mathrm{KL}}\Big(\mathbb{P}_t\big(Y_t^{(a)}\mid\pi^*(X_t;\tau)=\tilde{a}\big)\,\|\,\mathbb{P}_t\big(Y_t^{(a)}\big)\Big)\bigg]$$

*Proof.* Note that

$$I_t\big(\pi^*(X_t;\tau);(Y_t^{(A_t)},A_t)\big)\underbrace{=}_{(a)}I_t\big(\pi^*(X_t;\tau);Y_t^{(A_t)}\mid A_t\big)$$

$$\underbrace{=}_{(b)}\mathbb{E}\bigg[\sum_{a\in\mathcal{A}_\tau}\mathbb{P}_t(A_t=a)I_t\big(\pi^*(X_t;\tau);Y_t^{(a)}\mid A_t=a\big)\bigg]\underbrace{=}_{(c)}\mathbb{E}\bigg[\sum_{a\in\mathcal{A}_\tau}\mathbb{P}_t(A_t=a)I_t\big(\pi^*(X_t;\tau);Y_t^{(a)}\big)\bigg]$$

$$\underbrace{=}_{(d)}\mathbb{E}\bigg[\sum_{a\in\mathcal{A}_\tau}\mathbb{P}_t(A_t=a)\sum_{\tilde{a}\in\mathcal{A}_\tau}\mathbb{P}_t\big(\pi^*(X_t;\tau)=\tilde{a}\big)\mathrm{D}_{\mathrm{KL}}\Big(\mathbb{P}_t\big(Y_t^{(a)}\mid\pi^*(X_t;\tau)=\tilde{a}\big)\,\|\,\mathbb{P}_t\big(Y_t^{(a)}\big)\Big)\bigg].$$

Above, equality (a) holds since $\pi^*(X_t;\tau)$ and $A_t$ are independent conditional on $\mathcal{H}_t$. Equality (b) holds by the definition of conditional mutual information. Equality (c) holds because $Y_t^{(a)}$ and $\pi^*(X_t;\tau)$ are independent of $A_t$ conditional on $\mathcal{H}_t$. Equality (d) holds by the KL divergence form of mutual information. $\qquad\square$

### A.4.4 Lemma 4: Mutual information bound for policies.

**Lemma 4** (Mutual information bound for policies)**.**

$$\sum_{t=1}^{T}I_t\big(\pi^*(X_t;\tau);\big(Y_t^{(A_t)},A_t\big)\big)\le H(\boldsymbol{\pi}^*(X_{1:T})\mid Z_\tau)$$

*Proof.*

$$\sum_{t=1}^{T}I_t\big(\pi^*(X_t;\tau);\big(Y_t^{(A_t)},A_t\big)\big)\underbrace{\le}_{(i)}\sum_{t=1}^{T}I_t\big(\boldsymbol{\pi}^*(X_{1:T});\big(Y_t^{(A_t)},A_t\big)\big)$$

$$\underbrace{=}_{(ii)}I_1\big(\boldsymbol{\pi}^*(X_{1:T});\big(Y_t^{(A_t)},A_t\big)_{t=1}^{T}\big)$$

$$\underbrace{=}_{(iii)}H_1\big(\boldsymbol{\pi}^*(X_{1:T})\big)-H_1\big(\boldsymbol{\pi}^*(X_{1:T})\mid\big(Y_t^{(A_t)},A_t\big)_{t=1}^{T}\big)$$

$$\underbrace{\le}_{(iv)}H_1(\boldsymbol{\pi}^*(X_{1:T}))\underbrace{\le}_{(v)}H(\boldsymbol{\pi}^*(X_{1:T})\mid Z_\tau)$$

- For inequality (i), note that for any random variables $X_1,X_2,Y$ (where $X_1,X_2$ are discrete), by properties of mutual information and entropy,

$$I((X_1,X_2);Y)=H(X_1,X_2)-H(X_1,X_2\mid Y)$$
$$=H(X_1)-H(X_1\mid Y)+H(X_2\mid X_1)-H(X_2\mid Y,X_1)$$
$$=I(X_1;Y)+I(X_2;Y\mid X_1)$$

The above implies that $I((X_1,X_2);Y)\ge I(X_1;Y)$ since $I(X_2;Y\mid X_1)\ge 0$. Recall that $\boldsymbol{\pi}^*(X_{1:T}):=\{\pi^*(X_t;\tau)\}_{t=1}^{T}$. Thus, since $\pi^*(X_t;\tau)\in\boldsymbol{\pi}^*(X_{1:T})$ we have that

$$I_t\big(\boldsymbol{\pi}^*(X_{1:T});(Y_t^{(A_t)},A_t)\big)\ge I_t\big(\pi^*(X_t;\tau);(Y_t^{(A_t)},A_t)\big).$$

- Equality (ii) uses the chain rule for mutual information.

- Equality (iii) holds by the relationship between mutual information and entropy.

- Inequality (iv) holds since entropy is always nonnegative.

- Inquality (v) uses that $H_1\left(\boldsymbol{\pi}^*(X_{1:T})\right) = H(\boldsymbol{\pi}^*(X_{1:T}) \mid Z_\tau, X_1) \leq H(\boldsymbol{\pi}^*(X_{1:T}) \mid Z_\tau)$, where the first equality holds by the definition of $H_1$ and the final inequality holds by the chain rule for entropy.

$\square$

### A.4.5 Proof of Theorem 2

**Theorem 2** (Regret bound for Generative TS with an approximate imputation model). *For Algorithm 1 with imputation model $p_\theta$, $\mathbb{A}_{\text{TS-Gen}}(p_\theta)$,*

$$\Delta\big(\mathbb{A}_{\text{TS-Gen}}(p_\theta)\big) \leq \underbrace{\sqrt{\frac{|\mathcal{A}_\tau|}{2T} \cdot H(\boldsymbol{\pi}^*(X_{1:T}) \mid Z_\tau)}}_{\text{Regret bound for Thompson sampling}} + \underbrace{\sqrt{2\{\ell(p_\theta) - \ell(p^*)\}}}_{\text{Penalty for sub-optimal prediction}} .$$

*Proof.* Note that by the law of iterated expectations,

$$\Delta(\mathbb{A}_{\text{TS-Gen}}) = \mathbb{E}\left[\frac{1}{T}\sum_{t=1}^{T} R(Y_t^{(\pi^*(X_t;\tau))}) - R(Y_t^{(A_t)})\right] = \mathbb{E}\left[\frac{1}{T}\sum_{t=1}^{T}\mathbb{E}_t\left[R(Y_t^{(\pi^*(X_t;\tau))}) - R(Y_t^{(A_t)})\right]\right].$$

Consider the following for any $t \in [1:T]$:

$$\mathbb{E}_t\left[R(Y_t^{(\pi^*(X_t;\tau))}) - R(Y_t^{(A_t)})\right]$$

$$= \sum_{a \in \mathcal{A}_\tau} \mathbb{P}_t(\pi^*(X_t;\tau) = a) \cdot \mathbb{E}_t\big[R(Y_t^{(a)}) \mid \pi^*(X_t;\tau) = a\big] - \sum_{a \in \mathcal{A}_\tau} \mathbb{P}_t(A_t = a) \cdot \mathbb{E}_t\big[R(Y_t^{(a)}) \mid A_t = a\big]$$

$$\underset{(i)}{=} \sum_{a \in \mathcal{A}_\tau} \mathbb{P}_t(\pi^*(X_t;\tau) = a) \cdot \mathbb{E}_t\big[R(Y_t^{(a)}) \mid \pi^*(X_t;\tau) = a\big] - \sum_{a \in \mathcal{A}_\tau} \mathbb{P}_t(A_t = a) \cdot \mathbb{E}_t\big[R(Y_t^{(a)})\big]$$

$$= \sum_{a \in \mathcal{A}_\tau} \sqrt{\mathbb{P}_t(\pi^*(X_t;\tau) = a)\mathbb{P}_t(A_t = a)}\big(\mathbb{E}_t\big[R(Y_t^{(a)}) \mid \pi^*(X_t;\tau) = a\big] - \mathbb{E}_t\big[R(Y_t^{(a)})\big]\big)$$

$$+ \sum_{a \in \mathcal{A}_\tau} \big(\sqrt{\mathbb{P}_t(\pi^*(X_t;\tau) = a)} - \sqrt{\mathbb{P}_t(A_t = a)}\big)$$

$$\qquad\qquad \big(\sqrt{\mathbb{P}_t(\pi^*(X_t;\tau) = a)}\mathbb{E}_t\big[R(Y_t^{(a)}) \mid \pi^*(X_t;\tau) = a\big] + \sqrt{\mathbb{P}_t(A_t = a)}\mathbb{E}_t\big[R(Y_t^{(a)})\big]\big)$$

$$\underset{(ii)}{\leq} \sum_{a \in \mathcal{A}_\tau} \sqrt{\mathbb{P}_t(\pi^*(X_t;\tau) = a)\mathbb{P}_t(A_t = a)}\big(\mathbb{E}_t\big[R(Y_t^{(a)}) \mid \pi^*(X_t;\tau) = a\big] - \mathbb{E}_t\big[R(Y_t^{(a)})\big]\big)$$

$$+ \sum_{a \in \mathcal{A}_\tau} |\mathbb{P}_t(\pi^*(X_t;\tau) = a) - \mathbb{P}_t(A_t = a)|$$

$$\underset{(iii)}{\leq} \sqrt{\frac{|\mathcal{A}_\tau|}{2}\sum_{a \in \mathcal{A}_\tau}\mathbb{P}_t(A_t = a)\sum_{\tilde{a} \in \mathcal{A}_\tau}\mathbb{P}_t(\pi^*(X_t;\tau) = \tilde{a}) \cdot D_{\text{KL}}\left(\mathbb{P}_t\big(Y_t^{(a)} \mid \pi^*(X_t;\tau) = \tilde{a}\big) \,\|\, \mathbb{P}_t\big(Y_t^{(a)}\big)\right)}$$

$$+ \sum_{a \in \mathcal{A}_\tau} |\mathbb{P}_t(\pi^*(X_t;\tau) = a) - \mathbb{P}_t(A_t = a)|$$

Above, equality (i) holds since conditional on $\mathcal{H}_t$, the action $A_t$ and the outcome $Y_t^{(a)}$ are independent. Inequality (ii) uses that $R$ takes values in $[0, 1]$ in the second term. Inequality (iii) above holds because:

$$\sum_{a \in \mathcal{A}_\tau} \sqrt{\mathbb{P}_t(\pi^*(X_t; \tau) = a)\mathbb{P}_t(A_t = a)}\big(\mathbb{E}_t\big[R(Y_t^{(a)}) \mid \pi^*(X_t; \tau) = a\big] - \mathbb{E}_t\big[R(Y_t^{(a)})\big]\big)$$

$$\underset{(a)}{\leq} \sqrt{|\mathcal{A}_\tau| \sum_{a \in \mathcal{A}_\tau} \mathbb{P}_t(\pi^*(X_t; \tau) = a)\mathbb{P}_t(A_t = a) \left(\mathbb{E}_t\big[R(Y_t^{(a)}) \mid \pi^*(X_t; \tau) = a\big] - \mathbb{E}_t\big[R(Y_t^{(a)})\big]\right)^2}$$

$$\underset{(b)}{\leq} \sqrt{|\mathcal{A}_\tau| \sum_{a \in \mathcal{A}_\tau} \mathbb{P}_t(A_t = a) \sum_{\tilde{a} \in \mathcal{A}_\tau} \mathbb{P}_t(\pi^*(X_t; \tau) = \tilde{a}) \left(\mathbb{E}_t\big[R(Y_t^{(a)}) \mid \pi^*(X_t; \tau) = \tilde{a}\big] - \mathbb{E}_t\big[R(Y_t^{(a)})\big]\right)^2}$$

$$\underset{(c)}{\leq} \sqrt{\frac{|\mathcal{A}_\tau|}{2} \sum_{a \in \mathcal{A}_\tau} \mathbb{P}_t(A_t = a) \sum_{\tilde{a} \in \mathcal{A}_\tau} \mathbb{P}_t(\pi^*(X_t; \tau) = \tilde{a}) \cdot \mathrm{D_{KL}}\left(\mathbb{P}_t\big(Y_t^{(a)} \mid \pi^*(X_t; \tau) = \tilde{a}\big) \,\|\, \mathbb{P}_t\big(Y_t^{(a)}\big)\right)}$$

Inequality (a) uses Cauchy-Schwartz inequality. Inequality (b) uses an elementary equality of summation. Inequality (c) uses Fact 9 of Russo and Van Roy [2016] (which uses Pinsker's inequality).

Using the above result, averaging over $t$ and taking an expectation, we get

$$\Delta(\mathbb{A}_{\text{TS-Gen}}) = \mathbb{E}\left[\frac{1}{T}\sum_{t=1}^T \mathbb{E}_t\left[R(Y_t^{(\pi^*(X_t; \tau))}) - R(Y_t^{(A_t)})\right]\right]$$

$$\leq \mathbb{E}\left[\frac{1}{T}\sum_{t=1}^T \sqrt{\frac{|\mathcal{A}_\tau|}{2}\sum_{a \in \mathcal{A}_\tau}\mathbb{P}_t(A_t = a)\sum_{\tilde{a} \in \mathcal{A}_\tau}\mathbb{P}_t(\pi^*(X_t; \tau) = \tilde{a}) \cdot \mathrm{D_{KL}}\left(\mathbb{P}_t\big(Y_t^{(a)} \mid \pi^*(X_t; \tau) = \tilde{a}\big) \,\|\, \mathbb{P}_t\big(Y_t^{(a)}\big)\right)}\right]$$

$$+ \mathbb{E}\left[\frac{1}{T}\sum_{t=1}^T \sum_{a \in \mathcal{A}_\tau} |\mathbb{P}_t(\pi^*(X_t; \tau) = a) - \mathbb{P}_t(A_t = a)|\right]$$

$$\underset{(i)}{\leq} \sqrt{\mathbb{E}\left[\frac{1}{T}\sum_{t=1}^T \frac{|\mathcal{A}_\tau|}{2}\sum_{a \in \mathcal{A}_\tau}\mathbb{P}_t(A_t = a)\sum_{\tilde{a} \in \mathcal{A}_\tau}\mathbb{P}_t(\pi^*(X_t; \tau) = \tilde{a}) \cdot \mathrm{D_{KL}}\left(\mathbb{P}_t\big(Y_t^{(a)} \mid \pi^*(X_t; \tau) = \tilde{a}\big) \,\|\, \mathbb{P}_t\big(Y_t^{(a)}\big)\right)\right]}$$

$$+ \mathbb{E}\left[\frac{1}{T}\sum_{t=1}^T \sqrt{2 \cdot \mathrm{D_{KL}}\left(\mathbb{P}_t\big(\pi^*(X_t; \tau) = \cdot\big) \,\|\, \mathbb{P}_t\big(A_t = \cdot\big)\right)}\right]$$

$$\underset{(ii)}{=} \sqrt{\frac{|\mathcal{A}_\tau|}{2} \cdot \frac{1}{T}\sum_{t=1}^T I_t\big(\pi^*(X_t; \tau); (Y_t^{(A_t)}, A_t)\big)} + \mathbb{E}\left[\frac{1}{T}\sum_{t=1}^T \sqrt{2 \cdot \mathrm{D_{KL}}\left(\mathbb{P}_t\big(\pi^*(X_t; \tau) = \cdot\big) \,\|\, \mathbb{P}_t\big(A_t = \cdot\big)\right)}\right]$$

$$\underset{(iii)}{\leq} \sqrt{\frac{|\mathcal{A}_\tau|}{2} \cdot \frac{1}{T}\sum_{t=1}^T I_t\big(\pi^*(X_t; \tau); (Y_t^{(A_t)}, A_t)\big)} + \sqrt{\frac{1}{T}\sum_{t=1}^T 2 \cdot \mathrm{D_{KL}}\left(\mathbb{P}_t\big(\pi^*(X_t; \tau) = \cdot\big) \,\|\, \mathbb{P}_t\big(A_t = \cdot\big)\right)}$$

$$\underset{(iv)}{\leq} \sqrt{\frac{|\mathcal{A}_\tau| \cdot H(\boldsymbol{\pi}^*(X_{1:T}) \mid Z_\tau)}{2T}} + \sqrt{2\{\ell(p_\theta) - \ell(p^*)\}}$$

- Inequality (i) uses Jensen's inequality on the first term and Fact 9 of Russo and Van Roy [2016] (which uses Pinsker's inequality) on the second term.

- Equality (ii) uses Lemma 3 (Mutual information equivalency).

- Inequality (iii) uses Jensen's inequality.

- The first term in inequality (iv) uses Lemma 4 (Mutual information bound for policies) and the second term uses Lemma 2 (KL Divergence of next action distribution).

$\square$

## A.5  Regret bound for Generative TS with a perfectly calibrated imputation model $p^*$

**Theorem 1** (Regret bound for Generative TS with a perfectly calibrated imputation model $p^*$). *For Algorithm 1 with imputation model $p^*$, $\mathbb{A}_{\text{TS−Gen}}(p^*)$,*

$$\Delta(\mathbb{A}_{\text{TS−Gen}}(p^*)) \leq \sqrt{\frac{|\mathcal{A}_\tau|}{2T} \cdot H(\boldsymbol{\pi}^*(X_{1:T}) \mid Z_\tau)}.$$

*Moreover, $\Delta(\mathbb{A}_{\text{TS−Gen}}(p^*)) \leq \sqrt{\frac{\bar{\Gamma}}{T} \cdot H(\boldsymbol{\pi}^*(X_{1:T}) \mid Z_\tau)}$, where $\bar{\Gamma}$ bounds the information ratio [Russo and Van Roy, 2016], i.e., $\bar{\Gamma} \geq \max_t \Gamma_t$ a.s. for $\Gamma_t := \frac{\mathbb{E}[R(Y_t^{(\pi^*(X_t;\tau))}) - R(Y_t^{(A_t)})|\mathcal{H}_t]^2}{I(\pi^*(X_t;\tau);Y_t^{(A_t)},A_t|\mathcal{H}_t)}$.*

*Proof.* The first result that $\Delta(\mathbb{A}_{\text{TS−Gen}}(p^*)) \leq \sqrt{\frac{|\mathcal{A}_\tau|}{2T} \cdot H(\boldsymbol{\pi}^*(X_{1:T}) \mid Z_\tau)}$, holds as a direct corollary of Theorem 2 by setting $p_\theta = p^*$.

We now show the second result that $\Delta(\mathbb{A}_{\text{TS−Gen}}(p^*)) \leq \sqrt{\frac{\bar{\Gamma}}{T} \cdot H(\boldsymbol{\pi}^*(X_{1:T}) \mid Z_\tau)}$. It holds by a very similar argument as Proposition 1 of [Russo and Van Roy, 2016].

$$\Delta(\mathbb{A}_{\text{TS-Gen}}) = \mathbb{E}\left[\frac{1}{T}\sum_{t=1}^T R(Y_t^{(\pi^*(X_t;\tau))}) - R(Y_t^{(A_t)})\right] \underset{(i)}{=} \mathbb{E}\left[\frac{1}{T}\sum_{t=1}^T \mathbb{E}_t\left[R(Y_t^{(\pi^*(X_t;\tau))}) - R(Y_t^{(A_t)})\right]\right]$$

$$\underset{(ii)}{=} \mathbb{E}\left[\frac{1}{T}\sum_{t=1}^T \sqrt{\Gamma_t \cdot I_t(\pi^*(X_t;\tau);Y_t^{(A_t)},A_t)}\right] \leq \sqrt{\bar{\Gamma}} \cdot \mathbb{E}\left[\frac{1}{T}\sum_{t=1}^T \sqrt{I_t(\pi^*(X_t;\tau);Y_t^{(A_t)},A_t)}\right]$$

$$\underset{(iii)}{\leq} \sqrt{\bar{\Gamma} \cdot \mathbb{E}\left[\frac{1}{T}\sum_{t=1}^T I_t(\pi^*(X_t;\tau);Y_t^{(A_t)},A_t)\right]} \underset{(iv)}{\leq} \sqrt{\frac{\bar{\Gamma}}{T} \cdot H(\boldsymbol{\pi}^*(X_{1:T}) \mid Z_\tau)}$$

Equality (i) holds by the law of iterated expectations. Equality (ii) holds by the definition of $\Gamma_t$. Inequality (iii) holds by Cauchy-Shwartz. Inequality (iv) holds by Lemma 4 (Mutual information bound for policies). $\qquad\square$

## A.6  Comparison to existing regret bounds

**Lemma 5** (Bounding information ratio for linear, non-contextual bandits). *Suppose $\mathbb{E}[R(Y_t) \mid A_t = a] = \varphi(A_t)^\top \theta^*$ for some $\theta^* \in \mathbb{R}^d$. Let the policy class $\Pi$ be such that for any $\pi \in \Pi$, $\pi(a) = \varphi(A_t)^\top \theta$ for some $\theta \in \mathbb{R}^d$. Then, $\Gamma_t \leq \frac{d}{2}$ a.s.*

*Proof.* This result follows by Proposition 5 of Russo and Van Roy [2016]. $\qquad\square$

**Generative TS regret bound for linear and logistic reward settings.**  By our Theorem 1, we have that the per round average Bayesian regret is bounded by $\sqrt{\frac{|\mathcal{A}_\tau|}{2T}H(\boldsymbol{\pi}^*(X_{1:T}) \mid Z_\tau)}$. Note that by Theorem 29.7 in Shalev-Shwartz and Ben-David [2014] a linear multiclass predictor of the form $\arg\max_{a \in \mathcal{A}_\tau} \theta^\top \varphi(x,a)$ for $\theta \in \mathbb{R}^d$ has Nataranjan dimension less than or equal to $d$. Thus, by applying Proposition 2, we have that $H(\boldsymbol{\pi}^*(X_{1:T}) \mid Z_\tau) \leq d \cdot \log(T \cdot |\mathcal{A}_\tau|)$, so the per round average Bayesian regret is bounded by $\sqrt{\frac{d|\mathcal{A}_\tau|}{2T} \log(T \cdot |\mathcal{A}_\tau|)}$.

Alternatively we can use the second result Theorem 1 to conclude that the per round average Bayesian regret is bounded by $\sqrt{\frac{\bar{\Gamma}}{T} \cdot H(\boldsymbol{\pi}^*(X_{1:T}) \mid Z_\tau)}$. By Lemma 5 we can choose $\bar{\Gamma} = \frac{d}{2}$, so by applying the same Proposition 2 argument as above, we have that the per round average Bayesian regret is bounded by $\sqrt{\frac{d^2}{2T} \log(T \cdot |\mathcal{A}_\tau|)}$.

Thus, by combining the above two results, we have that

$$\Delta(\mathbb{A}_{\text{TS-Gen}}(p^*)) \leq \sqrt{\frac{d \min(d, |\mathcal{A}_\tau|)}{2T} \log(T \cdot |\mathcal{A}_\tau|)} \tag{9}$$

**Linear logistic bandits.** We compare to Theorem 4 of Neu et al. [2022]. We only provide a brief overview of their result here; Please see the paper for additional details. Additionally note that their result applies to adversarial contextual bandits, whereas our result only applies for stochastic contextual bandits.

In their problem setup, the rewards are generated using a logistic model where $\theta^* \in \mathbb{R}^d$:

$$R(Y_t) \mid X_t, A_t \sim \text{Bernoulli}\left(\text{expit}\left(\varphi(X_t, A_t)^\top \theta^*\right)\right)$$

They show that cumulative Bayesian regret of Thompson sampling (with a correctly specified Bayesian model) is bounded by $\sqrt{2|\mathcal{A}_\tau|Td\{\log(2SCT+1)+1\}}$, where $\|\theta^*\| \leq S$ and $C$ is related Lipschitz smoothness of the logistic function. This means that the per round average Bayesian regret is bounded by $\sqrt{\frac{2d|\mathcal{A}_\tau|}{T}\{\log(2SCT+1)+1\}}$. Our result from (9) matches up to log factors.

**Linear non-contextual bandits.** We now compare to the result in Section 6.5 of Russo and Van Roy [2018]. Again, we only provide a brief overview of their result here; Please see the paper for additional details.

In their non-contextual problem setup, the rewards are generated using a linear model where $\theta^* \in \mathbb{R}^d$:

$$\mathbb{E}\left[R(Y_t) \mid A_t = a\right] = \varphi(A_t)^\top \theta^*.$$

They show that cumulative Bayesian regret of Thompson sampling (with a correctly specified Bayesian model) is bounded by $\sqrt{\frac{1}{2}\log(|\mathcal{A}_\tau|)dT}$. This means that the per round average Bayesian regret is bounded by $\sqrt{\frac{d}{2T}\log(|\mathcal{A}_\tau|)}$. Our result from (9) differs by a factor $\sqrt{\min(d, |\mathcal{A}_\tau|)}$ and a $\log(T)$ term.

The additional $|\mathcal{A}_\tau|$ and $\log(T)$ factors, we believe, are not artifacts of our specific algorithm, but rather are a consequence of the generality of our analysis.

- The $\log(T)$ term comes from our use of the Natarajan dimension, a generalization of the VC dimension. This term is common in bandit regret bounds that rely on VC dimension-based analysis, as seen in other work (e.g., Beygelzimer et al. [2011]). It appears to be an unavoidable consequence of this type of generalized bound.

- The $|\mathcal{A}|$ term is a consequence of the generality of our analysis, which does not utilize a shared parameterization across actions. The Russo and Van Roy [2018] bound for linear bandits is tighter because it leverages the linear structure, where $\mathbb{E}[R(Y_t) \mid A_t = a] = \varphi(a)^\top \beta$. In this setting, the parameter $\beta$ is common to all actions, meaning information gained from observing an action can be used to inform beliefs about the rewards of all other actions. Our analysis, however, does not assume or utilize such a shared structure. Instead, our regret-bound scales with the number of actions, similar to bounds for multi-armed bandits where the reward distribution for each arm is learned independently. This makes our bound applicable to a broader class of problems, but also looser for specific settings like linear bandits with shared parameters across actions.

While our result does not provide the tightest possible regret bound for a specific parametric model, we present a general and robust theoretical framework that characterizes the performance of Thompson Sampling variants that use modern generative sequence models and general policy classes.

## B    Experiment details

### B.1    Data generating environment

### B.1.1    Synthetic bandit setting.

We form samples of tasks $\tau = \{Z, (X_t, Y_t^{(a)}\}_{a \in \mathcal{A}_\tau})_{t=1}^T\}$ as follows. The task features $Z$ for a given bandit task consist of one feature per action, i.e. $Z = \{Z^{(a)}\}_{a \in \mathcal{A}_\tau}$, where only $Z^{(a)} \in \mathbb{R}^2$. We sample task features $Z^{(a)} \sim N(0_2, I_2)$ independently across all $|\mathcal{A}_\tau| = 10$ actions and contexts $X_t \sim N(0_5, I_5)$ independently across time. We let $R(y) = y$ and use the following generative model

for $Y_t^{(a)}$:

$$Y_t^{(a)} \mid W_t^{(a)} \sim \text{Bernoulli}(\sigma(W_t^{(a)})), \tag{10}$$

where

$$W_t^{(a)} = U_{\text{const}}^{(a)} + U_Z^{(a)} Z^{(a)} + U_X^{(a)} X_t + X_{t,1:2}^\top U_{\text{cross}}^{(a)} Z^{(a)},$$

for $\sigma(w) := (1 + \exp(-w))^{-1}$. Above we use $X_{t,1:2}$ to denote the first two dimensions of $X_t$. The latent variables are multivariate Gaussian: $U_{\text{const}}^{(a)} \sim N(0,1)$, $U_Z^{(a)} \sim N(1_2, I_2 \cdot 0.25^2)$, $U_X^{(a)} \sim N(1_5, I_5 \cdot 0.25^2)$, and $U_{\text{cross}}^{(a)}$ is a diagonal matrix where the diagonal entries are drawn independently from $N(1, 0.25^2)$.

### B.1.2  Semi-synthetic setting.

We form samples of tasks $\tau = \{Z, (X_t, Y_t^{(a)}\}_{a \in \mathcal{A}_\tau})_{t=1}^T\}$ as follows. We consider a semi-synthetic news recommendation setting in which we use text headlines $Z^{(a)}$ for action $a$. We let $R(y) = y$ and use the following generative model for $Y_t^{(a)}$:

$$Y_t^{(a)} \mid W_t^{(a)} \sim \text{Bernoulli}(\sigma(W_t^{(a)})), \tag{11}$$

where

$$W_t^{(a)} = U_{\text{const}}^{(a)} + U_Z^{(a)} \phi_Z(Z^{(a)}) + U_X^{(a)} \phi_X(X_t) + \phi_X(X_t)_{1:2}^\top U_{\text{cross}}^{(a)} \phi_Z(Z^{(a)}).$$

Above, $\phi_X(X_t) \in \mathbb{R}^4$ and $\phi_Z(Z^{(a)}) \in \mathbb{R}^2$ are complex nonlinear function of $X_t, Z^{(a)}$, which increases the difficulty of the learning task; We describe these functions in detail below. Note, $\phi_X(X_t)_{1:2}$ denotes the first two dimensions of $\phi_X(X_t) \in \mathbb{R}^2$. The latent variables are multivariate Gaussian: $U_{\text{const}}^{(a)} \sim N(0,1)$, $U_Z^{(a)} \sim N(1_2, I_2 \cdot 0.25^2)$, $U_X^{(a)} \sim N(1_4, 0.25^2 \cdot I_4)$, and the matrix $U_{\text{cross}}^{(a)}$ is diagonal with diagonal entries drawn independently from $N(1, 0.25^2)$.

**Contexts and $\phi_X$.**  The contexts $X_t \sim N(0_5, I_5)$ independently over time. We use

$$\phi_X(X_t) = X_{t,1:4} \cdot \text{sign}(X_{t,5}),$$

i.e., $\phi_X$ multiplies the first four dimensions of $X_t$ by the sign of the fifth dimension. Above, $X_{t,1:4}$ denotes the first 4 dimensions of $X_t$.

**Tasks features and $\phi_Z$.**  To form a task, we sample $|\mathcal{A}_\tau| = 10$ headlines $Z^{(a)}$ uniformly from the MIND large dataset [Wu et al., 2020]. $\phi_Z(Z^{(a)}) \in \mathbb{R}^2$ where the each dimension is the output of a pre-trained binary classifier evaluated on the news article. The first dimension is the output of probability output of a pre-trained sentiment classifier [Savani, 2022] and the second dimension is the probability output of a pre-trained formality classifier [Babakov et al., 2023]; The outputs are normalized to have mean 0 and variance 1 based on their distribution in the training set. Both classifier models were obtained from huggingface.com.

## B.2  Offline pretraining

### B.2.1  Sequence model architecture

**Synthetic setting.**  This architecture is described by Figure 7 except the $X$ MLP head and Distil-BERT head should be replaced by identity mappings. In the synthetic setting $p_\theta$ is simple recurrent neural network where the MLP takes as input $Z^{(a)}$, current context $X_t$, as well as summary statistics of the history $\mathcal{H}_t$ (discussed below). Before being fed into the MLP head, the summary statistics are then repeated 100 times and concatenated into a single vector. The $Z^{(a)}$, the current context $X_t$, and the repeated summary statistics of the history are fed into the final MLP head, which has 3 hidden layers, each with width 100. Note that the MLP consists of a linear layer taking the input to the first hidden layer, the 3 hidden linear layers, and finally a linear layer taking the output from the last hidden layer to the output before the sigmoid, which is a total of 5 linear layers. The output of the MLP head is fed through a sigmoid function to obtain a prediction for the probability that the next outcome is 1 (rather than 0).

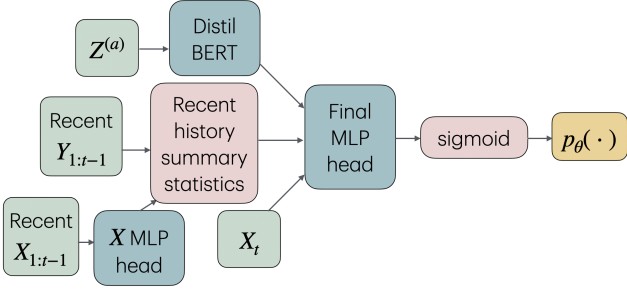

Figure 7: Diagram of model architecture for $p_\theta$, for semisynthetic settings. In synthetic settings, the model architecture is the same, except that it does not include the DistilBERT [Sanh et al., 2019] encoder to process text, or the MLP encoder to process contexts $X_t$.

The summary statistic of $\mathcal{H}_t$ only contains information about action $a$, i.e., $\{(X_s, Y_s) : s < t, A_s = a\}$. For these summary statistics, we aggregate the context vectors $X_s$ into a matrix $\mathbf{X}$, where each row is one element in $\{X_s : s < t, A_s = a\}$. We do the same for $\{Y_s : s < t, A_s = a\}$ to construct vector $\mathbf{Y}$. The $X_s$ and $Y_s$ appear in $\mathbf{X}$ and $\mathbf{Y}$ in order according to timestep $s$. The summary statistics are $(\mathbf{X}^\top \mathbf{X} + I)^{-1}$ and $\mathbf{X}^\top \mathbf{Y}$.

**Semisynthetic setting.** This architecture is described by Figure 7. In the semisynthetic setting, $p_\theta$ is implemented to take as input action-specific task feature $Z^{(a)}$, current context $X_t$, as well as summary statistics of the history $\mathcal{H}_t$ (discussed below). As displayed in Figure 7, the model architecture is as follows. We concatenate a DistilBert [Sanh et al., 2019] embedding of headline $Z^{(a)}$ with $X_t$, and a summary statistics of the history (desribed below) that is repated 100 times. Then, this concatenated vector is fed into the final MLP head (3 hidden layers, width 100). Finally, the output of the MLP is fed through a sigmoid function to obtain a prediction for the probability that the next outcome is 1 (rather than 0).

The summary statistic of $\mathcal{H}_t$ only contains information about action $a$, i.e., $\{(X_s, Y_s) : s < t, A_s = a\}$. For these summary statistics, we aggregate a learnable MLP embedding $\hat{\phi}_X$ (of depth 2 and width 100, labeled "X MLP Head" in Figure 7) of the context vectors $\hat{\phi}_X(X_s)$ into a matrix $\hat{\phi}_X(\mathbf{X})$, where each row is one element in $\{\hat{\phi}_X(X_s) : s < t, A_s = a\}$. We do the same for $\{Y_s : s < t, A_s = a\}$ to construct vector $\mathbf{Y}$. The $\hat{\phi}_X(X_s)$ and $Y_s$ appear in $\hat{\phi}_X(\mathbf{X})$ and $\mathbf{Y}$ in order according to timestep $s$.

### B.2.2 Forming approximate complete task datasets from partial datasets

As described in Section 4, $\mathcal{D}^{\text{offline}}$ ideally consists of bandit tasks $\tau \sim p^*$ as described in (1). In practice, one may not have "complete" task datasets $\tau = \{Z_\tau, X_{1:T}, \{Y_1^{(a)}, \ldots, Y_T^{(a)}\}_{a \in \mathcal{A}_\tau}\}$, but instead have some partial datasets, e.g., $\{Z_\tau, (X_1, A_1, Y_1), \ldots, (X_T, A_T, Y_T)\}$, collected by a behavior policy. In our experiments we use a several heuristics to construct approximate complete tasks $\tilde{\tau}$ from the the partial datasets. We use these approximate task datasets to form $\mathcal{D}^{\text{offline}} = \{\tilde{\tau}_1, \tilde{\tau}_2, \tilde{\tau}_3, \ldots, \}$.

The bootstrapping procedure we use makes several modeling simplifying assumptions, which are all common in the bandit literature:

- **Stationarity over time.** We model the $X_t$'s as being drawn i.i.d. from an unknown distribution. Additonally, we model the $(X_t, Y_t^{(a)})$ as exchangeable over time, i.e., $(X_t, Y_t^{(a)})_{t \in [1:\,T]} \overset{D}{=} (X_{\sigma(t)}, Y_{\sigma(t)}^{(a)})_{t \in [1:\,T]}$.

- **Independence across actions.** For a given task $\tau$, we model the outcomes $Y_{1:T}^{(a)}$ as i.i.d. conditional on $X_{1:T}$ and $Z$. This means that the outcomes $Y_{1:T}^{(a)}$ are not correlated with those from other actions, given contexts and task features.

Due the independence across actions assumption, instead of generating $\tilde{\tau}$, we instead impute rows $\tilde{\tau}^{(a)} = \{X_{1:T}, Y_{1:T}^{(a)}\}$ for individual actions $a$. We use a bootstrapping procedure to construct $\tilde{\tau}^{(a)}$, described in Algorithm 4 below.

---

**Algorithm 4** Bootstrapping historical data to form $\tilde{\tau}^{(a)}$

---

**Require:** Historical data from action $a$, denoted $S^{(a)} \leftarrow \{(X_t, Y_t) : A_t = a\}$
1: Sample (with replacement) $T$ tuples from $S^{(a)}$:

$$(\tilde{X}_1, \tilde{Y}_1^{(a)}), \dots (\tilde{X}_T, \tilde{Y}_T^{(a)}) \mid S^{(a)} \overset{i.i.d.}{\sim} \frac{1}{|S^{(a)}|} \sum_{(x,y) \in S^{(a)}} \delta_{(x,y)}$$

2: **return** $\tilde{\tau}^{(a)} = \left\{ (\tilde{X}_1, \tilde{Y}_1^{(a)}), \dots (\tilde{X}_T, \tilde{Y}_T^{(a)}) \right\}$

---

### B.2.3    Additional sequence model training details

**Synthetic setting.**    For offline training of $p_\theta$, we sample 20k independent "task action" datasets $\{Z^{(a)}, X_{1:N^{(a)}}, Y_{1:N^{(a)}}^{(a)}\}$ according to the data generating process from Appendix B.1.1; Specifically we use $N^{(a)} = 1000$ for all $a$. This dataset is split into training and validation sets where 10k actions are in each set. The training set is used for training $p_\theta$ via gradient descent for 100 epochs, with loss from display (6); Note for approximating the distribution of $X_t$, we use the empirical distribution of 1000 contexts $X$'s from the training set (no gradient descent training). In each training batch, we use bootstrap resampling, specifically, Algorithm 4. The validation set is for choosing best hyperparameters and training epoch. We optimize weights in $p_\theta$ with the AdamW optimizer. We try learning rates $\{0.1, 0.01, 0.001\}$ and choose the learning rate with the lowest validation loss, which is 0.01. We set weight decay to 0.01. The batch size is 500 actions $a$ per batch.

**Semi-synthetic setting.**    For offline training of $p_\theta$, we sample independent "task action" datasets $\{Z^{(a)}, X_{1:N^{(a)}}, Y_{1:N^{(a)}}^{(a)}\}$. For $Z^{(a)}$'s use 104k headlines from the MIND dataset [Wu et al., 2020]; 20k are used for the training set, 10k are used for validation, and 74k are used for bandit evaluation. The outcomes $X$ and $Y$ are generated according to the process described in Appendix B.1.2; Specifically we use $N^{(a)} = 1000$ for all $a$. The training set is used for training $p_\theta$ via gradient descent for 40 epochs, with loss from display (6); Note for approximating the distribution of $X_t$, we use the empirical distribution of 1000 contexts $X$'s from the training set (no gradient descent training). In each training batch, we use bootstrap resampling, specifically, Algorithm 4. We optimize weights in $p_\theta$ with the AdamW optimizer. We try learning rates $\{0.1, 0.01, 0.001\}$ and choose the learning rate and also the training epoch with the lowest validation loss; the learning rate chosen is 0.01. We set weight decay to 0.01. The batch size is 500. We do not fine-tune the DistilBERT encoder, i.e., its weights are frozen.

## B.3    Online learning

Bandit datasets are constructed as described in Appendix B.1. In the semisynthetic setting, the headlines used are as described in Appendix B.2.3.

### B.3.1    TS-Gen policy-fitting details

Here we describe additional details used to fit $\pi^*(\,\cdot\,; \hat{\tau}_t) \in \Pi$ given an imputed task dataset $\hat{\tau}$. Using $\hat{\tau}_t$, for each action $a \in \mathcal{A}_\tau$, we fit an action-specific model to predict (binary) outcome $Y$ given context $X$; We use $f^{(a)}(X; \hat{\tau}_t)$ to denote this fitted action-specific model. Note that these models do not incorporate task features $Z^{(a)}$. Then,

$$\pi^*(x; \hat{\tau}_t) = \operatorname{argmax}_{a \in \mathcal{A}_\tau} f^{(a)}(x; \hat{\tau}_t).$$

In our experiments we choose $f$ to be either a logistic regression function or an MLP.

- For logistic $f$, we use the default logistic regression implementation from `scikit-learn` [Pedregosa et al., 2011].

- For MLP-based policies, we use the default MLP classifier implementation (including hyperparameters), also from `scikit-learn` [Pedregosa et al., 2011]. This is an MLP with one hidden layer of width 100, with ReLU activation, trained with Adam optimizer, with initial learning rate 0.001, and batch size 200. There is no early stopping or additional validation split.

### B.3.2 Baseline bandit methods

The first three (Greedy, Epsilon-Greedy, and Softmax) are alternative ways to make decisions using an existing pre-trained sequence model $p_\theta$. The others (Linear Thompson Sampling, LinUCB) are contextual bandit methods that do not use $p_\theta$.

**Greedy.** We use the samed trained sequence model $p_\theta$ as used by TS-Gen. In the online step, at time $t$, we feed the history $\mathcal{H}_t$ (which includes the current context $X_t$) into the model $p_\theta$. We look at the predicted mean reward $\mathbb{E}\left[R(Y_t) \mid \mathcal{H}_t, A_t = a\right]$ for each action $a$ according to $p_\theta$ and select the action with the largest predicted mean reward.

**Epsilon-Greedy** This algorithm also uses $p_\theta$ and at each decision time follows the Greedy policy with probability $1 - \epsilon$ and selects an action uniformly at random from $\mathcal{A}_t$ with probability $\epsilon$. We use $\epsilon = 0.1$.

**Softmax sampling.** Softmax sampling also uses the sequence model $p_\theta$ to select actions. Just like the Greedy algorithm, at time $t$, we feed the history $\mathcal{H}_t$ (which includes the current context $X_t$) into the model $p_\theta$. We look at the predicted mean reward $\mathbb{E}\left[R(Y_t) \mid \mathcal{H}_t, A_t = a\right]$ for each action $a$ according to $p_\theta$ and put these values through a softmax function with temperature $b > 0$. We then sample the action $A_t$ according to the softmax probabilities. Note that softmax sampling is also called Boltzmann sampling and is also called PreDeToR-$\tau$ in Mukherjee et al. [2024]. Following Mukherjee et al. [2024], we set $b = 0.05$.

For lack of space, this is omitted in the main text but we compare PreDeToR-$\tau$ with Greedy and Epsilon-Greedy later in this Appendix.

**Linear Thompson Sampling (Isotropic Gaussian prior).** We use Linear TS [Agrawal and Goyal, 2013] with the following Bayesian model with a non-informative prior. For each arm $a \in \mathcal{A}_\tau$ and time $t$, outcomes are modeled as a linear function of $X_t$,

$$Y_t^{(a)} = X_t^\top \beta^{(a)} + \epsilon_t^{(a)} \qquad \text{where} \quad \beta^{(a)} \sim N(\mu, \Sigma) \ \text{ and } \ \epsilon_t^{(a)} \sim N(0, \sigma^2)$$

where $\epsilon_t^{(a)}$ is modeled as Gaussian with mean 0 and variance 1/4 (since the maximum variance of a Bernoulli is 1/4). Note that unlike TS-Gen, linear Thompson sampling does not learn a rich and flexible prior based on task features $Z_\tau$.

**Linear Thompson Sampling (Fitted prior).** We use Linear TS [Agrawal and Goyal, 2013] with the following Bayesian model with a prior fit using historical data $\mathcal{D}^{\text{offline}}$. We use $\mathcal{A}^{\text{offline}}$ to denote all actions across all tasks in $\mathcal{D}^{\text{offline}}$. We fit the following Bayesian linear regression model for each action $a \in \mathcal{A}^{\text{offline}}$:

$$Y_t^{(a)} = X_t^\top \beta^{(a)} + \epsilon_t^{(a)} \qquad \text{where} \quad \beta^{(a)} \sim N(\mu, \Sigma) \ \text{ and } \ \epsilon_t^{(a)} \sim N(0, \sigma^2)$$

where $\beta^{(a)}$ are drawn iid across $a$, and $\epsilon_t^{(a)}$ are drawn iid across $a, t$, so that

$$Y_t^{(a)} \mid X_t \sim N(\mu^\top X_t, \sigma^2 + X_t^\top \Sigma X_t).$$

For fitting $\mu, \Sigma, \sigma^2$, we do the following:

- For each action $a \in \mathcal{A}^{\text{offline}}$ in the available historical data (see Appendix B.2.3), we fit the action-specific least squares model:

$$\hat{\beta}^{(a)} = \text{argmin}_\beta \ \sum_{t \in \mathcal{T}_1} (Y_t^{(a)} - X_t \beta)^2,$$

where $\mathcal{T}_1$ denotes the first 80% of timesteps in $[1, 2, \ldots, T]$.

- Then we set $\hat{\mu} = \frac{1}{|\mathcal{A}^{\text{offline}}|} \sum_{a \in \mathcal{A}^{\text{offline}}} \hat{\beta}^{(a)}$ and $\hat{\Sigma}$ to be the sample covariance of the $\hat{\beta}^{(a)}$ across $a \in \mathcal{A}^{\text{offline}}$. We set $\hat{\sigma}^2$ to the sample variance of the residuals, i.e. the sample variance of $Y_t^{(a)} - X_t \hat{\beta}^{(a)}$ across $a$ and $t$, where $a \in \mathcal{A}^{\text{offline}}$ and $t \in \mathcal{T}_2$, and where $\mathcal{T}_2$ denotes the final 20% of timesteps in $[1, 2, \ldots, T]$.

**Linear Thompson Sampling Using Learned Features (Isotropic Gaussian prior)** Here, we propose a variant of Linear Thompson Sampling above, but using features extracted from the learned sequence model $p_\theta$. Let $\phi_\theta(Z^{(a)}, X_t)$ denote the last-layer feature embedding (using the output of the last hidden layer) in the MLP head in the sequence model $p_\theta$ used for TS-Gen (see Section B.2.1) evaluated for the current context $X_t$ and action feature $Z^{(a)}$; note we do not feed any history into the sequence model $p_\theta$ when forming $\phi_\theta(Z^{(a)}, X_t)$.

We use the following Bayesian linear regression model, which is linear in $\phi_\theta(Z^{(a)}, X_t)$:

$$Y_t^{(a)} = \phi_\theta(Z^{(a)}, X_t)^\top \beta^{(a)} + \epsilon_t^{(a)} \quad \text{where} \quad \beta^{(a)} \sim N(0, I_d) \quad \text{and} \quad \epsilon_t^{(a)} \sim N(0, 1/4).$$

where $\beta^{(a)}$ are drawn iid across $a$, and $\epsilon_t^{(a)}$ are drawn iid across $a, t$. Above, the noise variance is set to $1/4$, the maximum variance of a Bernoulli random variable. Note that while this version of linear Thompson sampling does use $p_\theta$ to form the context $\phi_\theta(Z^{(a)}, X_t)$, it does not utilize a fitted prior.

**Linear Thompson Sampling Using Learned Features (Fitted prior)** Here, we propose a variant of the Linear Thompson Sampling Using Learned Features method above, but fit the prior using historical data $\mathcal{D}^{\text{offline}}$. We use $\mathcal{A}^{\text{offline}}$ to denote all actions across all tasks in $\mathcal{D}^{\text{offline}}$. We fit the following Bayesian linear regression model for each action $a \in \mathcal{A}^{\text{offline}}$:

$$Y_t^{(a)} = \phi_\theta(Z^{(a)}, X_t)^\top \beta^{(a)} + \epsilon_t^{(a)} \quad \text{where} \quad \beta^{(a)} \sim N(\mu, \Sigma) \quad \text{and} \quad \epsilon_t^{(a)} \sim N(0, \sigma^2)$$

where $\beta^{(a)}$ are drawn iid across $a$, and $\epsilon_t^{(a)}$ are drawn iid across $a, t$.

For fitting $\mu, \Sigma, \sigma^2$, we do the following:

- For each action $a \in \mathcal{A}^{\text{offline}}$ in the available historical data (see Appendix B.2.3), we fit the action-specific least squares ridge-regression model using the corresponding historical data from $\mathcal{D}^{\text{offline}}$:

$$\hat{\beta}^{(a)} = \text{argmin}_\beta \left\{ \alpha \|\beta\|_2^2 + \sum_{t \in \mathcal{T}_1} (Y_t^{(a)} - \phi_\theta(Z^{(a)}, X_t)^\top \beta)^2 \right\},$$

where $\mathcal{T}_1$ denotes the first 80% of timesteps in $[1, 2, \ldots, T]$. We set the ridge parameter $\alpha = 0.1$. We add the ridge penalty term because $\phi_\theta(Z^{(a)}, X_t)$ is 100-dimensional and we found that the adding the ridge penalty leads to more stable coefficient estimates.

- Then we set $\hat{\mu} = \frac{1}{|\mathcal{A}^{\text{offline}}|} \sum_{a \in \mathcal{A}^{\text{offline}}} \hat{\beta}^{(a)}$ and $\hat{\Sigma}$ to be the sample covariance of the $\hat{\beta}^{(a)}$ across $a \in \mathcal{A}^{\text{offline}}$; to ensure the covariance matrix is well-conditioned (to avoid numerical issues when computing posteriors), we add $10^{-4} \cdot I_d$ to the sample covariance. We set $\hat{\sigma}^2$ to the sample variance of the residuals, i.e. the sample variance of $Y_t^{(a)} - \phi_\theta(Z^{(a)}, X_t)^\top \hat{\beta}^{(a)}$ across $a$ and $t$, where $a \in \mathcal{A}^{\text{offline}}$ and $t \in \mathcal{T}_2$, and where $\mathcal{T}_2$ denotes the final 20% of timesteps in $[1, 2, \ldots, T]$.

**LinUCB.** We implement LinUCB-disjoint in [Li et al., 2010], on contexts $X_t$. We set $\alpha = 0.1$ as it performs well in comparison to a small set of other values tried ($\{0.1, 1, 2\}$). Note that unlike TS-Gen, LinUCB does not learn a rich and flexible prior based on task features $Z_\tau$.

## B.4 Additional simulation results

### B.4.1 Sequence loss vs. regret under TS-Gen (Figure 8)

We examine the relationship between sequence model loss $\ell(p_\theta)$ and regret of TS-Gen using $p_\theta$ in the SYNTHETIC setting. Our Theorem 2 suggests that the lower the loss of a sequence model $p_\theta$ the lower the regret of TS-Gen using that sequence model $p_\theta$. We examine this by varying the amount of training tasks used to learn $p_\theta$ and thus obtain sequence models with different losses. Indeed, in Figure 8, models trained on more data tend to have lower sequence loss, which tend to have lower regret.

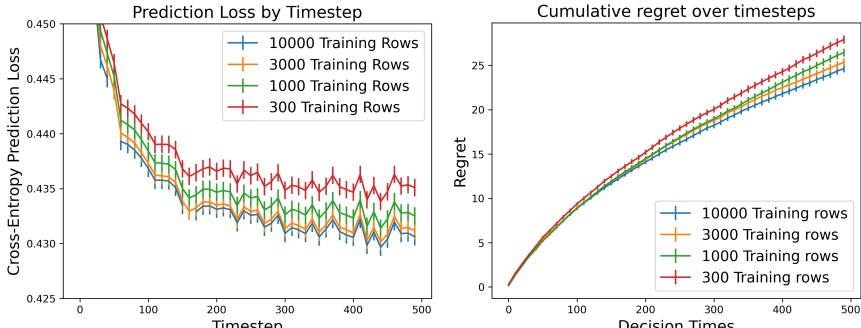

Figure 8: **Sequence loss vs. bandit regret:** We demonstrate the relationship between sequence loss and regret for TS-Gen by pre-training our sequence models offline on varying dataset sizes in the semisynthetic setting. As training dataset sizes are smaller, sequence loss (left) is higher (worse), and bandit regret (right) is higher (worse). "Training rows" refers to the number of actions used in the pool of actions to select from to form tasks (Appendix B.2.3). **(Left)**: Prediction loss by timestep. We plot an empirical estimate of the per-timestep (non-cumulative) loss from (4) by evaluating our sequence models on an held-out validation set. Error bars represent $\pm 1$ s.e. **(Right)**: Cumulative regret for TS-Gen using the corresponding sequence models, with logistic policy class, and relative to the logistic "oracle". Error bars represent $\pm 1$ s.e. averaged over 500 re-drawn bandit environments.

### B.4.2 Policy class for TS-Gen (Figure 9)

The choice of policy class $\Pi$ affects both the reward achieved by TS-Gen, and the "oracle"; see Figure 9. In the semisynthetic setting, TS-Gen has moderately greater reward using an MLP-based policy than a logistic policy. In contrast, the "oracle" using an MLP-based policy is much better than the "oracle" using a logistic policy.

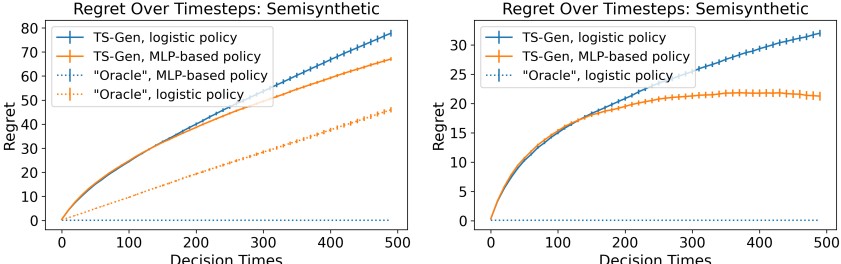

Figure 9: Varying policy classes in the semisynthetic setting. The same experimental results are plotted on the left and the right. The plot on the right calculates regret relative to the logistic "oracle", while the left calculates regret relative to the MLP-based "oracle". Error bars are $\pm 1$ s.e. across 500 bandit environments.

### B.5 TS-Gen with truncated imputation horizon

TS-Gen imputes missing outcomes up to the horizon $T$. In practice, one may want to truncate the number of imputed timesteps in Algorithm 3 to a smaller number than $T$ in order to reduce computation cost in the decision-making step for TS-Gen, or to run TS-Gen when the total number of timesteps $T$ is unknown. Fortunately, regret does not degrade quickly when the imputation horizon is truncated, which we observe in Figure 10.

### B.6 TS-Gen with simpler sequence models

To understand how the regret for TS-Gen depends on the complexity of the sequence model $p_\theta$, we compare TS-Gen with simpler sequence models. Specifically, we compare the regret results in Section 6 with their counterparts where the final MLP head of the sequence model (Figure 7) is

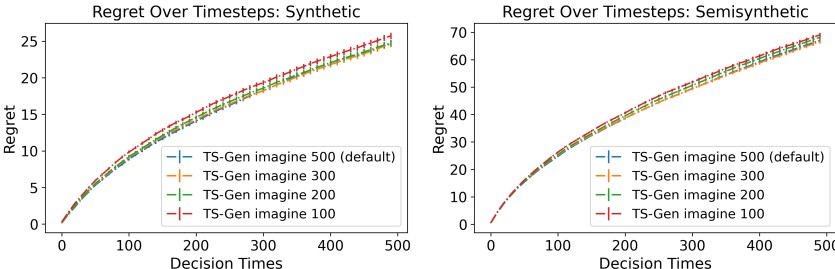

Figure 10: Regret for TS-Gen with truncated imputation horizon in the synthetic (left) and semisynthetic (right) settings. Performance degrades slowly and smoothly with reduced number of imputation steps. Error bars are ±1 s.e. across 500 Monte Carlo repetitions.

replaced with an MLP with fewer layers. Recall that the usual TS-Gen has an input layer, 3 hidden layers, and an output layer (Section B.2.1), adding up to 5 total layers. We compare regret for such variants of TS-Gen in Figure 11.

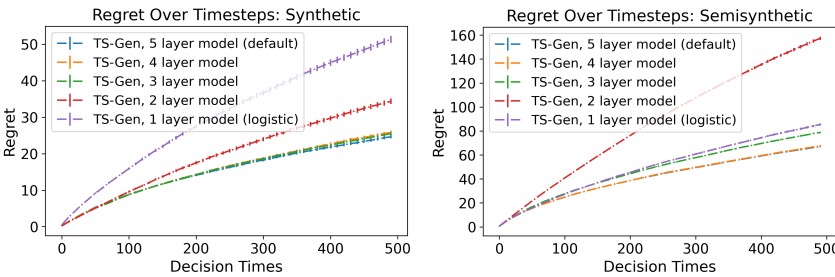

Figure 11: Regret for TS-Gen with simpler sequence models $p_\theta$ in the synthetic (left) and semisynthetic (right) settings. Error bars are ±1 s.e. across 500 bandit environments.

## B.7    Softmax Sampling vs. Greedy

Here we compare Softmax Sampling as described in Appendix B.3.2, with Greedy, and $\epsilon$-Greedy in Figure 12. Softmax Sampling is another bandit algorithm that uses $p_\theta$. Like $\epsilon$-Greedy, it "explores" while using $p_\theta$, but it does not adequately handle uncertainty as TS-Gen does, as evidenced by the difference in regret.

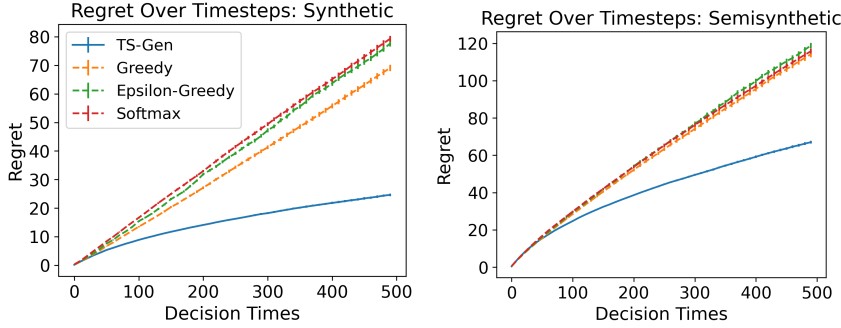

Figure 12: Regret for Softmax Sampling vs Greedy vs $\epsilon$-Greedy in the synthetic (left) and semisynthetic (right) settings. Error bars are ±1 s.e. across 500 bandit environments.

## B.8   Compute resources

**Offline pretraining.**   Pretraining a single $p_\theta$ for the semisynthetic setting took at most $\approx 12$ hours; We use a CPU cluster at Columbia GSB and request at most 50GB of memory per job. The semisynthetic data generating process also involves evaluating two pre-trained text classifiers, and then caching their outputs and/or embeddings (DistilBERT embeddings + text classifier outputs in the semisynthetic setting); this was done once on a single GPU at negligible time cost (several minutes).

**Online decision-making.**   For online decision-making, we also use a CPU cluster at Columbia GSB and for each job we request at most 10GB of memory. Below is a sample of decision-making time *per timestep*, across 20 sampled semisynthetic bandit tasks ($10,000$ decisions total). Note that we cache the DistilBERT embedding representing the news article text so this is not included in the computation. In each sampled bandit task, we compute the average per-timestep time in seconds for generation vs policy fitting; then, we report mean and variance of these quantities across the sampled bandit tasks. We write these times below as mean $\pm$ standard deviation across the $10,000$ decisions.

- **TS-Gen, using logistic policies:** $3.1 \pm 0.5$ seconds for generating $\hat{\tau}_t$, $0.01 \pm 0.02$ seconds for policy fitting, $3.1 \pm 0.5$ seconds total
- **TS-Gen, using MLP-based policies:** $4.2 \pm 0.5$ seconds for generating $\hat{\tau}_t$, $2.2 \pm 0.03$ seconds for policy fitting, $6.4 \pm 0.5$ seconds total
- **Neural Linear Thompson Sampling:** $1.9 \pm 0.2$ seconds total

## B.9   Constrained policy classes

Algorithmic fairness is a topic of general interest [Mehrabi and Wager, 2024, Mitchell et al., 2021], and fairness can be thought of as a modeling constraint [Corbett-Davies et al., 2017]. Because our proposed method takes a policy class as an input, results can be immediately adapted to settings that require specific kinds of constraints, such as fairness or balancing constraints.

As a simple example, we could enforce the constraint that at any given timestep $t$, a fitted policy $\pi^*(\cdot; \hat{\tau}_t)$ must satisfy the condition that it would give a specific treatment to approximately the same proportion of user contexts $X_t$ across two pre-specified groups. For example, these groups can be two sets of specific individuals, representatively drawn from the population, where each group selects individuals from a different geographic region, and where the groups are not related to contexts drawn in $\hat{\tau}_t$. This kind of fairness constraint is essentially the notion of predictive parity [Verma and Rubin, 2018].

To implement such a policy class, we would modify the policy fitting procedure in Line 4 in Algorithm 1 as follows: Letting $\hat{\tau}_t$ be the imputed table, and letting $G_1 = (X_{1,1}, X_{1,2}, \ldots, X_{1,N_1})$ and $G_2 = (X_{2,1}, X_{2,2}, \ldots, X_{2,N_2})$ be these predefined sets of user contexts $X$, we would be solving $\pi^* = \operatorname{argmax}_\pi \sum_{t=1}^{T} Y_t^{\pi(X_t; \hat{\tau}_t)}$ subject to the constraint that $\left| \frac{1}{N_1} \sum_{i=1}^{N_1} \mathbf{1}\{\pi(X_{1,i}) = a\} - \frac{1}{N_2} \sum_{i=1}^{N_2} \mathbf{1}\{\pi(X_{2,i}) = a\} \right| \leq \epsilon$ for some chosen $\epsilon > 0$.

## B.10   Licenses

**MIND news dataset**   We use the MIND news dataset [Wu et al., 2020]. It is under a Microsoft Research License at https://github.com/msnews/MIND/blob/master/MSR%20License_Data.pdf, which we comply with. The terms of use are at https://www.microsoft.com/en-us/legal/terms-of-use.

**DistilBERT**   Our semisynthetic sequence models use DistilBERT [Sanh et al., 2019] from https://huggingface.co/distilbert/distilbert-base-uncased. It has an apache-2.0 license, with license and terms of use at https://choosealicense.com/licenses/apache-2.0/.

**Text classifiers for semisynthetic setting**   We use text classifiers for the data generating process in the semisynthetic experiment setting. We use a sentiment classifier [Savani, 2022], accessed at https://huggingface.co/bhadresh-savani/distilbert-base-uncased-sentiment-sst2, and a formality classifier [Babakov et al., 2023], accessed at https://huggingface.co/s-nlp/roberta-base-formality-ranker. Both models were obtained from huggingface.com. The sentiment classifier is not associated

with a paper and is under an Apache 2.0 license https://choosealicense.com/licenses/apache-2.0/, which we comply with. The formality classifier is associated with a paper, as cited, and is under a cc-by-nc-sa-4.0 license https://spdx.org/licenses/CC-BY-NC-SA-4.0, which we also comply with.

