# OpenReview forum: "Contextual Thompson Sampling via Generation of Missing Data"
_NeurIPS.cc/2025/Conference — NeurIPS 2025 poster_

### Official Review · Reviewer_BmbW · 2025-06-17

**Clarity:** 3
**Significance:** 3
**Originality:** 3
**Rating:** 4
**Confidence:** 3

**Summary:**

This paper proposes a novel framework for Thompson Sampling (TS) in stochastic contextual bandit settings, where a generative model is used to impute all missing data points (with future context vectors), rather than sampling from the posterior distribution (classical approach). Notably, the quality of the generative model affects the regret bound only as an additive term, rather than multiplicatively influencing the leading term. Leveraging this generative approach, the authors also explore the potential for incorporating meta-learning, which could offer practical benefits in real-world applications. Experimental results demonstrate strong empirical performance. However, a significant limitation is the computational cost of TS-Gen, which is approximately 1000 times higher than that of simpler baselines, posing a serious bottleneck for deployment in real-time or resource-constrained online settings.

**Questions:**

### Questions

1. Although Proposition 1 provides an upper bound on the entropy in terms of the Natarajan dimension $d$ and $\log ( T  |A_\tau|))$, to the best of my knowledge, the Natarajan dimension can be quite large when the function class is parameterized by neural networks.
For example, Jin (2023) shows that the Natarajan dimension of a neural network class with ReLU activations is upper bounded by  $O(|A_\tau| \cdot p^2)$, where $p$ denotes the number of parameters of neural networks. This suggests that the function class may need to be sufficiently restricted to ensure its Natarajan dimension remains below $O(\sqrt{T})$, in order for the resulting bound to be meaningful. Since my understanding might be wrong, I would appreciate the authors' opinion on this issue.

2. In the current formulation, the algorithm requires the time horizon $T$ as input, since the generative model is used to impute all future outcomes in advance. If $T$ is unknown, a natural alternative would be to generate a fixed number of future outcomes or to adopt a doubling strategy for the number of imputations. Could the authors clarify whether such modifications would affect the theoretical guarantees or empirical performance of the proposed method?

---
Jin, Ying. "Upper bounds on the natarajan dimensions of some function classes." 2023 IEEE International Symposium on Information Theory (ISIT). IEEE, 2023.

### Comments

1. two overlapped citations: see Lines 467 and 468.
2. In appendix A.3., Line 982: Lemma 2 -> Jensen's inequality?

**Ethical Concerns:**

["NO or VERY MINOR ethics concerns only"]

**Final Justification:**

I have also read the discussion with the other reviewers and identified several points that could help improve the paper’s quality. While I consider the paper borderline, some of the authors’ responses (to other reviewers) seem reasonable to me, so I think it is acceptable. Therefore, I will keep my score unchanged but lower my confidence.

**Limitations:**

yes

**Quality:**

3

**Strengths And Weaknesses:**

### Strength

* This paper is well-written and figures are very helpful to capture the overall idea.
* A general framework that can be used for single/multiple tasks, several imputation models including neural networks, and various settings such as non-stationary settings.
* The regret bound clearly separates the quality of generative model from the algorithmic performance and it is confirmed in experiments. The additive dependence on model quality is notable since it would scale as $\log T$ under mild regularity conditions as the authors mentioned.
* By relying on supervised learning rather than probabilistic modeling, TS-Gen broadens the applicability of TS to settings where approximate posteriors are difficult to compute or unavailable.

### Weakness

As acknowledged by the authors, TS-Gen requires significantly more computation than baseline methods (up to 1000× slower in some cases). This scalability issue is a serious bottleneck in online decision-making applications where response time is critical. One possible remedy could be to use an (adaptive or increasing) batch size strategy, where the generative model re-samples the missing outcomes only when a new batch threshold is reached, incorporating the currently observed outcomes and contexts.

---

> ### Author Rebuttal · Authors · 2025-07-30
>
> Thank you so much for your thoughtful comments and questions!
>
> ## (1) Natarajan dimension
> We agree that known bounds on the Natarajan dimension of general neural networks can indeed be large, as highlighted by Jin (2023). This critique seems to apply to nearly all learning theory with neural networks. We have several additional comments related to this:
>
> 1. **Policy class versus imputation model.** Our Proposition 1 upper bounds $H(\\mathbf{\\pi}^\\ast(X_{1:T}) \\mid Z\_\\tau)$ by
> $O(d \\cdot \\log(T \\cdot |\\mathcal{A}|\_\\tau))$, where $d$ is the Natarajan dimension of the policy class $\\Pi$. Recall that the policy class $\\Pi$ is the class of functions that take in $X$ and output an action in $\\mathcal{A}$. We first want to clarify that the *policy class* $\\Pi$ is distinct from the sequence model, $p^\\ast$ or $p_\\theta$, used for imputation. After we use the sequence model to impute missing outcomes, we fit a policy $\\pi \\in \\Pi$ that is used to select actions. In summary, the complexity of the sequence model does not impact the bound in Theorem 1 at all, only the complexity of the policy class $\\Pi$ does.
>
> 2. **Policy class flexibility.** While the Natarajan dimension can be large for arbitrary neural networks, our framework offers the flexibility to choose different policy classes $\Pi$ depending on how complex of a model is needed to effectively model $R(Y) \mid X, A$. If the policy class $\Pi$ is simpler, the Natarajan dimension would be significantly smaller. For example, in our synthetic simulation results, one version of our algorithm uses a logistic regression-based policy class.
>
> 3. **Scaling with $T$.** Each policy $\pi \in \Pi$ is a function that maps any context $X$ to an action in $\mathcal{A}$. Generally, *we do not expect the Natarajan dimension of the policy class $\Pi$ to depend on $T$.* For example, linear policy classes of the form $\text{argmax}_{a} \theta^\top \varphi(x, a)$ for $\theta \in \mathbb{R}^p$ have Natarajan dimension $p$ (see Appendix A.5 for additional details and the derivation of the Natarajan dimension for other special case policy classes). In summary, for reasonable policy classes the Natarajan dimension does not grow with $T$, making the overall regret bound meaningful. We will make this point more overt in the revision.
>
> ## (2) Unknown horizon $T$
> In this work, we consider a finite horizon setting in which $T$ is known and used by the algorithm. A simple way to modify our algorithm when $T$ is not known is to have the sequence model only impute past missing outcomes and a fixed number of timesteps, $m$, into the future. Although our theoretical results do not apply to this version of algorithm, we find empirically that the performance of the algorithm for sufficiently large $m$ is often like that of our original algorithm, which imputes outcomes over the entire horizon $T$ (see results table below); performance degrades smoothly as $m$ is decreased. Furthermore, this modified approach increases the computational efficiency of our method in the online decision-making phase. We will include these additional results and discussion in our revision.
>
> **Table 1: Cumulative regret of TS-Gen in synthetic setting, when imputing up to m=100, 200, 300, 500 timesteps steps into the future.** Note that when $m=500$, this matches our original TS-Gen algorithm and synthetic setting results in Figure 5 in the draft.
>
> |  | t=100 | t=200 | t=300 | t=400 | t=500 |
> |:--------------------------|--------|--------|--------|--------|--------|
> | TS-Gen, m=500=T (original) | 8.20   | 13.65  | 17.98  | 21.43  | 24.58  |
> | TS-Gen, m=300              | 8.43   | 13.96  | 18.47  | 21.90  | 25.14  |
> | TS-Gen, m=200              | 8.60   | 14.31  | 18.63  | 22.27  | 25.48  |
> | TS-Gen, m=100              | 9.21   | 14.97  | 19.44  | 23.04  | 26.28  |
>
>
> ## (3) Computational efficiency
> When implementing Thompson Sampling with neural network models, a variety of methods for approximate posterior inference methods have been proposed. Many of these existing methods (e.g., MCMC and variational inference [Riquelme et al]) are computationally costly. Our approach, as presented in Algorithm 1, is also computationally costly but we believe it can be extended to be more computationally efficient in practice:
> 1. **Truncated generation.** As discussed above, one can truncate the generation length to at most a fixed number of $m$ timesteps into the future. For sufficiently large $m$, we find that the decision-making performance is comparable to doing the full imputation (see Table 1 above).
> 2. **Batch updating.** Following your comment, a well-known practical strategy to increase the computational efficiency of bandit algorithms in practice is to update after taking a batch of action, rather than updating the algorithm after every action taken [Yi et al, Yu et al].
> 3. **Policy distillation.** In the draft we also mention that one could explore policy distillation methods [Czarnecki et al] to increase computational efficiency, as these methods are designed to transfer knowledge from one policy to another, typically to speed up computation.
>
> ## (4) Typos
> - Thank you for pointing out the overlapped citations on Lines 467 and 468. We will correct this!
> - We appreciate the careful reading of the appendix. You are correct that in Appendix A.3, Line 982, the inequality should indeed refer to Jensen's inequality, not Lemma 2. We will fix this in the revision.
>
> ## References
> - Czarnecki, Pascanu, Osindero, Jayakumar, Swirszcz, & Jaderberg. *Distilling policy distillation.* AISTATS 2019
> - Gelman, Carlin, Stern, Dunson, Vehtari, Rubin. *Bayesian data analysis.* Chapman and Hall/CRC 1995
> - Riquelme, Tucker, & Snoek. *Deep Bayesian Bandits Showdown: An Empirical Comparison of Bayesian Deep Networks for Thompson Sampling.* ICLR 2018
> - Yi, Wang, He, Chandrasekaran, Wu, Heldt, Hong, Chen, Chi. *Online Matching: A Real-time Bandit System for Large-scale Recommendations.* RecSys 2023
> - Yu & Oh. *Optimal and Practical Batched Linear Bandit Algorithm.* ICML 2025

---

> > ### Comment · Reviewer_BmbW · 2025-08-03
> >
> > Thank you for your responses and the effort in addressing the concerns I raised.

---

> > ### Comment · Reviewer_BmbW · 2025-08-04
> > **Additional question**
> >
> > As far as I understand, Theorem 1 provides a general result that yields an explicit regret bound once an imputation model $p^*$ is specified.
> > As the authors noted, when this model is chosen as a linear multiclass predictor, the Natarajan dimension becomes smaller than the dimension of the linear model, which (almost) recovers previously known results. However, as mentioned in Appendix A.5, the resulting regret bound is looser than the prior one.
> >
> > Q1. Do you think these additional terms can be reduced, are they unavoidable due to the generality of Theorem 1, or are they inherent to the specific algorithm being analyzed?
> >
> > Q2.  While Theorem 1 offers a general formulation, as I noted earlier, the Natarajan dimension of certain neural networks can be very large. This suggests that using more complex models could lead to looser regret bounds, even though the empirical performance may improve, an issue frequently observed in deep learning. In the bandit setting, how should we interpret this phenomenon? Roughly speaking, can we view it as a form of overfitting to limited data, which results in a larger worst-case regret?

---

> ### Author Response · Authors · 2025-08-05
>
> Thank you for your questions and engagement with our paper!
>
> > *As far as I understand, Theorem 1 provides a general result that yields an explicit regret bound once an imputation model $p^\*$ is specified. As the authors noted, when this model is chosen as a linear multiclass predictor, the Natarajan dimension becomes smaller than the dimension of the linear model, which (almost) recovers previously known results. However, as mentioned in Appendix A.5, the resulting regret bound is looser than the prior one.*
>
> Based on your comment, we think there may still be some confusion. The complexity of the sequence model $p^\*$ **does not** impact the  Theorem 1 bound and **we do not ever discuss the Natarajan dimension of $p^\*$ in our work**. We only mention the Natarajan complexity of the policy class $\Pi$, which is used in our Proposition 1 bound.
>
> To clarify the distinction between the sequence model and the policy class, and how the Natarajan dimension fits in, recall:
> - After we use the sequence model $p^\*$ to impute missing outcomes, we fit a policy $\pi \in \Pi$ that is used to select an action
> - Our Theorem 1 (when sequence model $p^\*$ is used) bounds the per-timestep regret with $\sqrt{\frac{|\mathcal{A}|}{2T} H(\mathbf{\pi}^\*(X_{1:T}) \mid Z_\tau)}$. Our Proposition 1 bounds the term $H(\mathbf{\pi}^\*(X_{1:T}) \mid Z_\tau)$ by $O(b \cdot \log(T \cdot|\mathcal{A}|))$ where $b$ is the Natarajan dimension of the **policy class** $\Pi$. Note, this is **not** the sequence model $p^\*$.
>
>
> > *Q1. Do you think these additional terms can be reduced, are they unavoidable due to the generality of Theorem 1, or are they inherent to the specific algorithm being analyzed?*
>
> Thanks for your question. For clarity, we'll use the linear (non-contextual) bandits example from Appendix A.5, where the expected reward is $\mathbb{E}[ R(Y) | A_t = a] = \varphi(a)^\top \beta$ for some $\beta \in \mathbb{R}^d$.
>
> Combining our Theorem 1, Proposition 1, and the Natarajan dimension of linear classifiers gives us a per-timestep regret bound of $\sqrt{\frac{d \cdot |\mathcal{A}|}{2T} \log(T \cdot|\mathcal{A}|)}$. The reviewer correctly points out this bound is looser than the known result from Russo and Van Roy: $\sqrt{\frac{d }{2T} \log(|\mathcal{A}|)}$.
>
> The additional terms in our bound—the $\log(T)$ and $|\mathcal{A}|$ factors—we believe are not artifacts of our specific algorithm, but rather are a consequence of the generality of our analysis.
> - The $\log(T)$ term comes from our use of the Natarajan dimension, a generalization of the VC dimension. This term is common in bandit regret bounds that rely on VC dimension-based analysis, as seen in other work (e.g., Beygelzimer et al). It appears to be an unavoidable consequence of this type of generalized bound.
> - The $|\mathcal{A}|$ term is a consequence of the generality of our analysis, which does not utilize a shared parameterization across actions. The Russo and Van Roy bound for linear bandits is tighter because it leverages the linear structure, where $\mathbb{E}[ R(Y) | A_t = a] = \varphi(a)^\top \beta$. In this setting, the parameter $\beta$ is common to all actions, meaning information gained from observing an action can be used to inform beliefs about the rewards of all other actions. Our analysis, however, does not assume or utilize such a shared structure. Instead, our regret bound scales with the number of actions, similar to bounds for multi-armed bandits where the reward distribution for each arm is learned independently. This makes our bound applicable to a broader class of problems, but also looser for specific settings like linear bandits with shared parameters across actions.
>
> Ultimately, the primary contribution of our work is not to provide the tightest possible regret bound for a specific parametric model. Instead, our goal is to present a general and robust theoretical framework that characterizes the performance of Thompson Sampling variants that use modern generative sequence models and general policy classes. The generality of our approach means our bounds hold even with misspecified Bayesian models, a scenario where standard Thompson Sampling bounds do not apply.
>
> ## References
> - Beygelzimer, Langford, Li, Reyzin, & Schapire. *Contextual Bandit Algorithms with Supervised Learning Guarantees.* AISTATS 2011
> - Russo and Van Roy. *An Information-Theoretic Analysis of Thompson Sampling.* JMLR 2016

---

> ### Author Response · Authors · 2025-08-05
>
> > *Q2. While Theorem 1 offers a general formulation, as I noted earlier, the Natarajan dimension of certain neural networks can be very large. This suggests that using more complex models could lead to looser regret bounds, even though the empirical performance may improve, an issue frequently observed in deep learning. In the bandit setting, how should we interpret this phenomenon? Roughly speaking, can we view it as a form of overfitting to limited data, which results in a larger worst-case regret?*
>
> Thank you for this question. It gets to the heart of a crucial challenge: the potential trade-off between improved empirical performance and looser theoretical guarantees. You're right that a more complex model can improve performance but may also lead to looser worst-case regret bounds. Our framework addresses this by considering how model complexity impacts two distinct parts of our algorithm.
>
> ### **(a) Complexity of the sequence model $p_\theta$.**
> The sequence model $p_\theta$ is pre-trained on offline data. A key advantage of our Theorem 2 is that its regret bound depends only on the offline prediction loss of $p_\theta$ compared to the optimal sequence model $p^\*$, i.e., $\ell(p_\theta)-\ell(p^\*)$. The bound **does not** depend on the intrinsic complexity of $p_\theta$ itself. This implies that using a more complex neural network is not an issue, and in fact is encouraged, as long as it helps to minimize this offline loss.
>
> ### **(b) Complexity on the policy class $\Pi$.**
> Your question about the Natarajan dimension of complex models is most relevant here, as it directly relates to the policy class $\Pi$. We assume the algorithm designer defines a policy class $\Pi$ and is able to fit a "best-in-class" oracle policy $\pi^* \in \Pi$ given a complete dataset. Our theory bounds the average per-timestep regret between this oracle policy and our algorithm's decisions.
>
> A more complex policy class $\Pi$ in the worst case, it can be much harder for our algorithm to learn $\pi^\*$. This difficulty is captured by the $H(\mathbf{\pi}^\*(X_{1:T}) \mid Z_\tau)$ term in our regret bounds (Theorems 1 and 2). Our Proposition 1 bounds this term by $O(b \cdot \log(T \cdot ∣\mathcal{A}∣))$, where $b$ is the Natarajan dimension of $\Pi$. You are correct that for a complex neural network policy class, a standard characterization of the Natarajan dimension $b$ can be very large and may not be the tightest possible characterization for a specific neural network architecture.
>
> Our goal was to provide a widely applicable theoretical framework for bandit algorithms that can easily take advantage of tighter characterizations of supervised learning models as they are developed in the future. Even though existing Natarajan dimension bounds are loose for certain models, our regret bound results will improve alongside any tighter bounds that researchers develop for supervised learning in the future.

---

> > ### Comment · Reviewer_BmbW · 2025-08-06
> >
> > I greatly appreciate the detailed explanation and for pointing out my misunderstanding!
> >
> > That point is now clear, and I apologize for the confusion. My remaining question concerns the interpretation of Theorem 2 (related to the previous response Q2-(b)), which may also stem from a misunderstanding on my part.
> >
> > Theorem 2 decomposes the regret into two terms:
> > (i) the regret of TS, and
> > (ii) the regret due to suboptimal prediction by the imputation model.
> >
> > The choice of imputation model affects only term (ii), where a good approximation of the generative model makes this term small. Since we do not know the exact data distribution, as the authors explained, using a complex model for $p_\theta$ is encouraged.
> >
> > Term (i) depends on the entropy of the best-fitted policy $\pi^*$ , which is determined by the policy class $\Pi$ and is independent of $p_\theta$ (it may depend indirectly through data generation, but likely not crucial).
> > As the authors noted, when the policy class is large, it becomes harder to learn the best policy, which increases term (i).
> >
> > My question is how to interpret this phenomenon. Intuitively, if we could obtain the best-fitted policy in a large policy class, we might expect the regret to be small. Thus, I would like to understand which effect is more critical here: does a large $\Pi$ increase regret directly through the entropy term $H$, or indirectly via Proposition 1 (due to looseness from Natarajan dimension)?

---

> > > ### Author Response · Authors · 2025-08-06
> > >
> > > Thank you for your question! We are glad hear that the previous response helped clarify our work significantly.
> > >
> > > Your interpretation of term (ii) of our Theorem 2 regret bound is on point and your interpretation of term (i) is *largely* accurate. We first want to clarify the following points that will help in answering your question:
> > > - **Recall, we prove a "best-in-class" regret bound.** Our regret bound is against the best fitting policy $\pi^\*(\cdot; \tau)$ fit on the complete dataset $\tau$ in a class of policies $\Pi$. Our bounds correctly reflect that bigger classes $\Pi$ represent more ambitious learning goals through the entropy term $H \big(\pi^\*(X_{1:T}) \mid Z_\tau \big)$.
> > > - **The entropy term $H \big(\pi^\*(X_{1:T}) \mid Z_\tau \big)$ depends on both the policy class $\Pi$ and the task distribution $\tau \sim p^\*$.** The entropy term $H \big(\pi^\*(X_{1:T}) \mid Z_\tau \big)$ captures how "variable" $\pi^\*(X_{1:T})$ is across tasks given task features $Z_\tau$. The term $\pi^\*(X\_{1:T}) = \\{ \pi^\*(X\_t; \tau) \\}\_{t=1}^T$ is the oracle policy evaluated at contexts $X_{1:T}$. $\pi^\*(X\_{1:T})$ is random due to the randomness in the "complete" dataset $\tau = \\\{  Z\_{\tau}, X\_{1:T}, \\{ Y\_{1:T}^{(a)} : a \in \mathcal{A} \\} \\} \sim p^\*$.
> > >
> > >     Intuitively, the entropy $H \big(\pi^\*(X_{1:T}) \mid Z_\tau \big)$ could be low if one more of the following hold:
> > >     - The policy class $\Pi$ is simple so no matter how variable the task datsets $\tau \sim p^\*$ are, the oracle policy $\pi^\*(\cdot \mid \tau)$ is relatively stable and $\pi^\*(X_{1:T})$ has low entropy.
> > >     - If the variability across tasks $\tau \sim p^*$ is low and as a result $\pi^\*(\cdot \mid \tau)$ is relatively stable and $\pi^\*(X_{1:T})$ has low entropy.
> > > - **Proposition 1 bounds the entropy term for worst case task distributions $\tau \sim p^\*$.** Proposition 1 bounds the entropy $H \big(\pi^\*(X_{1:T}) \mid Z_\tau \big)$ in a way that *only* depends on the policy class $\Pi$ and *does not depend* on the task distribution $\tau \sim p^\*$. This bound is conservative because it holds for the *worst case* task distribution $\tau \sim p^\*$.
> > >
> > > **To directly address your question**, the increase in regret for more complex policy classes is captured by the entropy term, which measures the variability of the best-fitting policy across tasks. Proposition 1 provides a worst-case upper bound on this entropy term (worst case over task distributions $\tau \sim p^*$) by relating it to the Natarajan dimension of the policy class. In short, the entropy term directly captures the increased difficulty of learning a best-in-class policy for a larger class, and Proposition 1 provides a worst-case theoretical guarantee on this effect.

---

> > > > ### Comment · Reviewer_BmbW · 2025-08-07
> > > >
> > > > Thank you for the detailed explanation! The points are now clear to me.
> > > > I will finalize my score after reviewing the discussion with the other reviewers and the AC

---

### Official Review · Reviewer_y7h5 · 2025-07-02

**Clarity:** 3
**Significance:** 2
**Originality:** 3
**Rating:** 3
**Confidence:** 4

**Summary:**

The paper describes a new bandit algorithm that addresses the cold start problem by meta learning a generative model with offline data regarding a multitude of tasks. The pre-trained generative model is used to sample complete task data based on which a policy is fitted.  The paper claims that this approach is equivalent to contextual Thompson Sampling.  Some regret bounds are provided.  Experiments show that the proposed generative thompson sampling technique has lower regret than various baselines in a proof of concept that involve one synthetic problem or one semi-synthetic problem.

**Questions:**

* Can you prove that the proposed algorithm is equivalent to Thompson sampling?
* What are the advantages of proposed algorithm over Thompson sampling?
* In the new recommendation example, why does Z correspond to a news article?  It seems more natural to think of each action as a news article.  If the actions are not news articles, then what are the actions?

**Ethical Concerns:**

["NO or VERY MINOR ethics concerns only"]

**Final Justification:**

I read several times the author rebuttal.  I really appreciate all the effort put into responding to my questions.  Nevertheless, my concerns remain.  First the paper claims that the proposed algorithm is equivalent to Thompson sampling, but a formal proof is lacking.  The authors explained a good direction for constructing a proof.  I appreciate the definition of Thompson sampling as probability matching.  The authors provide informal arguments to suggest that their algorithm provides a form of probability matching.  But a proper proof should include a line by line derivation.  In particular, Proposition 2 lacks a proof that the statement holds with probability 1. Also, the argument that it is okay to replace $\tau$ by $\hat{\tau}$ because they are sampled from the same distribution lacks a mathematical derivation.

I do not agree with the claim that TS-Gen generalizes Thompson sampling.  The fact that the optimal policy is replaced by a best fitting policy simply means that an approximation is introduced.  All Thompson sampling algorithms introduce approximations because estimating the posterior is intractable.  If we follow the logic that an approximation leads to a generalization then all previous Thompson sampling techniques should also be considered as different forms of generalized Thompson sampling, but this is misleading.

My concern about the fairness of the empirical evaluation remains. The paper proposes TS-Gen which does two things: it provides a new way of estimating the posterior and it uses a transformer based seq2seq model.  The paper claims that their new posterior estimation is the key for low regret, but the transformer model could also be a major factor.  This is why I asked the authors to provide a fair comparison with other techniques that use the same transformer model instead of a linear model.  In their last response below, the authors generated new results that use the same transformer model with a neural linear estimation technique. Technically, this is not doing probability matching because the posterior is only computed with respect to the last layer (instead of all parameters), but we can view this as an approximation and therefore I am fine with this.  The main issue is that this baseline is only fed with a single $(X_t, Z)$ while their approach is also fed with the history.  Again, this is unfair.

Overall, the work is promising, but it is not ready for publication.

**Limitations:**

The paper acknowledges that the empirical evaluation is limited to a synthetic problem and a semi-synthetic problem.  It also acknowledges that the proposed algorithm will be computationally intensive in complex tasks.

**Paper Formatting Concerns:**

No concern

**Quality:**

2

**Strengths And Weaknesses:**

Strengths:
* The proposed approach is novel and interesting.
* Regret bounds are provided, though I did not verify them.
* The paper is well written and generally clear.

Weaknesses:
* Claim that the proposed algorithm is equivalent to Thompson Sampling is not substantiated.  Please provide a proof.
* It is not clear what are the advantages of the proposed algorithm over regular Thompson Sampling.  Let $p_\theta(Y|X,A)$ be a distribution parameterized by $\theta$ over the reward $Y$ when executing $A$ in context $X$.  Regular Thompson Sampling would learn a posterior distribution over $\theta$, then sample $\theta$ based on which a reward $Y$ is sampled for each action $A$ and the action with the highest reward is selected.  Since Bayesian learning is rarely tractable, estimating a posterior over $\theta$ will need to be approximated.  In contrast, the proposed approach learns $p_\theta(\tau)$ by maximum likelihood.  If the proposed approach is indeed equivalent to Thompson sampling, then it must learn to implicitly approximate Bayesian learning based on many tasks.  This seems harder than regular Thompson sampling in the sense that it will need a lot of data just to learn to approximate Bayesian learning.
* The experiments do not compare to regular Thompson sampling that learn a posterior distribution over $\theta$.
* The empirical evaluation is limited to one synthetic problem and one semi-synthetic problem

---

> ### Author Rebuttal · Authors · 2025-07-30
>
> Thank you for your helpful comments and questions!
>
> ## (1) Experiments comparing to standard Thompson Sampling
> *The experiments do not compare to regular Thompson sampling (TS) that learn a posterior distribution over $\theta$.*
>
> We **do** compare to standard linear Thompson Sampling (TS) in our simulations. See *"TS-Linear"* and *"TS-Linear (fit w history)"* in Figure 5 and Section 6. These use Gaussian priors: one non-informative and one fit using historical data (Appendix B.3.2).
>
> ## (2) Relationship to Thompson Sampling
> *Can you prove that the proposed algorithm is equivalent to Thompson sampling?*
>
> Thank you for the question. In the revision we will add the Proposition 2 (see below) and associated proof. We first provide some relevant context on Thompson Sampling and our work, which we will clarify in our revision.
>
> **(2a) Thompson Sampling is probability matching.** "Standard" Thompson sampling (TS) uses *probability matching* to select actions, which means *TS selects an action $a$ according to the posterior probability that action $a$ is optimal.* For example a linear contextual bandit TS algorithm selects actions according to the posterior probability that action $a$ is optimal with context $X_t$ (note $X_t \\in \\mathcal{H}_t$):
> $$\\mathbb{P}(A\_t = a \\mid \\mathcal{H}\_t)
> = \\mathbb{P}( \\text{argmax}\_{a'} ~\\beta^{\\top} \\varphi(X\_t, a') = a \\mid \\mathcal{H}\_t).$$  Above, the probability on the right hand side averages over the posterior distribution of the latent parameter $\\beta$ given $\\mathcal{H}_t$.
>
> In the bandit and RL literature, the *"probability matching"* property is commonly taken to be the abstract definition of TS. When generalizing TS to MDPs [Osband et al], the algorithm samples an MDP from the posterior, picks the optimal policy for that sampled MDP, and applies that policy to the states encountered. Our algorithm fits this framework: we sample a dataset $\hat{\tau}$ from the posterior, compute the best-fitting policy $\pi^*(\cdot; \hat{\tau})$, and act according to that policy. We formalize the probability matching property of our algorithm next.
>
> **(2b) Our algorithm uses probability matching to select actions.** Our TS-Gen algorithm uses probability matching to select actions:
> $$\\mathbb{P}(A\_t = a \\mid \\mathcal{H}\_t) = \\mathbb{P}(\\pi^*(X\_t; \\tau) = a \\mid \\mathcal{H}\_t)$$
>
> Above, $\\mathbb{P}(\\pi^*(X\_t; \\tau) = a \\mid \\mathcal{H}\_t)$ denotes the posterior probability that under the best-fitting policy on the true dataset $\tau$ that $a$ is the best action; the probability averages over randomness over unobserved outcomes in $\tau$. The above property holds because through our sampling procedure, the conditional distributions of the true $\tau$ and imputed $\hat{\tau}$ datasets are the same conditional on the history $\mathcal{H}_t$; see Proposition 2 below for the proof.
>
> With approximate models $p_\theta$, our algorithm performs probability matching under the $p_\theta$-defined distribution; Theorem 2 quantifies the regret due to this misspecification.
>
> **Proposition 2.** *Algorithm 1 with imputation model $p^\*$ implements Thompson Sampling (probability matching), i.e., the following holds with probability $1$:*
> $$\\mathbb{P}(A\_t = a \\mid \\mathcal{H}\_t)
> = \\mathbb{P}(\\pi^*(X\_t; \\tau) = a \\mid \\mathcal{H}\_t).$$
>
> *Proof:* Recall that Algorithm 1 selects actions as follows:
> $\\mathbb{P}(A\_t = a \\mid \\mathcal{H}\_t) = \\mathbb{P}(\\pi^*(X\_t; \\hat{\\tau}) = a \\mid \\mathcal{H}\_t).$
>
> Since $\\hat{\\tau} \sim p^*( \tau \in \cdot \mid \mathcal{H}_t)$ and from Eq (1) $\\tau \\sim p^\*$, the distributions of $\tau$ and $\hat{\tau}$ are equal given $\mathcal{H}_t$. Hence, with probability $1$ for any $j$:
> $\\mathbb{P}(\\hat{\tau} = j \\mid \\mathcal{H}\_t) = \\mathbb{P}(\\tau = j \\mid \\mathcal{H}\_t).$
>
> The above implies that $\\mathbb{P}(\\pi^\*(X\_t; \\hat\\tau) = a \\mid \\mathcal{H}\_t) = \\mathbb{P}(\\pi^\*(X\_t; \\tau) = a \\mid \\mathcal{H}\_t).$ Combining the above statements gives the result. $\square$
>
> **(2c) Our work is a generalization of Thompson Sampling (TS).** We make a substantial generalization of standard TS in this work by replacing the "optimal policy" with an abstracted notion of a "best-fitting oracle policy" $\pi^\*(X_t; \tau)$, which is the policy you would want to use if you had the entire dataset $\tau$. It is appropriate to view our method as a "generalized" TS that targets learning a "best-fitting oracle policy". A more "standard" TS algorithm is recovered when the best-fitting policy is correctly specified. We will clarify these points in the revision.
>
>
> ## (3) Advantages of our algorithm
> *What are the advantages of the proposed algorithm over Thompson sampling?*
>
> We note several advantages and discuss what revisions we plan to make below.
>
> **(3a) Learning the Bayesian model from data.** Recall that in our news recommendation problem setting, we need learn across tasks how to form prior beliefs conditioned on article text, i.e., fit an appropriate Bayesian model given high-dimensional text features. A strength of our approach is that an appropriate Bayesian model can be learned through standard offline sequence prediction and gradient descent (see empirical Bayes example below); furthermore, our regret bound (Theorem 2) shows that minimizing this loss provably controls regret. This benefit of being able to leverage existing methods for training offline sequence models relies on the fact taht we directly modeling potentially observable outcomes $\tau$, rather than unobserved latent variables (as is standard in Thompson Sampling). Below we discuss an example of how our pretraining procedure relates to standard methods for fitting Bayesian models from data, i.e., empirical Bayes (Type-II maximum likelihood). We mention this connection to empirical Bayes in the draft briefly in line 209, but will expand on this point in the revision.
>
> **Empirical Bayes connection:**  *As a simple example, in a Beta-Bernoulli model:*
> $$\mu_a \sim \text{Beta}(\alpha^\*, \beta^\*), \qquad Y_1^{(a)}, Y_2^{(a)}, \dots \big| \, \mu_a \overset{iid}{\sim} \text{Bernoulli}(\mu_a).$$ *The above model is associated the following class of sequence models (posterior predictives):*
> $$p_{\theta} \big( Y_{t+1}^{(a)} = 1 \mid Y_{1:t}^{(a)} \big) =  \frac{ \alpha  + \sum_{i=1}^{t} Y_{i}^{(a)} }{ \alpha + \beta + t } \quad \text{where} \quad \theta = (\alpha, \beta).$$ *The true sequence model $p^\*$ is $p_\theta$ evaluated at $\theta = (\alpha^\*, \beta^\*)$. Minimizing our negative log likelihood sequence loss from (6) is equivalent to maximizing the marginal likelihood of the data, the criterion that is used in Empirical Bayes to fit prior distributions to observed data.*
>
> **(3b) Agnostic approach that accommodates complex outcome/reward models.** There has been a lot of interest in being able to effectively generalize Thompson Sampling beyond parametric models to settings that require more complex outcome/reward, and at the same time still provide theoretical guarantees. Our work does this, as we allow our outcome (imputation) model, $p^*$, to be exceptionally complex and provide a regret bound (Theorem 1) that does not directly depend on the complexity of this imputation model (but instead on the complexity of the chosen "oracle" policy discussed next). Moreover, our algorithmic procedure (which involves fitting and sampling from a sequence model), is computationally more straightforward than many methods for Thompson Sampling with Bayesian neural networks [Riquelme et al, Tran et al].
>
> **(3c) Tolerance for general "oracle" policy construction.** Many works on bandit algorithms assume that the true policy class is known (e.g., the expected reward is linear in the context, so the true policy class is based on a linear model). Our work is a generalization of this standard approach in that (i) if the true policy class is known, one can choose that to be the "oracle" policy, i.e., the definition of $\pi^*$ from line 137, and (ii) if the true policy class is unknown or the policy class is misspecified, our regret bound provides a best-in-(policy)-class guarantee.
>
>
> ## (4) Z in the news recommendation setting
> *In the new recommendation example, why does Z correspond to a news article? It seems more natural to think of each action as a news article. If the actions are not news articles, then what are the actions?*
>
> We use $\\mathcal{A}\_\\tau$ to denote the action space and $Z\_\\tau = \\{ Z\_\\tau^{(a)} \\}_{a \\in \\mathcal{A}\_\\tau}$ to denote *action features*. In the news recommendation setting an action $a \\in \\mathcal{A}\_\tau$ refers to a particular news article and $Z\_\\tau^{(a)}$ refers to features associated with that news article; in general, features could include the news outlet, article type, article text, and article photos. We separate actions $a$ and their associated features $Z\_\\tau^{(a)}$ since for some choices of features (e.g. news outlet) two distinct actions may have the same action feature (e.g. two articles are distinct but come from the same news outlet).  Note though that even with action features $Z\_\\tau^{(a)}$ there is still residual uncertainty in how well that action will perform (e.g., two articles with similar features $Z\_\\tau^{(a)}$ could perform differently), which requires exploration to learn. We will make these points clearer in the revision.
>
> ## References
> - Tran, Snoek, & Lakshminarayanan. *Practical uncertainty estimation and out-of-distribution robustness in deep learning.* NeurIPS Tutorial 2020.
> - Osband, Russo, & Van Roy. *(More) Efficient Reinforcement Learning via Posterior Sampling.* NeurIPS 2013
> - Riquelme, Tucker, & Snoek. *Deep Bayesian Bandits Showdown: An Empirical Comparison of Bayesian Deep Networks for Thompson Sampling.* ICLR 2018

---

> > ### Author Response · Authors · 2025-08-05
> >
> > Hello! We're writing to gently follow up on our rebuttal, as we haven't heard from you yet. If you have any further questions or would like additional clarification, please don't hesitate to let us know. We'd be happy to provide more information.

---

> > ### Comment · Reviewer_y7h5 · 2025-08-06
> > **response to rebuttal**
> >
> > Thank you for the explanation.  Let me clarify some of my questions.
> >
> > (1) First regarding the empirical comparison, I realize that TS-Gen is compared to TS-Linear.  My question is about comparing TS-Gen with regular TS that maintains a distribution over $\theta$ for the **same** architecture as $p_\theta$.  The problem with the comparison with TS-Linear is that it uses a linear model to predict $Y$.  In contrast, TS-Gen uses a sequence model (i.e., transformer) to compute a distribution over $Y$.  TS-Gen has an unfair advantage since it uses a more powerful model than TS-Linear.
> >
> > To be clear, here is the comparison I have in mind that would be fair.  Let regular TS use the same $p_\theta$ architecture and parameterization as TS-Gen.  In regular TS, we do not use $p_\theta$ to impute missing observations, but simply to predict the outcome $Y$ for a given context $X$ and action $A$.  In other words, while TS-Gen works with the full joint distribution $p_\theta$ to sample any $X$, $A$ or $Y$, in regular TS, we will use only the conditional distribution $p_\theta(Y|X,A,Z)$ since this conditional distribution is effectively modelling the reward distribution.  Regular TS would then operate as follows:
> >
> > Initialize history to empty set: $H \leftarrow \emptyset$
> >
> > Initialize $q(\theta|H)$ to be the prior over $\theta$
> >
> > Repeat:
> > 1. Receive $Z$ and $X$
> > 2. Sample $\theta \sim q(\theta|H)$
> > 3. For each action $A$, sample $Y_A \sim p_\theta(Y|Z,X,A)$
> > 4. Select action: $A' = argmax_A Y_A$
> > 5. Observe outcome $Y$ for $A'$
> > 6. Update posterior by approximate Bayesian learning: $q(\theta|H,(Z,X,A',Y)) \propto$ $q(\theta|H) p_\theta(Y|Z,X,A')$
> > 7. $H \leftarrow H \cup \{(Z,X,A',Y)\}$
> >
> > Until end of task
> >
> > (2) Regarding the relationship between regular TS and TS-Gen, the explanation provided is a good step.  I encourage you to formalize this further.  First provide a formal description of regular TS.  Then show whether TS-Gen is mathematically equivalent to regular TS if all computation is exact.
> >
> > I'm not following why TS-Gen is a generalization of regular TS.  Again, provide a formal description of regular TS and then show whether instantiating some aspect of TS-Gen yields regular TS.
> >
> > (3) Regarding the advantages of TS-Gen, can you elaborate on the expected accuracy and speed of TS-Gen vs regular TS?  First both algorithms require some form of approximate computation.   Regular TS cannot perform exact Bayesian learning and therefore the approximation will impact accuracy.  TS-Gen approximates the estimation of $p_*$ with $p_\theta$ which also impacts accuracy.  In both algorithms some steps are expensive: approximate Bayesian learning in regular TS and fitting a policy in TS-Gen.

---

> ### Author Response · Authors · 2025-08-06
> **Response to Q1 (we are still working on Q2 and Q3 responses!)**
>
> Thanks for your questions and comments! Here's our response to Q1:
>
> ## **Clarifying "Regular TS".**
> **The algorithm you've outlined, which selects the action with the highest sampled outcome ($Y_A$), is a common misunderstanding of standard TS.** To clarify, we rely on the standard definition of Thompson Sampling as established in Section 4 of "A Tutorial on Thompson Sampling" by Russo et al. (2018).
>
> ### Standard TS algorithm:
> Specify the prior $q(\mu)$. At the start of the bandit task, receive $Z$. Then, for each decision time $t = 1, 2, ... T$:
> 1. Receive $X_t$
> 2. Sample $\tilde{\mu} \sim q(\mu \mid H_t)$
> 3. Select action $A_t = \mathrm{argmax}\_{a \in \mathcal{A}} \mathbb{E}\_{\tilde{\mu}}[ Y \mid Z, X_t, A = a]$
> 4. Observe outcome $Y_t$ for action $A_t$
> 5. Update the history $H_{t+1} \gets H \cup (X_t, A_t, Y_t)$ and the posterior $q(\mu \mid H_{t+1})$
>
> For example, for Beta-Bernoulli multi-armed bandits (no $X$'s or $Z$'s),
> - The unknown parameters $\mu = (\mu_a)_{a \in \mathcal{A}}$ are the mean rewards associated with each action
> - **Specifying the prior:** $\mu_a \sim \mathrm{Beta}(\alpha_{1,a}, \beta_{1,a})$ for each $a \in \mathcal{A}$
> - **Step 2:** Form $\tilde{\mu} = (\tilde{\mu}\_a)_{a \in \mathcal{A}}$ by sampling from the posterior $\tilde{\mu}\_a \sim \mathrm{Beta}(\alpha\_{t,a}, \beta\_{t,a})$ for each $a \in \mathcal{A}$
> - **Step 3:** Select action $\mathrm{argmax}\_{a \in \mathcal{A}} ~ \tilde{\mu}_a$, i.e., $\mathbb{E}\_{\tilde{\mu}}[ Y \mid Z, X\_t, A = a] = \tilde{\mu}\_a$
> - **Step 5:** Update posterior of Beta distribution: $\alpha_{t+1,a} \gets \alpha_{t,a} + Y_t$ and $\beta_{t+1,a} \gets \beta_{t,a} + (1-Y_t)$
>
> Your proposed algorithm would replace step 3 above with
> - Sample $\tilde{Y}_a \sim \mathrm{Bernoulli}(\tilde\mu_a)$ and select $A_t \gets \mathrm{argmax} ~ \tilde{Y}_a$.
>
> Note that your proposed algorithm would likely perform poorly in practice because $\tilde{Y}_a$ are binary and there would be many ties. Even if the algorithm knows the true mean reward, $\mu_a$, for each action, it would not select the best action with probability $1$.
>
> ## **Comparing our algorithm to "regular TS".**
>
> We understand your concern about wanting to compare to a "stronger" regular TS baseline beyond a linear TS algorithm, by using a neural network approach with "approximate Bayesian learning". The literature has been quite stuck on what this "approximate Bayesian learning" should be and it's especially unclear how to learn across tasks and use very rich $Z$'s and $X$'s. We discuss several approaches one might take:
>
> 1. **Adapt our pretrained sequence model $p_\theta$ to use with standard TS.** The algorithm you proposed suggested that we might be able to use our $p_\theta$ sequence model with standard Thompson Sampling. However, our $p_\theta$ model is **not** a standard reward model that models $\mathbb{E}[Y \mid X, A, Z]$, but rather a sequence model that generates $Y$'s conditional on the history $H_t$. Furthermore, we do not put priors on the weights $\theta$ **as we do not treat them as a latent parameter**. Our pretraining procedure for learning $p_\theta$ is therefore not appropriate for learning a prior on the weights $\theta$, so it is not directly compatible with standard TS where one samples the latent parameters $\theta$ from the posterior distribution. To reiterate, the sampling in TS-Gen is not from sampling a latent parameter, but rather unknown outcomes in $\tau$, and $\theta$ is not a latent parameter in our setting to sample at all.
> 2. **Use a standard Bayesian Neural Network approach.** Our work considers a meta-bandit problem, where the goal is to use data from past tasks to perform well on new tasks. The Thompson sampling and Bayesian neural network literature do not have many practical methods for learning an informative prior over the weights of a neural network for meta-learning problems. We could compare to Bayesian neural network approaches use an *uninformative prior*, but a Bayesian neural network with a completely uninformative prior that ignores previous task data and uses approximate posterior inference seems like a complex and poor alternative algorithm.
> 4. **Use other approximate Bayesian methods with TS proposed in the literature.** When we looked into the literature for TS algorithms with neural networks, the most prominent and comprehensive work comparing methods empirically was *"Bayesian Bandit Showdown"* by Riquelme et al (ICLR 2018). Surprisingly, they found that simple methods like neural linear (linear TS with the last layer from a neural network as context) was the among the most promising methods and generally outperformed more complex methods like variational inference/Bayes-by-backprop, expectation-propagation, bootstrap, direct noise injection, dropout, Bayesian non-parametrics, etc. This is what led us to believe that a linear based TS algorithm would be a reasonable baseline.
>
> Please let us know if you have further comments on this point.

---

> > ### Comment · Reviewer_y7h5 · 2025-08-07
> > **concern remains**
> >
> > Thank you for the additional response.  My concern remains.  There should be a comparison with TS that uses the same architecture as $p_\theta$ used in your approach for fairness.

---

> > > ### Author Response · Authors · 2025-08-07
> > > **Addressing your question 2 from earlier**
> > >
> > > > (2) Regarding the relationship between regular TS and TS-Gen, the explanation provided is a good step. I encourage you to formalize this further. First provide a formal description of regular TS. Then show whether TS-Gen is mathematically equivalent to regular TS if all computation is exact. I'm not following why TS-Gen is a generalization of regular TS. Again, provide a formal description of regular TS and then show whether instantiating some aspect of TS-Gen yields regular TS.
> > >
> > > We really appreciate your engagement with the paper. To our understanding, the Proposition we provided in our earlier response does exactly what you're asking, so we're not sure if there is a misundersanding or what other result you think is missing. In particular the proability matching definition we provided is widely accepted as *the mathmatical definition of TS*, see *"An Information-Theoretic Analysis of Thompson Sampling"* by Russo et al, *"(More) Efficient Reinforcement Learning via Posterior Sampling"* by Osband et al, and also *"A modern Bayesian look at the multi-armed bandit"* by Steven Scott, which was a pioneering early work on TS. We recognize that you may be more familiar with an algorithmic definition, which involves sampling latent parameters and maximizing, but the two are mathmatically equivalent and this is well known: see "An Information-Theoretic Analysis of Thompson Sampling", section 4, first two paragraphs. The lemma we shared shows formally that TS-Gen exactly implements TS (i.e. the probability matching definition) if all computation is exact.
> > >
> > > Our guess is that you are looking for us to provide the alternative (equivalent) definition of regular TS that is also used in the litearture: sample a latent parameter and maximize. Our model in the paper does not explicitly involve latent parameter, but we can provide an appropriate formalization in the appendix to draw a rigorous connection, similar to Appendix D of *"Active Exploration via Autoregressive Generation of Missing Data"* by Cai et al.

---

> > > > ### Author Response · Authors · 2025-08-07
> > > > **Addressing your question 3 from earlier**
> > > >
> > > > > (3) Regarding the advantages of TS-Gen, can you elaborate on the expected accuracy and speed of TS-Gen vs regular TS? First both algorithms require some form of approximate computation. Regular TS cannot perform exact Bayesian learning and therefore the approximation will impact accuracy. TS-Gen approximates the estimation of $p^*$ with $p_\theta$ which also impacts accuracy. In both algorithms some steps are expensive: approximate Bayesian learning in regular TS and fitting a policy in TS-Gen.
> > > >
> > > > A natural way to incorporate Bayesian modeling with neural networks is to place a prior on each weight of the neural network, and to specify the likelihood of the observed data given those weights, i.e. Bayesian neural networks (BNNs). The comparison between TS-Gen and regular TS (especially when combined with BNNs) can be broken down into two key aspects: prior specification and posterior updating. Both of these are crucial for a TS algorithm's performance. Our approach offers significant advantages in both areas, which we'll detail below.
> > > >
> > > > The core difference is in how the two methods handle the Bayesian model. While both rely on approximation, TS-Gen's approach of using a sequence model offers a simpler, more flexible, and more scalable way to learn the prior and update the posterior.
> > > >
> > > > ## 1. Specifying the prior/Bayesian model
> > > > A well-specified prior is essential for good performance, especially in settings with a finite number of interactions. A key advantage of TS-Gen is its simple and practical method for learning this Bayesian model. Instead of placing a prior on each weight of a neural network (the standard approach for Bayesian neural networks), we fit a sequence model ($p_\theta$) to historical data. This approach has two major benefits:
> > > > 1. **Task-Specific Priors:** We can use standard machine learning tools to fit the sequence model to a particular task distribution, creating a prior that is naturally tailored to the problem. In contrast, for BNNs, the standard approach is to place independent Gaussian priors on neural network weights, which is not specific to a task or fit it with historical data. Furthermore, the literature lacks straightforward methods to fitting priors for BNNs, which could have millions or billions of weights.
> > > > 2. **Handling High-Dimensional Features:** Our sequence model can easily learn to condition on high-dimensional task features $Z$, a capability that is not easily achieved with traditional BNN priors.
> > > >
> > > > ## 2. Posterior updating
> > > > Posterior updating is where the computational and accuracy trade-offs are most apparent. For BNNs, the posterior distribution of the weights is not closed-form, requiring approximate inference methods. Here's a summary of the common methods and their challenges:
> > > > - **MCMC:** which is based on Monte Carlo sampling, can form exact posterior samples when converged. But it is often slow and it's difficult to know when it has converged. MCMC is especially challenging with a large number of parameters, as its convergence is often a function of the number of parameters (*"Bayesian neural networks via MCMC"* by Chandra et al.)
> > > > - **Variational Inference** approximates the posterior distribution with the best fitting distribution in a family. This method is generally faster than MCMC, but it is known to underestimate posterior variance (*"Variational inference: A review for statisticians"* by Blei et al.) and it has been shown that the inaccurate posterior approximation using variational inference can lead to significant increases in regret (*Thompson Sampling with Approximate Inference* by Phan et al).
> > > > - **Ensembles** attempt to approximate posterior distributions by training many ensembles or bootstrapped versions of a model. This can be computationally costly and updating these models generally involves fine-tuning the weights online, which can be computationally challenging. Theoretical guarantees for ensembling approaches with arbtitrary neural networks are limited.
> > > > - **Neural Linear** uses the last layer of a neural network as the context vector for a multivariate Gaussian model. Posteriors can easily be computed in closed form. This method is computationally stable, but does not model the uncertainty in the weights from all layers of the neural network.
> > > >
> > > > In contrast, the computational cost of TS-Gen comes from autoregressively generating outcomes from our sequence model ($p_\theta$). The main advantage here is that we can leverage extensive work in optimizing the speed of autoregressive generation from large language models.
> > > >
> > > > Regarding the quality of the posterior approximations, the main advantages of our work are:
> > > > 1. **Optimized Quality:** We can improve the quality of the posterior approximation by minimizing the sequence loss $\ell(p_\theta)$ using standard machine learning methods.
> > > > 2. **Quantifiable Regret:** We provide a formal way to quantify the cost in regret due to having a suboptimal sequence model, which is a significant theoretical contribution.

---

> > > > > ### Author Response · Authors · 2025-08-07
> > > > > **Responding to your earlier comment**
> > > > >
> > > > > Thank you again for your engagement with our paper! We now address your comment from earlier!
> > > > >
> > > > > > Thank you for the additional response. My concern remains. There should be a comparison with TS that uses the same architecture as $p_\theta$ used in your approach for fairness.
> > > > >
> > > > > To address your concern about a fair comparison, we have implemented an additional baseline: a neural linear Thompson Sampling (TS) algorithm that uses the exact same neural network model as our proposed TS-Gen method. This new baseline, which we refer to as "Neural Linear TS", uses the 100-dimensional last layer of the $p_\theta$ model (when $X_t$ and $Z$ are fed in) as the context for a linear TS model. We emphasize that both methods use the same $p_\theta$ model, which helps ensure the comparison is fair.
> > > > >
> > > > > The following table presents the cumulative regret for both our proposed TS-Gen method and the new Neural Linear TS baseline on our synthetic simulation setting.
> > > > >
> > > > > **Table 1:** Cumulative regret comparison of TS-Gen and Neural Linear TS.
> > > > >
> > > > > | | Regret t=100 | Regret t=200 | Regret t=300 | Regret t=400 | Regret t=500 |
> > > > > :------------------|------|------|------|------|------|
> > > > > | TS-Gen | 8.204| 13.652| 17.978| 21.434| 24.578|
> > > > > | Neural linear | 18.936| 28.694| 36.374| 42.464| 48.062|
> > > > >
> > > > > As shown in Table 1, the TS-Gen algorithm consistently outperforms the Neural Linear TS baseline, achieving a significantly lower cumulative regret across all time steps. The performance gap widens considerably as the number of rounds increases, demonstrating that the uncertainty quantification method used by our TS-Gen algorithm provides a distinct advantage over the traditional neural linear approach, even when using the same underlying $p_\theta$ model. Note that while TS-Gen uses $p_\theta$ as a autoregressive sequence model that takes in $H_t$, Neural Linear TS simply uses $p_\theta$ by feeding in a single $(X_t, Z)$, and outputting the last layer.
> > > > >
> > > > > Our choice of the neural linear approach as a baseline is motivated by its strong performance in existing literature (as discussed in one of our previous responses). The *"Bayesian Bandit Showdown"* paper by Riquelme et al. (ICLR 2018) found that simple methods like neural linear were among the most promising, often outperforming more complex Bayesian deep learning techniques for bandit problems like variational inference and Bayes-by-backprop. This makes neural linear a strong and relevant baseline for comparison. The results of this new experiment demonstrate that our generative approach improves upon this method within the synthetic setting we've evaluated.

---

> > > > > > ### Comment · Reviewer_y7h5 · 2025-08-09
> > > > > > **thank you**
> > > > > >
> > > > > > Thank you for the additional explanation.  I have no further question.

---

### Official Review · Reviewer_iwTT · 2025-07-02

**Clarity:** 3
**Significance:** 2
**Originality:** 2
**Rating:** 2
**Confidence:** 2

**Summary:**

This paper studies the meta contextual bandit problem where the learner aims to minimize the regret in learning a number of tasks (with ground truth distribution $p\_*$). The learner first learns the distribution of the tasks and then performs online decision making.

**Questions:**

1. In Theorem 2, it does not appear very clear to me why $\ell(p_\theta)\ge \ell(p_*)$. Could the authors explain it?
2. Theorem 1 seems to good to be true to me. The "best-fitting" policy is not quite restrictive in this paper, but if the policy incurs a constant gap compared with the optimal policy, then there should be a constant regret, instead of the $1/\sqrt{T}$ regret in Theorem 1. Moreover, if we restrict the setting to a single task of a stochastic linear bandit, then $p_*$ is trivial, and the regret is $\tilde O(d\sqrt T)$ (combining Theorem 1 and Proposition 1), which does not seem to be able to be achieved by the simple algorithm in this submission.
3. Does it make sense if $p_\theta$ is learned online? If it does, can the constant term $\sqrt{\ell(p_\theta)-\ell(p_*)}$ be replaced by some sub-constant term?

**Ethical Concerns:**

["NO or VERY MINOR ethics concerns only"]

**Final Justification:**

I would still hesitate to recommend accepting this work, after the discussion with other reviewers. My major concerns are listed in the [official comment](https://openreview.net/forum?id=Fqsl9IfbfJ&noteId=P9uHPy29kT).

For point 1, I think the setting is reasonable but probably not very practical. For point 2, I think the lack of a ground truth policy is a major issue.

**Limitations:**

I don't find any undiscussed limitations or potential negative societal impact of this work.

**Quality:**

2

**Strengths And Weaknesses:**

## Strengths

1. Although I am unfamiliar with the literature, the problem formulation section is very informative, and the example of news recommendation system makes the problem setting very clear.
2. The theoretical findings of this work are solid and discussed with sufficient details.

## Weaknesses

1. I would doubt whether the the algorithm is essentially Thompson sampling. Classical Thompson sampling typically involves constructing a distribution over the **reward function**, and the action is selected to maximize the sampled reward function. However, the action selection of the algorithm in this paper seems to depend on the policy fitting. I would agree with the discussion in the section of related works that the algorithm framework in this work is completely different from previous works, but I would hesitate to refer to this method as Thompson sampling.
2. The algorithm design looks quite straightforward, and I cannot evaluate the novelty of the algorithm because the authors only compare it against the TS literature but not the meta contextual bandit literature. Although there is a comparison in the section of previous works, the comparison is not technical enough to show the technical contributions. It may seem to me that this paper is some simple techniques disguised in a complex framework.

I find the technical findings of this paper quite solid, but currently score this submission as "reject" mostly due to my unfamiliarly of the literature and inability to evaluate the contributions of this paper. As the proof is quite short, I am unsure about the contributions of the proof techniques either. I would be happy to discuss the contributions of this submission with other reviewers and the AC.

---

> ### Author Rebuttal · Authors · 2025-07-30
>
> Thank you for your comments and questions!
>
> ## (1) Relationship to Thompson Sampling (TS)
> In the revision, we’ll clarify the following points and include Proposition 2 and its proof, which shows our algorithm performs probability matching, i.e., Thompson Sampling.
>
> **Thompson Sampling (TS) is probability matching and our algorithm uses probability matching.** TS selects actions according to the posterior probability that they are optimal. In the bandit/RL literature, the *"probability matching"* property is commonly taken to be the abstract definition of TS. When generalizing TS to MDPs [Osband et al], the algorithm samples an MDP from the posterior, picks the optimal policy for that sampled MDP, and applies that policy to the states encountered. Our algorithm fits this framework: we sample a dataset $\hat{\tau}$ from the posterior, compute the best-fitting policy $\pi^*(\cdot; \hat{\tau})$, and act according to that policy.
>
> We show in Proposition 2 (below) that, when using the correct model $p^\*$, our algorithm satisfies: $\mathbb{P}(A_t = a \mid \mathcal{H}_t) = \mathbb{P}(\pi^\*(X_t; \tau) = a \mid \mathcal{H}_t).$ Above, $\mathbb{P}(\pi^\*(X_t; \tau) = a \mid \mathcal{H}_t)$ which is probability matching with respect to the oracle policy. This aligns with the abstract definition of TS.
>
> **Proposition 2.** *Algorithm 1 with imputation model $p^\*$ implements Thompson Sampling (probability matching), i.e., the following holds with probability $1$:*
> $$\\mathbb{P}(A\_t = a \\mid \\mathcal{H}\_t)
> = \\mathbb{P}(\\pi^*(X\_t; \\tau) = a \\mid \\mathcal{H}\_t).$$
>
> *Proof:* Recall that Algorithm 1 selects actions as follows:
> $\\mathbb{P}(A\_t = a \\mid \\mathcal{H}\_t) = \\mathbb{P}(\\pi^*(X\_t; \\hat{\\tau}) = a \\mid \\mathcal{H}\_t).$
>
> Since $\\hat{\\tau} \sim p^*( \tau \in \cdot \mid \mathcal{H}_t)$ and from Eq (1) $\\tau \\sim p^\*$, the distributions of $\tau$ and $\hat{\tau}$ are equal given $\mathcal{H}_t$. Hence, with probability $1$ for any $j$:
> $\\mathbb{P}(\\hat{\tau} = j \\mid \\mathcal{H}\_t) = \\mathbb{P}(\\tau = j \\mid \\mathcal{H}\_t).$
>
> The above implies that $\\mathbb{P}(\\pi^\*(X\_t; \\hat\\tau) = a \\mid \\mathcal{H}\_t) = \\mathbb{P}(\\pi^\*(X\_t; \\tau) = a \\mid \\mathcal{H}\_t).$ Combining the above statements gives the result. $\square$
>
> **Our algorithm generalizes TS** by replacing the true reward-maximizing policy with a best-fitting policy under a given policy class $\Pi$. A more "standard" TS algorithm is recovered when the best-fitting policy is correctly specified. We will clarify these points in the revision.
>
>
> ## (2) Novelty and Relationship to Meta-Bandits
> *The algorithm design looks quite straightforward, and I cannot evaluate the novelty...*
>
> The algorithm we present is "straightforward" by design. The real novelty of our work is that we are able to provide theoretical guarantees, even when our algorithm takes advantage of neural network sequence models.
>
> **(2a) Relationship to meta-bandit literature.** Our work differs from much of the meta-bandit literature in two key ways (see Related Works line 306 for further details). To summarize:
> - **Model complexity.** Much of the meta-bandit literature focuses on algorithms that use simple models, e.g., linear or parametric models [Cella et al, Kveton et al]. Our approach uses autoregressive sequence models to capture complex dependencies across task-features $Z_\tau$, contexts $X$, and actions $A$, enabling richer reward modeling.
> - **Regret guarantees.** Unlike typical meta-bandit algorithms, our regret bounds hold under misspecification of both the Bayesian model ($p_\theta$) and the policy class $\Pi$.
>
> We also include comparisons to a meta-TS baseline in experiments (TS-Linear fitted with history, Section 6 and Appendix B.3.2).
>
> **(2b) Novelty.** Even beyond the meta-bandit literature, our theoretical guarantees have no precedent in the literature (see Section 3.1, especially line 172). To clarify further:
>
> - **Entropy-based regret bound.** Our regret bound depends on $H (\mathbf{\pi}^*(X_{1:T}) \mid Z_\tau)$ the entropy of oracle policy decisions, rather than the entropy of latent parameters [Neu et al, Dong et al]. We boud this term by the VC/Natarajan dimension of the policy class $\Pi$, which is new in the information-theoretic stochastic bandit literature.
> - **Accommodates complex reward models.** There is great interest in algorithms that generalize TS beyond parametric models and still have theoretical guarantees. Our work does this, as we allow our outcome (imputation) model to be highly expressive and provide a regret bound (Theorem 1) that does not directly depend on the complexity of this imputation model. We also characterize the regret when the imputation model $p_\theta$ is misspecified (Theorem 2).
> - **Flexible policy classes.** Many bandit works that assume the true policy class $\Pi$ is known (e.g. linear bandits). Our work takes a flexible approach: (i) if the true policy class is known, one can choose that to be the "oracle" policy, and (ii) if the true policy class is unknown or misspecified, our regret bound provides a best-in-(policy)-class guarantee.
> - **Dependence on pretraining loss.** It is common in TS regret bounds to assume the true Bayesian model (prior and likelihood) is known by the algorithm. For our algorithm, when an approximate $p_\theta$ model is used, the regret of our algorithm is bounded in terms of the pretraining sequence loss $\ell(p_\theta)$, which can be optimized offline. This connects to recent work on analyzing TS under misspecified priors for parametric bandits  [Simchowitz et al, Liu et al].
>
>
>
> ## (3) In Theorem 2 questions
> ### **(3a) Loss inequality**
> *In Theorem 2, it does not appear very clear to me why $\ell(p_\theta) \geq \ell(p^\*)$.*
>
> The inequality $\ell(p_\theta) \geq \ell(p^\*)$ follows directly from Lemma 1 (Appendix A.3.1), which shows:
> $$\ell(p\_\theta) = \ell(p^\*) + D\_{\text{KL}} \left( p^\* \big(X\_{1:T}, \{ Y\_{1:T}^{(a)} \}\_{a \in \mathcal{A}\_\tau} \mid Z\_\tau \big) \| ~p_\theta \big(X\_{1:T}, \{ Y\_{1:T}^{(a)} \}\_{a \in \mathcal{A}\_\tau} \mid Z\_\tau \big) \right).$$ Since the KL divergence is non-negative, the the result follows. Intuitively, this inequality is saying that $p^\*$ minimizes the sequence prediction loss, and any misspecification in $p_\theta$ results in higher loss.
>
> ### **(3b) Choice of policy class**
> *The "best-fitting" policy is not quite restrictive in this paper, but if the policy incurs a constant gap compared with the optimal policy, then there should be a constant regret...*
>
> It is common in the bandit literature to assume that the correct policy class is known (e.g., the expected reward is linear in the context). **Our theoretical results cover this case in which the correct policy class is known to the algorithm design a priori, and also provide theoretical results when this policy class is misspecified.** As discussed in point (1) above, we make a substantial generalization of standard TS in this work by replacing the "optimal policy" with an abstracted notion of a "best-fitting oracle policy" $\pi^\*(X_t; \tau)$, which is the policy you would want to want to fit if you had the entire dataset $\tau$. It's appropriate to view our method as a "generalized" TS that targets learning a "best-fitting oracle policy", which may or may not be the one that exactly maximizes the reward in each context, depending on the choice of the oracle policy fitting procedure $\pi^\*(\cdot)$. A more "standard" TS algorithm is recovered when the best-fitting policy is correctly specified.
>
> ### **(3c) Regret achievable by our algorithm**
> *Theorem 1 seems to be good and true to me... If we restrict the setting to a single task of a stochastic linear bandit, then $p^\*$ is trivial, and the regret is $O(d \sqrt{T})$ (combining Theorem 1 and Proposition 1), which does not seem to be able to be achieved by the simple algorithm in this submission.*
>
> We did not fully understand your comment. As you state, *"Theorem 1 seems to be good and true to me"* and the regret bound follows from Theorem 1 and Proposition 1. We clarify below:
> - Theorem 1 proves the following upper bound the *cumulative* regret: $\sqrt{\frac{1}{2} |\mathcal{A}| T \cdot H(\mathbf{\pi}^*(X_{1:T}) \mid Z_\tau) }$.
> - Proposition 1, shows that $H(\mathbf{\pi}^*(X_{1:T}) \mid Z_\tau) = O(d \cdot \log(T \cdot |\mathcal{A}|))$, where $d$ is the Natarajan dimension of the policy class $\Pi$.
> - In stochastic linear bandits with policies $\text{argmax}_a ~\beta^\top \varphi(X_t, A_t)$ for some $\beta \in \mathbb{R}^d$ have Natarajan dimension $d$ (Appendix A.5).
> - The resulting bound is $O \sqrt{ |\mathcal{A}| T d \log(T |\mathcal{A}|) }$
>
>
> ## (4) Learning $p_\theta$ online
> *Does it make sense if $p_\theta$ is learned online?*
>
> We assume access to past bandit tasks (offline meta-bandit), as in prior work [Cella et al, Hong et al]. To extend to a sequential task setting, one could retrain $p_\theta$ online (as new tasks arrive), which we expect would decrease the gap $\ell(p_\theta) - \ell(p^*)$, and thus the regret bound. Future work could explore this retraining more formally.
>
> ## References
> - Cella et al. *Meta-learning with stochastic linear bandits.* ICML 2020
> - Dong & Van Roy. *An information-theoretic analysis for TS with many actions.* NeurIPS 2018
> - Hong et al. *TS with a mixture prior.* AISTATS 2022
> - Kveton et al. *Meta-TS.* ICML 2021
> - Liu & Li. *On the prior sensitivity of thompson sampling.* ALT 2016
> - Neu et al. *Lifting the information ratio: An information-theoretic analysis of TS for contextual bandits.* NeurIPS 2018
> - Osband, Russo, & Van Roy. *(More) Efficient Reinforcement Learning via Posterior Sampling.* NeurIPS 2013
> - Simchowitz et al. *Bayesian decision-making under misspecified priors with applications to meta-learning.* NeurIPS 2021

---

> > ### Author Response · Authors · 2025-08-05
> >
> > Hello! We're writing to gently follow up on our rebuttal, as we haven't heard from you yet. If you have any further questions or would like additional clarification, please don't hesitate to let us know. We'd be happy to provide more information.

---

> ### Comment · Reviewer_iwTT · 2025-08-05
>
> Many thanks for the rebuttal! I am sorry but I have been trying to understand the paper and the rebuttal. However, I still do not think this work is up to the standard of the acceptance:
>
> 1. **Problem with learning objective.** The definition of the "best-fitting" policy makes the objective a bit ambiguous. Commonly in the bandit literature, the objective is to learn an optimal policy, but in this work, the policy given the history comes from the oracle. Since the policy is not directly learnable, my understanding is that the main goal is to learn $p^*$, i.e., item (i) in Line 73. However, **nowhere in the algorithm design permits the update of the imputation model $p$.**
> 2. **Contributions and novelty.** The authors claim the following technical novelties: (i) Entropy-based regret bounds, which is interesting but not surprising given the intuition that the regret of the algorithm should depend on the uncertainty of the policy; (ii) Complex reward model, which is possible because the a general best-fitting policy is assumed and no further assumptions are needed to obtain (approximations of) good policies; (iii) flexible policy class & dependence on pretraining loss, which is rendered by the simplicity of analysis. In other words, my understanding is that the work is able to study very general settings because the analysis does not pose too much restrictions to the policy class, imputation model class, etc.
> 3. **Definition of the regret.** I made the point that the regret bound was too good to be true, and it turned out that the definition of the regret is defined as the average of the difference between the **best-fitting policy** and the reward of the action at step $t$. The regret bound becomes unsurprising because the the action at step $t$ is drawn from the best-fitting policy, so there should be no such problem as the constant estimation error in Question 2. I find this definition problematic because this definition disables the discussions about the error between the learner's estimated policy and the best-fitting policy (which I believe is the main source of error), and reduces the regret to some metric of "variance error" or "uncertainty". For lines 100-103, I do not think the regret is the same as the previous works because in this work, the learner has access to the best-fitting policy but not in previous works.
>
> In conclusion, I would like to maintain my score for now. I have also read all other reviews, and find the weaknesses raised by other reviewers mostly minor. I would ask the AC to carefully consider this message, and would like to hear the clarification of the authors.

---

> > ### Author Response · Authors · 2025-08-06
> >
> > Thanks for your response.
> >
> > ## **Clarifying the best-in-class/best-in-hindsight regret objective.**
> >
> > > Q1: The definition of the "best-fitting" policy makes the objective a bit ambiguous. Commonly in the bandit literature, the objective is to learn an optimal policy, but in this work, the policy given the history comes from the oracle.
> >
> > > Q3: I made the point that the regret bound was too good to be true, and it turned out that the definition of the regret is defined as the average of the difference between the best-fitting policy and the reward of the action at step $t$. The regret bound becomes unsurprising because the the action at step is drawn from the best-fitting policy, so there should be no such problem as the constant estimation error in Question 2. I find this definition problematic because this definition disables the discussions about the error between the learner's estimated policy and the best-fitting policy (which I believe is the main source of error), and reduces the regret to some metric of "variance error" or "uncertainty". For lines 100-103, I do not think the regret is the same as the previous works because in this work, the learner has access to the best-fitting policy but not in previous works.
> >
> > Your comments suggest that there may be a misunderstanding of what we call the "best-fitting" policy and what this means for our regret notion.
> >
> > The "best-fitting" policy is well-defined and is a well-established concept in the bandit literature, representing a generalization of the optimal policy. Refer for example to Section 2 of *Contextual Bandit Algorithms with Supervised Learning Guarantees* by Beygelzimer et al. (2011) for a very analogous objective. Their work seeks to attain low regret with respect to the best policy *in hindsight over a fixed policy class* -- i.e the policy that would have maximized reward if deployed across all time periods. This notion of a best-in-hindsight optimal policy is the most notable special case of the "best-fitting" policy we define. In that case, our regret notion is the same one used in Beygelzimer et al. or in the contextual bandits chapter of the textbook *Regret Analysis of Stochastic and Nonstochastic Multi-armed Bandit Problems* by Bubeck and Cesa-Bianchi. We emphasize that learner does not **does not** *"have access to the best-fitting policy"* -- which would trivialize our problem. They have access to a **means to compute** the best-fitting policy *if they had observed rewards of every arm in every context*.
> >
> > One possible source of confusion is that our analysis lets a practitioner pick other notions of a "best-fitting policy" too. Instead of "reward-maximizing policy in hindsight" they can specify other learning targets --- like those that are more tractable to compute, incorporate fairness or constraints, or regularizations --- and attain regret bounds relative to their specified goal. This is discussed in the paragraph starting on line 89. We emphasize that just like the optimal policy is unknown, the "best-fitting policy" is unknown to the learner because they lack the full data on rewards $\tau$.
> >
> > ## **Learning $p^*$ is not the main goal.**
> > > Q1: Since the policy is not directly learnable, my understanding is that the main goal is to learn $p^\ast$, i.e., item (i) in Line 73. However, nowhere in the algorithm design permits the update of the imputation model $p^\ast$.
> >
> > There is some understandable confusion here, as we've generalized and relaxed regular Thompson sampling. First, traditional Bayesian formulations all assume $p^\ast$ is known. (It is given by $p(y_{1:T}) = \intop \prod_{t=1}^{T} p(y_t \mid \eta ) p(\eta) d\eta )$ where $\eta$ is some latent parameter, $p(y_t\mid \eta)$ is a known likelihood and $p(\eta)$ is a known prior). We relax this by allowing $p^*$ to be learned offline across meta-learning tasks an providing rigorous theory on how approximation errors impact regret. While we provide this generalization, the main goal of our algorithm is to minimize best in class regret in the online exploration. In fact, Theorem 1 provides a regret bound assuming $p^\ast$ is known which is akin to assuming the prior and likelihood for a Bayesian reward model are known -- typical assumptions made for Thompson Sampling analyses.

---

> > > ### Author Response · Authors · 2025-08-06
> > >
> > > ## **Contributions and novelty.**
> > > > Q2: Contributions and novelty. The authors claim the following technical novelties: (i) Entropy-based regret bounds, which is interesting but not surprising given the intuition that the regret of the algorithm should depend on the uncertainty of the policy; (ii) Complex reward model, which is possible because the a general best-fitting policy is assumed and no further assumptions are needed to obtain (approximations of) good policies; (iii) flexible policy class & dependence on pretraining loss, which is rendered by the simplicity of analysis. In other words, my understanding is that the work is able to study very general settings because the analysis does not pose too much restrictions to the policy class, imputation model class, etc.
> > >
> > > We feel it is a major strength of this work that we provide intuitive regret bounds in that hold rigorously in *"very general settings."* We view it as very desirable that our  *"analysis does not pose too much restrictions to the policy class, imputation model class,"* instead providing a theory that enables flexibility by practitioners to (a) use engineering tricks to reduce the offline validation loss of the imputation model and (b) a select policy class that may incorporate auxiliary considerations (like fairness constraints) if they choose.

---

> ### Comment · Reviewer_iwTT · 2025-08-06
>
> Many thanks for the reply!
>
> **About best-fitting policy:** My current understanding is that the best-fitting policy is the best policy if all data, including data from other tasks and arms, were given. If this understanding is correct, then I have two questions: (i) What does the policy fitting oracle output during the process of the algorithm, when the learning agent does not have to the entire dataset? (ii) How to define the rewards from other tasks and arms if the reward is not queried?
>
> These two questions are currently my major confusions.

---

> ### Author Response · Authors · 2025-08-06
>
> Thanks for your engagement with our work.
>
> ### (i) What does the policy fitting oracle output during the process of the algorithm, when the learning agent does not have to the entire dataset?
> You're hitting on a key point here. During the online decision-making process, the algorithm does not have a complete dataset of each arm's rewards for all contexts in the current task. Our method proceeds by probabilistically imputing/sampling the unknown rewards with an autoregressive sequence model and then computing the best-fitting policy under that imputed dataset. This is done very carefully to preserve appropriate randomness in the imputation/sampling of the unknown rewards, and that randomness drives careful exploration that provably controls regret.
>
> ### (ii) How to define the rewards from other tasks and arms if the reward is not queried?
>
> Rewards associated with arms that may or may not be selected are defined as *potential outcomes*. This is a standard framework in causal inference and bandit literature. Under this framework, all the rewards from a task (from all contexts and arms) are sampled by the environment ahead of time, prior to the algorithm starting to make decisions. Then when the algorithm makes a decision, the environment reveals rewards to the algorithm associated with the selected action.

---

> > ### Comment · Reviewer_iwTT · 2025-08-08
> >
> > Many thanks for the clarification! This explanations would greatly benefit understanding if added after Eq. (1).
> >
> > May I also ask for the references about the setting, since you mentioned in the answer to Question (ii) that "this is a standard framework in causal inference and bandit literature"?
> >
> > I will discuss with other reviewers and the AC before I compose the final justification.

---

> > > ### Author Response · Authors · 2025-08-08
> > >
> > > Thanks your your reply!
> > >
> > > We can add additional clarifications after Eq (1) in the revision.
> > >
> > > Regarding references on potential outcomes here are some:
> > > - In the causal inference literature, see the textbook *"Causal Inference for Statistics, Social, and Biomedical Sciences"* by Imbens and Rubin. See chapter 1 for where potential outcomes are first introduced. They are also used throughout the textbook.
> > > - A reference in the bandit literature is the *"Introduction to multi-armed bandits"* by Slivkins. Though they do not use the term "potential outcomes" they describe a list of potential rewards as a "reward tape" in Section 1.3.1. (Note, the order in which potential rewards are revealed slightly differ between Slivikin's paper compared to our work. But both works use the concept of potential rewards/outcomes.)

---

### Official Review · Reviewer_ik19 · 2025-07-03

**Clarity:** 2
**Significance:** 3
**Originality:** 3
**Rating:** 5
**Confidence:** 3

**Summary:**

The authors propose a generative approach for contextual Thompson sampling by leveraging an offline pretrained model for imputing trajectories. The authors provide a meta-framework with (i) a task-dependent variable $Z$ and (ii) observed trajectories $(X_t^{a}, Y_t^{a})_{a \in \mathcal{A}}$ for imputing missing potential outcomes. Using the task-dependent vector $Z$ with observed histories to impute missing observations (e.g. potential outcomes, future observations, etc.), one samples tasks from a posterior sample and fit a policy $\pi \in \Pi$ that solves the task on the imputed data. The authors provide methods of imputing data through autoregression and theoretical results relating regret with the quality of the imputed data. Synthetic experiments are provided to demonstrate the efficacy of this approach.

**Questions:**

* It still seems somewhat unclear why the design-based approach (viewing observations as fixed, but unobserved) helps with the interpretation of the approach provided by the authors. It would be nice to have a section discussing the distinction between this "fixed-sample" approach vs. distributional approach, and how this perspective results in the development and superior performance of the authors' approach.
* Does the performance of this approach depend heavily on the author's choice of imputation architecture? It would be of interest to see how these approaches perform with generic imputation methods beyond the neural network proposed by the authors - using a neural network for the low-dimensional examples provided may be an "easy" task relative to having either (i) a more complex DGP and/or (ii) a less powerful model. While there is theory for the regret bounds based on the quality of imputations, it would be beneficial to see how empirical performance degrades (as bandit regret bounds often provide little guidance for the actual empirical performance of standard approaches.)

**Ethical Concerns:**

["NO or VERY MINOR ethics concerns only"]

**Final Justification:**

I maintain my positive score for the reasons discussed above in the review. The authors have answered the major questions posed.

**Limitations:**

The authors adequately address their limitations (small synthetic experiments), and discuss how the computation cost may pose issues. They provide guidance for the latter issue at the end of Section 6.

**Paper Formatting Concerns:**

I have no major paper formatting issues.

**Quality:**

3

**Strengths And Weaknesses:**

**Strengths**
* The authors provide theoretical guarantees for their approach based on the quality of imputation, which provides nontrivial regret bounds. The authors provide context for these regret bounds relative to existing works, and show that they are competitive with existing optimal bounds.
* The authors provide clear motivations and examples throughout their work. Even without familiarity with the meta-bandit literature, the example in Figure 1 provides clear exposition for applications of this work.
* The authors provide practical guidelines for their method, including implementation details with modern imputation approaches (e.g. neural network models).

**Weaknesses**
* As the authors note, the empirical validation of this approach is too limited to draw significant conclusions, even with the clear outperformance of their approach compared to other methods. It would be interesting to see how these approaches perform when $Z$ is less informative and/or the complexity of the synthetic data-generating process is increased.

---

> ### Author Rebuttal · Authors · 2025-07-30
>
> Thank you for your helpful comments and questions!
>
> ## (1) "Design based" perspective
>
> *It still seems somewhat unclear why the design-based approach (viewing observations as fixed, but unobserved) helps with the interpretation of the approach provided by the authors. It would be nice to have a section discussing the distinction between this "fixed-sample" approach vs. distributional approach, and how this perspective results in the development and superior performance of the authors' approach.*
>
> Thanks for the comment. There are two relevant points, which we will make clearer in the revision.
>
> **(1a) We don't view the outcomes in the table $\tau$ as *fixed*, but rather as drawn from a task distribution.** We first want to clarify that in contrast to design-based inference in which potential outcomes $\tau$ are fixed, we view them as a random variable drawn from a "meta-task" distribution: $\tau \sim p^\*$ (from Eq (1) in the draft). Our decision-making objective is to minimize the regret on average across sampled tasks $\tau \sim p^\*$:
> $$\\mathbb{E} \\left[ \\frac{1}{T} \\sum\_{t=1}^T \\left\\{ R(Y\_t^{(\\pi^\*(X\_t; \\tau))}) - R(Y\_t^{(A\_t)}) \\right\\} \\right].$$
>
> **(1b) Our approach is distributional, and models potentially observable outcomes $\tau$ instead of latent variables.** Standard Thompson Sampling explicitly specifies *latent* environment parameters, places a prior on them, and performs posterior sampling. Performing standard Thompson sampling with neural networks, requires approximate inference methods and can be computationally very challenging [Riquelme et al, Tran et al, Osband et al]. In contrast, our generative Thompson Sampling algorithm maintains a posterior distribution over the *missing outcomes* in the potential outcomes table $\tau$. Practically, a motivation for our approach of directly modeling potentially observable outcomes $\tau$ is that it easily accommodates autoregressive sequence modeling; in contrast, we cannot as easily train a sequence model to predict unobserved latent variables. Our theory reflects the benefit of directly modeling observable outcomes, since the pretraining sequence prediction loss function directly controls the regret.
>
>
> ## (2) Effect of choice of imputation model on performance
>
> We can interpret your question about how "generic" imputation methods perform in our setting in two ways. We provide an answer for each interpretation.
>
> **(2a) "Generic imputation methods" refers to standard methods for imputation used in the missing data literature.** There are a wide variety of imputation methods in the missing data literature, including mean imputation, nearest neighbor imputation, model-based imputation, and Bayesian multiple imputation. Many of the standard imputation methods in the machine learning and statistics literature are not directly appropriate. Our generative Thompson sampling algorithm requires a sequence model be used for imputation that (i) has low offline sequence prediction loss, and (ii) can effectively use high-dimensional task features $Z_\tau$. So, while imputation is an old topic, the feasibility and promise of our approach hinges on recent advances in sequence modeling.
>
> **(2b) "Generic imputation methods" refers to sequence models for imputation that are simpler models than the neural network-based ones we use in our simulations.** Under this interpretation the main concern is understanding how our simulation results may change if we use simpler sequence models for imputation. To address this, we ran additional simulations in our synthetic setting and varied the complexity of the sequence model $p_\theta$ used for imputation. In our original experiments, we used a $p_\theta$ parameterized by a multi-layer perceptron (MLP) with three hidden layers; this model takes as input a vector summarizing the history $\mathcal{H}_t$ and the current context $X_t$, and outputs a distribution over $Y_t$. We ran additional simulation results for an MLP with one hidden layer and a simple logistic regression model (MLP with 0 hidden layers). The results are summarized below, and we will add full results to the draft.
>
> **Table 1: Sequence loss and cumulative regret of TS-Gen in synthetic setting for different sequence imputation models $p_\theta$.** All sequence models $p_\theta$ are pretrained using the same set of previous task data and hyperparameter tuning procedure. The 3 layer MLP results match those already presented in the draft (Figure 5). Sequence loss $\ell(p_{\theta})$ corresponds to Eq (6) in the draft computed on the validation set; we report $\frac{1}{T} \ell(p_{\theta})$, where we divide by $T=500$.
>
> | | Sequence loss $\frac{1}{T} \ell(p_{\theta})$ | Regret t=100 | Regret t=200 | Regret t=300 | Regret t=400 | Regret t=500 |
> |:-----------------------|:------------------|------|------|------|------|------|
> | 3 layer MLP (original) | 0.437             | 8.204| 13.652| 17.978| 21.434| 24.578|
> | 1 layer MLP            | 0.438             | 8.580| 14.474| 19.192| 23.028| 26.502|
> | Logistic               | 0.482             | 8.666| 16.746| 23.776| 29.564| 34.710|
>
>
> In the table above, observe that less expressive sequence models have higher (worse) sequence loss and worse regret.
>
> ## References
> - Osband, Wen, Asghari, Dwaracherla, Ibrahimi, Lu, & Van Roy. *Approximate Thompson Sampling via Epistemic Neural Networks.* UAI 2023
> - Riquelme, Tucker, & Snoek. *Deep Bayesian Bandits Showdown: An Empirical Comparison of Bayesian Deep Networks for Thompson Sampling.* ICLR 2018
> - Tran, Snoek, & Lakshminarayanan. *Practical uncertainty estimation and out-of-distribution robustness in deep learning.* NeurIPS Tutorial 2020

---

> > ### Author Response · Authors · 2025-08-05
> >
> > Hello! We're writing to gently follow up on our rebuttal, as we haven't heard from you yet. If you have any further questions or would like additional clarification, please don't hesitate to let us know. We'd be happy to provide more information.

---

> > ### Comment · Reviewer_ik19 · 2025-08-05
> >
> > Thank for the informative rebuttal - this has helped understand the results of this paper more clearly. In particular, I was hinting at interpretation 2b, and the additional experiments run are quite helpful for understanding how less complex models perform. I will maintain my positive score as is.

---

> ### Author Response · Authors · 2025-08-06
>
> Thank you for your positive comments and for taking the time to engage with our work! We're glad that our rebuttal and the additional experiments helped with understanding our method.

---

### Note · Authors · 2025-08-15

We deeply appreciate the reviewers engagement throughout the review process!

### Key Contributions
We introduce TS-Gen, a generative formulation of Thompson Sampling that models uncertainty as arising from missing but potentially observable outcomes rather than unobservable latent parameters. TS-Gen enables rigorous regret guarantees when using expressive neural networks.
- **Algorithm:** Replaces challenging posterior inference with autoregressive sampling from a sequence model that uses modern neural architectures.
- **Theory:** Novel contextual bandit regret bound depending on VC/Natarajan dimension of the policy class. Regret scales with generative model's offline prediction loss, enabling practical optimization. Handles misspecified Bayesian models with formal guarantees.

### Reviewer Comments and Clarifications
- **ik19:** We clarified design-based perspective, added experiments with simpler models showing graceful performance degradation, demonstrated sequence loss-regret connection.
- **iwTT:** They explicitly stated "reject mostly due to [their] unfamiliarity with the literature." We addressed all technical questions about best-in-class regret definition, learning objective, and potential outcomes framework.
- **y7h5:** We provided proof that our algorithm selects actions via probability matching, i.e., instantiates Thompson Sampling (Proposition 2). We clarified advantages: practical Bayesian learning through sequence learning, complex models without complicated approximate inference, flexible policy classes. We added an additional comparison to Neural Linear TS.
- **BmbW:** We addressed computational concerns via truncated generation, clarified sequence vs policy complexity, explained how bounds accommodate complex generative models.

### Issues Resolved
- **TS-Gen implements TS:** Formally proved TS-Gen is an instantiation of Thompson sampling as it uses probability matching
- **New neural baseline:** Added Neural Linear TS that used the same neural network model as in TS-Gen
- **Computational efficiency:** Showed truncated generation maintains performance
- **Model complexity:** Clarified regret depends on the complexity of the policy class, and not directly on the complexity of the sequence model

The reviewers appreciated the novelty (y7h5,BmbW) and applicability (BmbW) of TS-Gen, and the fact that our regret bounds that reflects quality of the generative model (BmbW) and are nontrivial (ik19). All commented positively on writing clarity.

---

### Decision · Program_Chairs · 2025-09-17

**Decision:**

Accept (poster)

**Comment:**

A generative TS algorithm is proposed, which meta-learns the distribution of tasks and uses the learned model to generate both future and counterfactual outcomes, which are used for optimizing the action-selection policy. The generation model is updated every time a new observation is acquired, and the process of optimizing the action-selection policy, observing rewards, and updating the generation model is repeated all along.
Authors introduce a novel analysis framework in which uncertainty is viewed as arising from missing data rather than from unknown parameters, and the oracle policy to which the algorithm is compared is the one that best fits the complete dataset without any missing data.
The advantage of this new analysis makes the algorithm compatible with complex task environments and complex policies (with finite VC dimension).

Overall, reviewers agreed that the proposed framework is novel, and that the paper is written with clarity. Having read the paper myself, I also find that the newly proposed framework, which views uncertainty as arising from missing data rather than unknown parameters, will provide fresh insights into the analysis of other bandit algorithms as well, thereby making the contribution of this paper significant.

One of the main issues was whether the proposed algorithm can truly be regarded as Thompson sampling, to which the authors responded affirmatively and with clarity, noting that the algorithm implements ‘probability matching’. Authors are encouraged to clarify this point once more in their final version. Also during reviewer-AC discussion period, some reviewers questioned the appropriateness of the definition of the best-fitting policy and the fairness of policy evaluation, with these issues clarified through conversations between the reviewers and the AC. Author are encouraged to provide additional details on the best-fitting policy and to include their new TS baseline that they proposed during the rebuttal (which uses $p_{\theta}$ as feature extractor) in their final version of the paper.

Since the main questions raised by the reviewers were resolved as mentioned above, I recommend acceptance of the paper despite its borderline score.